# PoTRE: Test-Time Reasoning inspired by Cognitive Heterogeneity

**Anmol Kankariya**                                                    *anmolkankariya@google.com*
*Applied AI, Google Cloud*

**Sercan Ö. Arık**                                                      *soarik@google.com*
*Applied AI, Google Cloud*

**Reviewed on OpenReview:** *https://openreview.net/forum?id=wApf83NZmh*

## Abstract

While Large Language Models (LLMs) excel at many tasks, they frequently struggle with complex reasoning that requires long-horizon planning and iterative error correction. Furthermore, standard single-stream prompting proves brittle when models encounter novel abstractions or rigorous domain constraints. We introduce PoTRE (**Po**ly-**T**opological **R**easoning **E**nsembles), a heterogeneous framework that decouples inference into four agents: (1) Adversarial Refinement Agent, (2) Hierarchical strategic Planning Agent, (3) Spectrum Search Agent, and (4) Direct Chain Agent. A final Task-Adaptive Aggregation Layer dynamically reconciles these perspectives—via final candidate selection, semantic synthesis, or neuro-symbolic verification—to produce a robust global solution. We evaluate PoTRE on three frontier benchmarks: ARC-AGI-2, Humanity's Last Exam (HLE), and PRBench Finance. PoTRE achieves state-of-the-art accuracy of 49.92% on HLE, surpassing the previous best official score. We demonstrate that this architectural heterogeneity achieves improved reasoning performance using similar or fewer inference tokens compared to heavily scaled homogeneous baselines.

## 1 Introduction

Scaling Large Language Models (LLMs) has yielded remarkable capabilities across general-purpose tasks, driven largely by the optimization of next-token prediction on massive corpora OpenAI (2024); DeepMind (2025). However, a persistent "reasoning gap" remains between fluent text generation and robust, multi-step problem solving. While test-time reasoning strategies like Chain of thought prompting Wei et al. (2022) and self-consistency Wang et al. (2023) have enhanced the performance of frontier models, they remain fundamentally brittle when facing tasks that require long-horizon planning, rigorous error correction, or novel abstraction—capabilities often categorized as "System 2" thinking Kahneman (2011).

This brittleness is most evident in the emergence of next-generation benchmarks designed explicitly to break simple heuristic pattern matching. *Humanity's Last Exam* (HLE) Hendrycks et al. (2025), for instance, poses multidisciplinary problems at the frontier of human knowledge where retrieval alone is insufficient. Similarly, *ARC-AGI-2* Chollet & the ARC Prize Team (2025) tests fluid intelligence through novel visual abstractions that cannot be solved via memorized recipes, and *PRBench* Research (2025) demands high-stakes professional judgment in finance and law. On these rigorous testbeds, standard single-stream prompting—and even homogeneous ensembles—often fail. We observe that such systems frequently suffer from topological mode collapse: because homogeneous agents share identical probabilistic priors, their errors are highly correlated. Consequently, simply scaling the ensemble size leads to "groupthink", where agents reinforce plausible but incorrect fallacies rather than correcting them via divergent reasoning strategies.

Current research attempts to mitigate this by structuring reasoning into trees or graphs Yao et al. (2023); Besta et al. (2024). However, these methods typically rely on a single reasoning topology—usually the

model's default generation style—replicated multiple times. We draw inspiration from ***cognitive heterogeneity***—the principle that complex problem-solving benefits from diverse, complementary thinking strategies. Building on this inspiration, our major contribution is an engineered heterogeneous pipeline that operationally achieves measurable semantic divergence in candidate answers and rationales by orchestrating distinct reasoning topologies. Unlike homogeneous ensembles where errors are highly correlated, architecturally diverse agents sometimes possess distributional failure modes. We observe this empirically in our ARC-AGI-2 analysis: the Spectrum search agent, despite lower average accuracy, acts as a "Specialist" that uniquely solves 14% of tasks missed by all other agents. Just as human decision-making benefits from the interplay between creative divergence, critical dialectic, and structured planning, LLM reasoning should decouple these processes into specialized agents that fail *differently* to succeed *collectively.*

We introduce **PoTRE** (Poly topological Reasoning Ensembles), an empirical framework and engineered heterogeneous pipeline inspired by cognitive heterogeneity. PoTRE is explicitly designed to orchestrate and evaluate distinct, well-established reasoning topologies. Unlike ensemble methods that simply average outputs from identical agents, PoTRE orchestrates four specialized agents: (1) an *Adversarial Refinement Agent* for debate, (2) a *Hierarchical strategic planning agent* for sub-goal decomposition, (3) a *Spectrum search agent* for breadth-first candidate search, and (4) a *Direct Chain Agent.* A final *Synthesis Agent* aggregates these candidate answers via task-adaptive aggregation. Whether through final candidate synthesis or neurosymbolic verification, PoTRE filters out agent-specific hallucinations to construct a robust global solution.

This evaluation allows us to contrast PoTRE's approach—orchestrating parallel breadth through diverse reasoning topologies—against the sequential depth optimized by current state-of-the-art systems. While Deep Research systems typically rely on long-context centralization Cai et al. (2026) or iterative homogeneous refinement Tang et al. (2026) using heavier frontier-model backbones (e.g., Gemini-3-Pro-Preview), PoTRE offers a distinct architectural alternative. Our results on the HLE Open-Book benchmark with Gemini-3-Flash-Preview (55.28% vs. 51.7%/52.2%) demonstrate that systematically combining distinct reasoning strategies in parallel is a highly effective method for decoupling performance from parameter scale. This establishes PoTRE as a robust empirical framework for maximizing the reasoning potential of lightweight models.

We evaluate PoTRE on three challenging reasoning benchmarks. As shown in Figure 1, our results demonstrate a significant performance lift from scaffolding. While it is a well-established result that spending substantially more test-time compute on a smaller model can enable it to outperform a larger model evaluated in a single pass (Snell et al., 2024), we validate the efficacy of this principle within our architecture. Specifically, we show that the smaller, highly efficient Gemini-3-Flash-Preview model, when augmented with our structured PoTRE framework, consistently surpasses the standalone Gemini-3-Pro-Preview baseline. On Humanity's Last Exam, PoTRE achieves a state-of-the-art accuracy of 49.92%. On ARC-AGI-2, our method scores 38.30%, nearly doubling the baseline performance. Finally, to demonstrate PoTRE's adaptability to complex, domain-specific reasoning, we establish a new state-of-the-art on the PRBench Finance Hard benchmark with an average clipped score of 0.5196.

Crucially, we also address the fundamental trade-off of inference-time compute. By mapping the cost-performance Pareto frontier of our architecture, we demonstrate that PoTRE is highly modular. We show that strategically pruning specific sub-agents tailored to the target domain can reduce token consumption by up to 85% while surprisingly *improving* overall accuracy by mitigating synthesis interference. These findings highlight the ability of our engineered heterogeneous pipeline to synthesize complex, conflicting information into accurate insights. Ultimately, we demonstrate that a structured allocation of test-time compute across diverse reasoning topologies consistently outperforms homogeneous allocations under comparable or reduced token budgets.

## 2 Related Work

Standard approaches to LLM reasoning rely on prompt engineering to induce intermediate computation. Chain of thought Wei et al. (2022) demonstrated that prompting models to emit reasoning steps significantly improves performance. Subsequent works extended this via Self-Consistency Wang et al. (2023), which marginalizes over multiple reasoning paths to filter stochastic errors.

Figure 1: Comparison of the smaller Gemini-3-Flash-Preview model equipped with PoTRE (Blue) against the significantly larger Gemini-3-Pro-Preview baseline (Grey) across three diverse benchmarks. In all cases, the architectural scaffolding of PoTRE enables the efficient model to outperform the frontier model operating with prior work prompts, demonstrating that inference-time reasoning structure can effectively substitute for parameter scale.

Recent methods have moved beyond linear chains toward non-linear topologies. Tree of Thoughts Yao et al. (2023) and Graph of Thoughts Besta et al. (2024) allow models to explore search trees and aggregate thoughts. However, we argue that these methods suffer from homogeneous failure modes: because they replicate the same underlying reasoning structure across all branches, they are vulnerable to correlated errors. If the model possesses a specific blind spot, it generates consistent but incorrect reasoning across the entire topology.

Recent advances in "Deep Research" have shifted focus from single-turn QA to long-horizon autonomous information seeking. Current approaches largely fall into two architectural paradigms: Sequential Depth and Homogeneous Refinement. Systems like Yunque DeepResearch Cai et al. (2026) and Tongyi Deep-Research Team et al. (2025) rely on centralized, compute-heavy planning. Yunque employs a hierarchical "Manager-Worker" topology with dynamic context folding to handle long horizons, while Tongyi utilizes end-to-end reinforcement learning to optimize tool-use trajectories.

Parallel efforts focus on improving reasoning via iterative self-correction. ReThinker Tang et al. (2026) introduces a "Solver-Critic-Selector" loop that permutes confidence scores to filter hallucinations. Similarly, SETS Chen et al. (2025a) unifies "Sampling, Self-Verification, and Self-Correction" into a framework that outperforms simple voting. However, both rely on homogeneous sampling—refining a single model's outputs rather than diversifying the reasoning topology itself.

In the multi-agent domain, MonoScale Shao et al. (2026) proposes a router-based framework to delegate sub-tasks, optimizing for "monotonic improvement" (performance stability) within an agent pool. TUMIX Chen et al. (2025b) extends this by mixing agents with distinct tool-use strategies. Yet, these approaches generally optimize a single reasoning topology. They risk topological mode collapse when the underlying model shares identical blind spots, as they do not simultaneously orchestrate fundamentally different reasoning strategies (e.g., adversarial debate alongside hierarchical planning) to mitigate the inherent biases and correlated errors of any single method.

Rather than mitigating these biases through diverse inference topologies, other recent advancements have attempted to solve the multi-agent orchestration problem by training dedicated routing models. A notable example is ToolOrchestra Su et al. (2025), which introduces a lightweight orchestrator model explicitly trained via Reinforcement Learning to dynamically route complex queries to specialized tools and frontier models. While ToolOrchestra demonstrates impressive gains on challenging benchmarks, it requires a highly specialized training pipeline, extensive data synthesis, and custom reward modeling to align the orchestrator's behavior.

Parallel to these orchestration efforts, significant progress in model reasoning has been driven by data-centric training strategies. Notably, OpenThoughts Guha et al. (2025) demonstrates that highly capable reasoning models can be developed through rigorous Supervised Fine-Tuning (SFT) data recipes. By systematically optimizing question sourcing, multi-answer sampling, and reasoning trace distillation from teacher models, OpenThoughts produced competitive open-source models. However, data-centric approaches fundamentally require expensive curation pipelines and permanent weight modifications to the underlying base model, posing practical difficulties.

In contrast to these approaches, PoTRE is explicitly designed as an empirical and engineering framework inspired by cognitive heterogeneity. Instead of optimizing solely for sequential depth or homogeneous refinement, PoTRE structurally enforces topological diversity by orchestrating four distinct reasoning agents in parallel. Our results demonstrate that this "width-over-depth" approach allows lightweight models to outperform the heavy-compute baselines of the sequential paradigm, effectively decoupling reasoning performance from parameter scale.

## 3 Methodology

We introduce PoTRE (Poly-Topological Reasoning Ensembles), a heterogeneous multi-agent framework designed to solve complex reasoning tasks by decoupling reasoning topologies into parallel reasoning agents. Unlike homogeneous ensembles that aggregate identical agents, PoTRE employs four distinct reasoning agents—Adversarial Refinement Agent via debate, Hierarchical Strategic Planning Agent, Spectrum Search Agent, Direct Chain Agent—to maximize the diversity of the solution space.

Given an input problem $P$, the system distributes the task in parallel to four independent sub-agents. The resulting set of intermediate solutions $\mathcal{S} = \{s_{debate}, s_{planning}, s_{spectrum}, s_{direct}\}$ is then aggregated by a final Synthesis agent to produce the optimal solution $\hat{y}$.

### 3.1 The Agents

Each agent is designed to target specific failure modes. The parallel orchestration and subsequent synthesis of these agents are detailed in Algorithm 1 and algorithms for each sub-agent is given in Appendix C

#### 3.1.1 Adversarial Refinement Agent

This agent addresses hallucination through iterative adversarial refinement. Two agents—a *Proposer* and a *Verifier*—engage in a multi-turn dialogue (up to $T = 5$ turns) to scrutinize and iteratively improve the solution quality. Rather than merely filtering outputs, the Verifier provides constructive critique that guides the Proposer toward a more robust answer.

The process follows a strict consensus:

- **Proposal & Critique:** The Proposer generates an initial candidate (rationale and answer). The Verifier acts as a rigorous critic, scrutinizing the output for logical gaps, factual inconsistencies, or constraint violations.

- **Iterative Refinement:** If the Verifier rejects the candidate, it provides specific feedback. The Proposer must then generate a revised candidate that addresses these critiques.

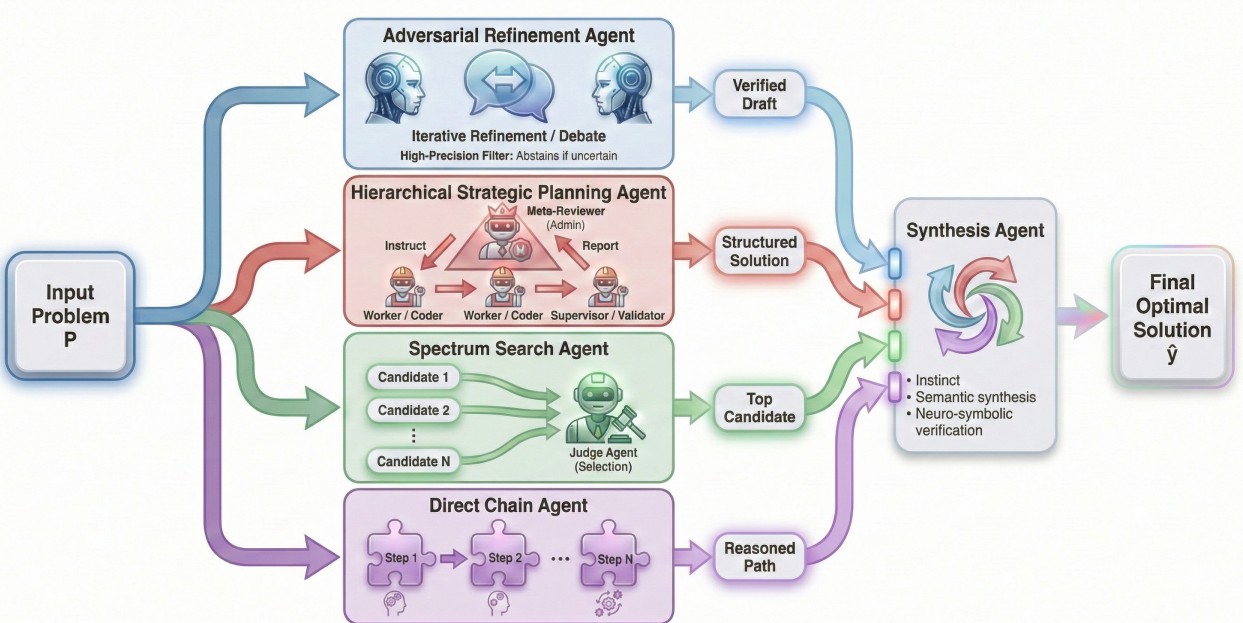

Figure 2: **Poly-Topological Reasoning Ensembles**. This diagram illustrates the proposed multi-agent reasoning process. An input problem is processed in parallel by four distinct agent-based strategies: (1) Adversarial Refinement, involving iterative debate for high-precision output, (2) Hierarchical Strategic Planning, using a nested team structure for structured solutions, (3) Spectrum Search, generating and selecting top candidate via a judge agent, and (4) Direct Chain for standard sequential reasoning. The outputs from these diverse reasoning paths (verified draft, structured solution, top candidate, and reasoned path) are aggregated by a centralized Synthesis Agent, which utilizes final candidate selection
, semantic synthesis, and neuro-symbolic verification to generate the final optimal solution.

- **Final Consensus:** The answer is only accepted if the Verifier explicitly outputs a `STATUS: APPROVED` token.

**Selective Abstention:** This agents operate on a strict "Pass/Fail" basis. If the Verifier remains unsatisfied after the maximum turns, the agent abstains (returning $\emptyset$). This agent ensures that only rigorously verified solutions are propagated to the Synthesizer, preventing the contamination of the final decision process with uncertain outputs.

### 3.1.2 Hierarchical Strategic Planning Agent

This agent implements a Hierarchical reasoning architecture designed for long-horizon planning and strategic course correction, the system functions as a specialized role-based team:

- **Planner:** Acts as the strategic core that structures the reasoning process prior to execution. Depending on the input modality, it either decomposes complex queries into a sequence of 2–3 logical sub-goals or abstracts a single universal transformation rule. This component enforces dynamic derivation, prioritizing the identification of global patterns (such as visual legends or alignment keys) over case-specific hard-coding.

- **Executor:** Implements the proposed solution, producing executable traces or detailed logical derivations.

- **Verifier:** Evaluates the execution against the Planner's proposal, strictly enforcing constraint satisfaction.

---

**Algorithm 1** PoTRE: Core Orchestration and Task-Adaptive Synthesis

---

**Require:** Input query $Q$, Task Type $\mathcal{T}_{type} \in \{$Constrained, Open-Ended, Rule-Based$\}$, Model $\mathcal{M}$
**Ensure:** Final synthesized answer $A^*$

 **% Phase 1: Parallel Agent Triggering (Independent Execution)**
1:  $s_{direct} \leftarrow$ Direct_Chain_Agent$(Q, \mathcal{M})$           ▷ Sequential step-by-step logic
2:  $s_{spectrum} \leftarrow$ Spectrum_Search_Agent$(Q, \mathcal{M})$          ▷ Solution diversity
3:  $s_{debate} \leftarrow$ Adversarial_Refinement_Agent$(Q, \mathcal{M})$        ▷ Adversarial refinement
4:  $s_{planning} \leftarrow$ Hierarchical_Strategic_Planning_Agent$(Q, \mathcal{M})$    ▷ Planner → Executor → Verifier

 **% Phase 2: Output Collection**
5:  $\mathcal{S} \leftarrow \{s_{direct}, s_{spectrum}, s_{debate}, s_{planning}\}$

 **% Phase 3: Synthesis Agent (Task-Adaptive Aggregation)**
6:  **if** $\mathcal{T}_{type} ==$ Constrained **then**
7:   $A^* \leftarrow$ Candidate_Selection$(Q, \mathcal{S}, \mathcal{M})$         ▷ Final Candidate selection
8:  **else if** $\mathcal{T}_{type} ==$ Open-Ended **then**
9:   $A^* \leftarrow$ Qualitative_Synthesis$(Q, \mathcal{S}, \mathcal{M})$        ▷ Semantic Synthesis
10:  **else if** $\mathcal{T}_{type} ==$ Rule-Based **then**
11:   $A^* \leftarrow$ Neuro_Symbolic_Verifier$(Q, \mathcal{S}, \mathcal{M})$      ▷ Logic-Consistency validation
12:  **end if**
13:  **return** $A^*$

---

- **Overseer:** Acts as a dynamic control agent: if the worker agents get stuck repeating the same error or fail to make progress, the Overseer intervenes to halt the current path and force the Planner to generate a fresh strategy.

**Active Meta-Reasoning Control Loop.** To mitigate the diminishing returns of unguided self-correction, we employ an agent of autonomous strategic pivoting. The "overseer" performs continuous state-space monitoring on the dialogue history:

1. **Trajectory Analysis:** The system scans for "Strategic Failures," such as repeated hypothesis cycling, logical circularity, or recurring execution errors.

2. **Diagnosis:** Upon detecting a failure mode, the system isolates the root cause—distinguishing between operational faults (e.g., calculation or syntax errors) and strategic misalignment (e.g., getting stuck in a local optimum).

3. **Strategic Intervention (The Pivot):** Upon diagnosing a strategic stall, the overseer executes a two-stage intervention:

   - *Hypothesis Rejection:* It explicitly invalidates the current reasoning path (e.g., "The assumption that color dictates movement is leading to contradictions; abandon this premise").
   - *Directive Synthesis:* Instead of selecting from a pre-defined list of error codes, the model dynamically generates a high-level natural language instruction tailored to the specific failure context (e.g., "Re-analyze the grid using connectivity-based object detection instead of pixel-wise color matching").

   This agent forces a hard transition from passive debugging (fixing code) to active re-planning (changing logic), effectively breaking the agent out of recursive loops without manual intervention.

### 3.1.3 Spectrum Search Agent

To address the cascading logical errors prevalent in single-trajectory generation, we employ a parallelized ensemble approach as a form of Test-Time Scaling Snell et al. (2024). This agent prioritizes breadth of exploration over depth of planning:

**Divergent Candidate Generation.** To ensure diverse solution discovery, the system spawns a parallel batch of $N$ independent worker threads. Each worker generates a unique candidate solution using a non-zero temperature. This stochastic sampling allows the system to escape local optima that typically trap deterministic greedy decoding strategies Wang et al. (2023).

**Task-Specific Candidate Selection.** To select a single final answer ($s_{spectrum}$) from the $N$ independent agents, we apply a filtering strategy that adapts to the nature of the specific task:

- **Constraint-Guided Filtering:** For tasks that feature formal test cases or verifiable rules, we leverage the given premises as a "weak verifier" to perform a multi-stage refinement process:

  1. *Hypothesis Verification:* First, the reasoning pipeline enforces a rigorous negative constraint check. Before finalizing a candidate solution, the agent must verify that its proposed logic holds conceptually true against all provided problems and boundary conditions. This internal critique loop filters out logical inconsistencies early, ensuring only robust, fully validated hypotheses proceed to the execution phase.
  2. *Execution Pruning:* When applicable, the validated hypotheses are translated into an executable format (e.g., programmatic code or formal logic) and evaluated against the known constraints. Any candidate that fails to strictly reproduce the expected target conditions is pruned as functionally invalid.
  3. *Consensus Selection:* From the surviving subset of verified candidates, we select the final answer via majority voting. This funnel ensures that consensus is built only upon candidates that have passed both conceptual (rule-based) and functional (execution-based) verification.

- **Semantic Judging:** For open-ended reasoning where no execution environment exists, we rely on a LLM as a Judge. This agent evaluates candidates based on internal logical consistency and adherence to problem constraints, selecting the single most robust solution without external execution signals.

### 3.1.4 Direct Chain Agent

To ensure the system remains grounded in direct reasoning, we retain a standard Chain of thought. This agent produces a solution $s_{direct}$ using step-by-step reasoning, serving as a baseline anchor for the more complex agentic systems. We employ a task-adaptive prompting strategy: Zero-Shot CoT Kojima et al. (2023) for open-ended reasoning to avoid biasing the model with potentially irrelevant examples, and Few-Shot CoT for inductive tasks where input-output demonstrations are essential for defining the problem space.

### 3.2 Synthesis Agent: Task-Adaptive Aggregation

The final component of PoTRE is the Synthesis Agent. Recognizing that different problem structures require distinct evaluation strategies, we employ a task-adaptive approach that selects the optimal synthesis method based on the expected output modality:

- **Final Candidate Synthesis for Constrained Outputs:** For tasks requiring highly specific or constrained responses (e.g., short-answer or multiple-choice formats), we intentionally transition away from rigid majority voting. Instead, we utilize an LLM-based comparative synthesis strategy. Rather than merely counting exact textual matches, the Synthesis Agent actively evaluates the reasoning traces within the candidate set $\mathcal{S}$. By comparing the underlying logical rationales, the agent identifies the most robust and well-supported conclusion, effectively filtering out flawed, consensus-driven answers.

- **Qualitative Synthesis for Open-Ended Generation:** For complex reasoning tasks requiring long-form professional judgment, exact-match statistical voting is inherently ill-defined, as no two reasoning trajectories are identical. In these scenarios, we deploy a qualitative integration mechanism to critically evaluate the independent answers from agents, resolve conflicting viewpoints, and merge their most verifiable arguments into a single, cohesive final response.

- **Logic-Consistency Verification for Rule-Based Tasks:** For tasks rooted in formal logic, pattern induction, or strict constraint satisfaction, we employ a neuro-symbolic verifier, it deeply evaluates the generalization logic of the surviving candidates. It is prompted to verify whether the proposed solution consistently and flawlessly applies the core rules derived from the problem's examples, prioritizing candidates that demonstrate high structural and logical coherence.

## 4 Experimental Setup

### 4.1 Benchmarks

We evaluate PoTRE across three diverse benchmarks. For each dataset, we utilize official evaluation splits to ensure comparability with public leaderboards.

- **Humanity's Last Exam (HLE):** Introduced by the Center for AI Safety Hendrycks et al. (2025), HLE represents the current ceiling for closed-ended academic reasoning, filtering out contamination by focusing on questions that require expert-level synthesis across 2,500 diverse subjects. We evaluate on the full dataset (N = 2, 500), covering domains from humanities to advanced STEM.

- **ARC-AGI-2:** Building on the original Abstraction and Reasoning Corpus, ARC-AGI-2 Chollet & the ARC Prize Team (2025) targets fluid intelligence. Unlike language-heavy tasks, ARC requires inducing programs from few-shot visual examples, a domain where standard CoT often fails due to a lack of grounded priors. We evaluate on a full public evaluation set of 120 tasks.

- **PRBench:** Scale AI's Professional Reasoning Benchmark Research (2025) evaluates performance in high-stakes domains (Finance and Law) using expert-curated rubrics rather than simple exact-match accuracy, testing a model's ability to adhere to rigorous constraints. We specifically utilize the "Hard" subset ($N = 300$ samples) to rigorously test the model's ability to apply specialized financial knowledge.

### 4.2 Baselines

To strictly isolate the impact of the PoTRE architecture from prompt engineering variables, we take prompts from prior work for baseline comparisons. Rather than using naive zero-shot prompts, we utilize state-of-the-art prompt templates derived from recent literature for both models Zhang et al. (2025b;a).

- **Baseline Models:** We use Gemini-3-Flash-Preview, Gemini-3-Pro-Preview and Gemini-3.1-Pro-Preview as small and large versions of Gemini 3 with prior work prompts.

- **Baseline Models:** We use Gemini-3-Flash-Preview, Gemini-3-Pro-Preview and Gemini-3.1-Pro-Preview as small and large versions of Gemini 3 with prior work prompts. All models used in this study have been released publicly, and their details can be found in the official Google DeepMind model cards [1].

- **Self-Consistency (SC) Baselines:** To distinguish the impact of reasoning topology from simple ensemble scaling, we evaluate all versions using Self-Consistency at two distinct scales ($N = 8$ and $N = 16$) Wang et al. (2023). We generate up to 16 independent trajectories per problem and aggregate results via majority voting. This serves as a rigorous sampling baseline, allowing us to strictly verify whether PoTRE's gains arise from *engineered heterogeneous pipeline* (orchestrating diverse reasoning topologies) rather than just *homogeneous repetition.*

- **Official Leaderboard Benchmarking:** To ensure external validity, we benchmark PoTRE against the official public leaderboards for *Humanity's Last Exam* (HLE) and the *PRBench* "Finance Hard" subset. This aligns our evaluation with the current industrial state-of-the-art, allowing us to strictly verify PoTRE's performance against the highest-performing commercial systems in a standardized, reproducible setting. Furthermore, we are not comparing ARC-AGI 2 with leaderboard because leaderboard evaluates on semi-private dataset and we are using public evaluation set.

---

[1] https://deepmind.google/models/model-cards/

### 4.3 Evaluation Protocol

To ensure robust and fair scoring, we strictly adhere to the official evaluation protocols for each benchmark:

- **ARC-AGI-2:** Evaluation is performed using standard Programmatic Validation. A solution is scored as correct only if the generated python script produces the exact pixel-perfect output grid for the test input. We do not use partial credit; the validation is strictly binary and executed in a deterministic environment.

- **Humanity's Last Exam (HLE):** Given the complexity of the dataset, we utilize a blind LLM-as-a-Judge (Gemini-3-Flash-Preview) to grade responses. The judge evaluates semantic equivalence between the predicted answer and ground truth, robustly handling variations in formatting (e.g., "50%" vs "0.5") to determine final accuracy. We utilize the official prompt for evaluation provided by the authors Hendrycks et al. (2025).

- **PRBench Finance:** To ensure strict comparability, we utilize the official evaluation script provided by the benchmark authors Research (2025), modifying only the inference backend to target our evaluated models (Gemini-3-Pro-Preview-Preview).

  The scoring logic follows the formal definition in Appendix D.1 Research (2025) of the original benchmark. For a model $M$ producing response $m_j$ to prompt $p_j$, an LLM judge assigns binary indicators $I_{j,i} \in \{0, 1\}$ for each rubric criterion with weight $w_{j,i}$. The score for a single response $s_j$ is calculated as:

$$s_j = \frac{\sum_{i=1}^{k_j} w_{j,i} I_{j,i}}{\sum_{i:w_{j,i}>0} w_{j,i}} \tag{1}$$

  The final score $S(M)$ is the mean over all $n$ prompts, clipped at zero to account for potential negative penalties:

$$S(M) = \max\left(0, \frac{1}{n}\sum_{j=1}^{n} s_j\right). \tag{2}$$

### 4.4 Implementation Details

To evaluate the scalability and efficiency of our framework, we implemented PoTRE using different backbone models. The architecture was orchestrated using an asynchronous parallel execution pipeline with distinct behavioral configurations for each agent:

- **Role-Conditioned Prompting:** Each agent was instantiated with a specialized system prompt defining its functional role. Full prompt templates for all agents are provided in Appendix A. This ensures that the multi-agent system generates diverse candidates through fundamental differences in problem-solving perspectives rather than mere stochasticity.

- **Modality-Specific Execution:** The framework adapts its output format to the intrinsic nature of each benchmark. For HLE and PRBench, agents generate natural language reasoning terminating in a final text option. For ARC-AGI-2, we deployed the Neuro-Symbolic interface: all four agents were instructed to generate python code. These scripts were executed in a secured sandboxed environment, with execution results (and errors) fed back to the agents, ensuring candidates were grounded in algorithmic logic rather than visual hallucination.

- **Operational Constraints:** To prevent unconstrained generation, we enforced strict turn-based limits tailored to each agent's depth: *Adversarial Refinement Agent* was capped at $T = 5$ turns (to force rapid consensus), while the *Hierarchical Strategic Planning Agent* was allocated single turn for both HLE and PRBench but ARC-AGI-2 was allocated $T = 10$ turns (to allow code fixes and refinement), *Spectrum Search Agent* is set to generate N=8 candidates and *Direct Chain Agent* is capped to generate single answer only.

Table 1: Performance comparison of PoTRE against established baseline prompts from prior work and Self-Consistency (SC) with 8 and 16 samples. Evaluation metrics comprise *Accuracy* for HLE and ARC-AGI 2, and *Average Clipped Score* for PRBench. All reported results denote Pass@1 performance. Key Insight: PoTRE demonstrates consistent improvements over SC baselines across most settings (with one exception for the Pro model on ARC-AGI 2). Notably, our four-agent scaffolding provides substantial lift(†), enabling the PoTRE-augmented Flash models to outperform the un-augmented Pro baselines.

| Benchmark | Model | Baseline Evaluation | | | PoTRE Evaluation | |
| | | Prior Work Prompts | SC ($N = 8$) | SC ($N = 16$) | Score | Improv.[‡] |
| --- | --- | --- | --- | --- | --- | --- |
| Humanity's Last Exam | Gemini-3-Flash-Preview | 35.36% | 39.12% | 37.32% | **39.80%**[†] | +4.44 |
| | Gemini-3-Pro-Preview | 36.88% | 39.00% | 40.20% | **44.00%** | +7.12 |
| | Gemini-3.1-Pro-Preview | 42.15% | 46.15% | 45.95% | **49.92%** | **+7.77** |
| ARC-AGI 2 | Gemini-3-Flash-Preview | 19.16%[*] | 33.33% | 36.67% | **38.33%**[†] | **+19.17** |
| | Gemini-3-Pro-Preview | 21.66%[*] | 22.50% | 28.33% | **33.33%** | +11.67 |
| | Gemini-3.1-Pro-Preview | 59.16% | 75.00% | **80.00%** | 78.33% | **+19.17** |
| PRBench Finance Hard | Gemini-3-Flash-Preview | 0.2357 | 0.1869 | 0.1888 | **0.3486**[†] | +0.1129 |
| | Gemini-3-Pro-Preview | 0.2603 | 0.2352 | 0.2427 | **0.3319** | +0.0716 |
| | Gemini-3.1-Pro-Preview | 0.2713 | 0.2404 | 0.2421 | **0.3931** | **+0.1218** |

[†]Indicates Scaffolding Lift (Flash PoTRE > Gemini-3-Pro-Preview Baseline).

[‡]Improvement calculated vs. Baseline (Prior work prompts).

[*]Prior work prompts degraded performance on ARC-AGI; standard CoT performs better (see Component Analysis).

- **Adaptive Aggregation:** The final Synthesis agent receives the original problem context $P$ and the set of candidate solutions $\mathcal{S}$. It applies a task-specific protocol to resolve conflicts: for open-ended tasks (e.g., PRBench), it integrates the strongest verifiable arguments into a cohesive response; for constrained tasks (e.g., HLE), it simply selects one candidate based on given candidates answer and their rationale; and for spatial reasoning (e.g., ARC), it employs neuro-symbolic verification to confirm logical consistency. This ensures the global solution is strictly grounded in the most robust signal available—whether that is semantic coherence, intuitive logic, or formal constraint satisfaction.

## 5 Results

We evaluate PoTRE across three benchmarks, comparing performance against standard single-agent baselines and Self-Consistency ensembles at two scales ($N = 8$ and $N = 16$) using Gemini-3-Pro-Preview, Gemini-3-Flash-Preview, and Gemini-3.1-Pro-Preview The results, summarized in Table 1, demonstrate that our heterogeneous multi-agent architecture generally yields superior returns compared to simply scaling homogeneous samples.

Notably, PoTRE outperforms the computationally expensive Self-Consistency ($N = 16$) baseline in all except one experimental settings in the Gemini-3.1-Pro-Preview model on ARC-AGI-2, where the $N = 16$ ensemble (80.00%) marginally surpasses the PoTRE system (78.33%).

### 5.1 Performance Analysis

**State-of-the-Art on HLE** On *Humanity's Last Exam* (HLE), PoTRE utilizing the Gemini-3.1-Pro-Preview backbone achieved an accuracy of **49.92%**, surpassing the baseline of 42.15% by a margin of **7.77** percentage points. These results represents a new state-of-the-art on the official leaderboards.[2]. Even

---

[2]Official Leaderboard: `https://scale.com/leaderboard/humanitys_last_exam` and `https://agi.safe.ai/`

Table 2: **Component Analysis on HLE with** $N = 2500$ **samples.** The "Oracle" row represents the theoretical upper bound where at least one agent produced the correct answer. Key Insight: We observe a divergence in optimal reasoning topology: the Pro models (Gemini-3-Pro-Preview and 3.1-Pro) benefit most from the spectrum search agent, while Gemini-3-Flash-Preview achieve their highest accuracy via the Adversarial Refinement Agent. Furthermore, PoTRE Final Synthesis successfully and consistently outperforms the best individual reasoning agent across all evaluated models, though a gap remains compared to the Oracle upper bound.

| | Gemini-3-Flash-Preview | | Gemini-3-Pro-Preview | | Gemini-3.1-Pro-Preview | |
|---|---|---|---|---|---|---|
| Sub-Agents | Solved | Accuracy | Solved | Accuracy | Solved | Accuracy |
| Adversarial Refinement Agent | **948** | **37.92%** | 978 | 39.12% | 1175 | 47.00% |
| Hierarchical strategic planning agent | 883 | 35.32% | 976 | 39.04% | 1134 | 45.36% |
| Spectrum search agent | 921 | 36.84% | **1048** | **41.92%** | **1210** | **48.40%** |
| Direct chain agent | 870 | 34.80% | 949 | 37.96% | 1018 | 40.72% |
| **Oracle Upper Bound** | **1199** | **47.96%** | **1269** | **50.76%** | **1399** | **55.96%** |
| PoTRE Final Synthesis | 995 | 39.80% | 1100 | 44.00% | 1248 | 49.92% |

with the smaller Flash model augmented with PoTRE reached 39.80%, outperforming the standard Gemini-3-Pro-Preview baseline.

**Scaffolding Lift on ARC-AGI** A striking finding is observed on the ARC-AGI 2 benchmark. Although the Gemini-3-Pro-Preview baseline (21.66%) initially outperformed the standalone Gemini-3-Flash-Preview model (19.16%), PoTRE effectively reversed this. When augmented with PoTRE, the Flash model's performance jumped to 38.33%, a massive +19.17 percentage points improvement over its baseline. Crucially, this result outperforms even the larger $N = 16$ self-consistency ensemble (36.67%), confirming that architectural diversity yields significantly higher returns than simple sampling scale.

## 5.2 Ablation Study: Contribution of Individual agents

To isolate the source of PoTRE's performance gains, we analyzed the standalone accuracy of each agent before generating the final answer.

### 5.2.1 Humanity's Last Exam (HLE)

Table 2 presents the component breakdown for HLE ($N = 2500$). The results reveal a significant "Oracle Gap": while the PoTRE agents collectively generated correct solutions for more than 50% of the dataset (Oracle: 55.96%), the realized performance was constrained by selection dynamics.

Our component analysis reveals a distinct divergence in optimal reasoning topologies across different model tiers. Specifically, Gemini-3-Flash-Preview model achieve their peak single-agent performance by leveraging the Adversarial Refinement Agent (37.92%). In contrast, the more capable Gemini-3-Pro-Preview and Gemini-3.1-Pro-Preview models benefit most from the broad exploration provided by the Spectrum Search Agent (yielding 41.92% and 48.40% accuracy). Crucially, the PoTRE Final Synthesis strictly and consistently outperforms the best individual agent for every evaluated model. We attribute this systematic lift directly to our final candidate selection. By framing the final answer resolution as a zero-shot—explicitly instructing the synthesis agent to simply select the candidate answer based on the given rationale without generating further deliberative text—the system significantly mitigates the verification bottleneck. This forces the model to rely purely on its strong discriminative priors rather than over-analyzing and rejecting valid paths. However, while PoTRE successfully drives final accuracy higher (e.g., 49.92% for Gemini-3.1-Pro-Preview), the remaining gap to the theoretical Oracle upper bound (55.96%) highlights that perfect candidate routing remains an open challenge.

Table 3: **Component Analysis on PRBench Finance (Hard, $N = 300$).** Performance breakdown of individual agents compared to the final synthesis. PoTRE consistently yields the highest Average Clipped scores across all evaluated models, highlighted by agent specific gains and significant lift by scaffolding for Gemini-3-Flash-Preview where it outperforms Gemini-3-Pro-Preview in all cases.

| | Gemini-3-Flash-Preview | | Gemini-3-Pro-Preview | | Gemini-3.1-Pro-Preview | |
|---|---|---|---|---|---|---|
| Sub-Agents | Avg Norm | Avg Clip | Avg Norm | Avg Clip | Avg Norm | Avg Clip |
| Adversarial Refinement Agent | **0.3575** | **0.3068** | 0.3550 | 0.3041 | **0.3916** | **0.3444** |
| Hierarchical Strategic Planning Agent | 0.2839 | 0.2257 | 0.3240 | 0.2705 | 0.3666 | 0.3181 |
| Spectrum Search Agent | 0.2456 | 0.1880 | 0.3695 | 0.3198 | 0.3913 | **0.3444** |
| Direct Chain Agent | 0.3433 | 0.2919 | **0.3752** | **0.3266** | 0.3679 | 0.3176 |
| PoTRE Final Synthesis | **0.3958** | **0.3486** | **0.3804** | **0.3319** | **0.4363** | **0.3931** |

### 5.2.2 PRBench Finance

Table 3 details our component analysis on the domain-specific PRBench Finance dataset, reinforcing the overarching trend of multi-agent superiority while revealing key differences in optimal single-agent topologies. The Gemini family exhibits varied preferences (Gemini-3-Flash-Preview peaks via the Adversarial Refinement Agent at 0.3068, while Gemini-3-Pro-Preview relies predominantly on Direct Chain Agent at 0.3266). Despite these differing baselines, the PoTRE Final Synthesis strictly outperforms the best individual reasoning agent for every evaluated model. We attribute this systematic lift to our Synthesis agent. Unlike the discriminative, final candidate selection utilized for general QA tasks, complex financial evaluations require constructive fusion. By prompting the synthesis agent as an "Expert Finance Judge," the system dynamically constructs a comprehensive "Super-Answer"—extracting the strongest computational, analytical, and formatting elements from all sub-agents while actively performing sanity checks on conflicting data. Furthermore, this ablation clearly illustrates the lift observed by scaffolding: the smaller Gemini-3-Flash-Preview model, when equipped with the PoTRE framework (0.3486), successfully surpasses both the baseline CoT and the fully scaffolded implementation of the larger Gemini-3-Pro-Preview model (0.3319). This indicates that robust, constructive multi-agent synthesis can effectively compensate for underlying parameter scale in specialized, highly technical domains.

### 5.2.3 ARC-AGI 2

Table 4 details our component analysis on the ARC-AGI 2 benchmark, revealing a stark contrast in how different models navigate non-linguistic reasoning. A striking observation is the critical role of the Spectrum Search Agent as a "specialist" for generating diverse structural hypotheses. This is most pronounced with Gemini-3-Flash-Preview, which relied heavily on this divergent exploration to crack spatial tasks that baffled the rest of the ensemble, achieving 35 standalone solves with a remarkable 17 being entirely exclusive. Despite this extreme variance in single-agent capabilities, the PoTRE Final Synthesis consistently outpaces the best standalone agent for every model evaluated (e.g., boosting Gemini-3-Flash-Preview from 35 to 46, and Gemini-3.1-Pro-Preview from 89 to 94). We attribute this robust lift to the active verification mechanics embedded in our synthesis agent. Rather than employing a naive majority vote, the synthesis agent utilize a neuro-symbolic approach. By explicitly prompting the agent to verify candidate grids against the task's provided training examples, the system effectively tests and validates the inferred rules against known input-output pairs before finalizing an answer. The synthesis agent dynamically selects flawless candidates, corrects divergent errors, or derives the solution from scratch. By restricting the final output to a strict JSON grid without textual explanation, we suppress hallucinated justifications, successfully harnessing the ensemble's diverse topological strengths to drive performance higher on this notoriously resistant benchmark.

**Pass@2 Performance on ARC-AGI 2:** Because the official ARC-AGI leaderboard permits two submission attempts per task, we extend our evaluation to include Pass@2 performance metrics. Under this relaxed constraint, all evaluated models demonstrated notable gains. Most prominently, Gemini-3.1-Pro-Preview

Table 4: **Component Analysis on ARC-AGI 2 (**$N = 120$**).** PoTRE Final Synthesis consistently elevates overall accuracy beyond the best standalone agent for every evaluated model. Furthermore, we observe the Spectrum Search Agent acting as a critical "specialist," uniquely solving hard spatial tasks that completely baffled the rest of the ensemble.

| Sub-Agents | Gemini-3-Flash-Preview | | Gemini-3-Pro-Preview | | Gemini-3.1-Pro-Preview | |
|---|---|---|---|---|---|---|
| | Standalone | Exclusive | Standalone | Exclusive | Standalone | Exclusive |
| Adversarial Refinement Agent | 3 | 3 | 25 | 12 | 31 | 0 |
| Hierarchical Strategic Planning Agent | 5 | 5 | 5 | 3 | 38 | 2 |
| Spectrum Search Agent | **35** | **17** | 18 | **14** | 50 | 8 |
| Direct Chain Agent | 28 | 10 | **27** | 8 | **89** | **30** |
| **Oracle Upper Bound** | **53** | – | **56** | – | **102** | – |
| **PoTRE Final Synthesis** | **46** | – | **31** | – | **94** | – |

improved from 94 to 104 successfully solved tasks, elevating its overall accuracy from 78.33% to a state-of-the-art **86.66**% on public evaluation set. Consistent improvements were observed across the broader model family: Gemini-3-Pro-Preview increased its yield from 40 to 45 solved tasks (33.33% $\to$ 37.50%), while the highly efficient Gemini-3-Flash-Preview advanced from 46 to 54 solved tasks (38.33% $\to$ 45.00%).

## 5.3 Summary of Ablation Study

Across three fundamentally diverse benchmarks—Humanity's Last Exam (advanced academic reasoning), PRBench Finance (domain-specific calculation), and ARC-AGI 2 (neuro-symbolic spatial logic)—our component analysis reveals several consistent insights regarding multi-agent architectures:

**1. Universal Superiority of Final Synthesis:** Regardless of the underlying model tier or the specific domain, the PoTRE Final Synthesis strictly and consistently outperforms the best individual agent. By adaptively shifting the synthesis strategy—final candidate selection for advanced academic reasoning (HLE), generative "Super-Answer" fusion for finance, and neuro-symbolic train-pair verification for spatial logic—the synthesis agent substantially mitigates the verification bottleneck and better leverages the diverse strengths of the ensemble. However, we note that a consistent gap remains between the Final Synthesis performance and the theoretical Oracle upper bound. This highlights that while our ensemble approach is highly effective, achieving perfect candidate selection remains an open challenge for future research.

**2. Task and Model-Dependent Topologies:** Our ablations demonstrate that there is no single "silver bullet" reasoning structure. Optimal single-agent performance is highly sensitive to both the task and the model's innate priors. For instance, Gemini-3-Flash-Preview relies heavily on the Adversarial Refinement Agent on PRBench Finance whereas it relies on divergent exploration of the Spectrum Search Agent to act as a crucial "specialist" on ARC-AGI 2 (yielding 17 exclusive solves). This extreme variance underscores the necessity of a multi-agent framework; a diverse ensemble ensures that at least one agent's topology aligns with the specific reasoning demands of the problem.

**3. Test-Time Compute and the "Scaffolding Lift" Phenomenon:** Perhaps the most practically significant finding is the consistent emergence of Scaffolding Lift. On both general and specialized tasks, deploying the smaller, more efficient Gemini 3 Flash Preview model within the PoTRE framework allows it to routinely surpass the single-pass inference performance of the significantly larger Gemini 3 Pro Preview baseline. Consistent with recent findings on test-time compute scaling (Snell et al., 2024), this demonstrates that allocating additional compute through robust, multi-agent structural scaffolding can effectively compensate for raw parameter scale, offering an alternative pathway to state-of-the-art reasoning capabilities.

Table 5: **Impact of Answer Divergence on Candidate Generation.** The rate at which at least one agent generates the correct answer (despite final synthesis failure) increases significantly in high-divergence scenarios across all models.

| Model | Divergence | Total Tasks | Oracle Correct | Oracle Acc. | Already Correct | Only Cand. Correct | % Only Cand. Correct |
|---|---|---|---|---|---|---|---|
| Gemini 3.1 Pro Preview | Low (1–2 unique) | 1705 | 1048 | 61.47% | 989 | 59 | 3.46% |
| | High (3–4 unique) | 795 | 351 | 44.15% | 259 | 92 | **11.57**% |
| Gemini 3 Flash Preview | Low (1–2 unique) | 1596 | 863 | 54.07% | 780 | 83 | 5.20% |
| | High (3–4 unique) | 904 | 336 | 37.17% | 215 | 121 | **13.38**% |
| Claude 4.5 Sonnet | Low (1–2 unique) | 987 | 236 | 23.91% | 185 | 51 | 5.17% |
| | High (3–4 unique) | 1513 | 356 | 23.53% | 195 | 161 | **10.64**% |
| DeepSeek V3.2 | Low (1–2 unique) | 580 | 183 | 31.55% | 145 | 38 | 6.55% |
| | High (3–4 unique) | 1578 | 513 | 32.51% | 286 | 227 | **14.38**% |

## 5.4 Evaluating Heterogeneity

A central premise of the PoTRE is that an engineered heterogeneous pipeline—the generation of structurally diverse reasoning paths—provides a measurable advantage over homogeneous sampling. To rigorously validate this, we conducted an empirical analysis connecting inter-agent answer divergence directly to oracle coverage and synthesis recovery across four frontier models on the HLE benchmark.

### 5.4.1 Impact of Divergence on Oracle Coverage

We first evaluated whether high divergence among agents is indicative of productive exploration rather than mere noise. For each task, we clustered the candidates into semantically unique answers to measure "Answer Divergence." Tasks were partitioned into two categories: Low Divergence (1–2 unique answers) and High Divergence (3–4 unique answers).

We define *Oracle Accuracy* as the presence of the correct answer anywhere within the candidate pool, regardless of whether the final synthesis layer selected it. Specifically, we tracked the rate of tasks where the synthesis prediction failed, but the correct answer was successfully generated by at least one sub-agent. As shown in Table 5, across all evaluated models, the rate of unselected correct candidates is consistently two to three times higher in the High Divergence category. This demonstrates that on highly complex tasks where agents disagree, the heterogeneous reasoning topologies successfully inject the correct answer and its underlying rationale into the candidate pool at a significantly accelerated rate.

### 5.4.2 Minority Recovery and Defeating Majority Vote

While heterogeneous generation successfully expands the oracle pool, the final performance relies entirely on the synthesis layer's ability to extract the correct answer from conflicting signals. To isolate the synthesis layer's efficacy, we analyzed its performance strictly on "Minority Tasks"—defined as tasks where at least one agent generated the correct answer, but this correct logic was numerically outvoted by a cohesive plurality or majority of incorrect candidates (i.e., a 1-vs-3 or 1-vs-2-vs-1 split, explicitly excluding 1-1-1-1 pure ties).

By definition, a standard statistical Majority Vote would achieve exactly 0% accuracy on this subset. As detailed in Table 6, the PoTRE synthesis layer successfully recovered the correct minority answer in 22.01% to 41.43% of these cases.

**Robustness to Synthesis Prompt Formulation**   To verify that this minority recovery capability is a fundamental structural property of the synthesis mechanism rather than an artifact of a specific prompt design, we conducted a prompt sensitivity ablation. Focusing strictly on the 170 minority tasks identified for Gemini 3 Flash Preview, we evaluated the synthesis agent's performance using three semantically distinct instruction paradigms, providing the candidate answer and their rationales in the context for all three:

Table 6: **Minority Recovery Rate.** Evaluated strictly on tasks where the correct answer was numerically outvoted by incorrect candidates (where a standard Majority Vote yields 0%).

| Model | Minority Tasks | Recovered by Synthesis | Recovery Rate | Net Accuracy Gain |
|---|---|---|---|---|
| Gemini 3.1 Pro Preview | 140 | 58 | 41.43% | +2.32% |
| Gemini 3 Flash Preview | 170 | 56 | 32.94% | +2.24% |
| Claude 4.5 Sonnet | 206 | 61 | 29.61% | +2.44% |
| DeepSeek V3.2 | 209 | 46 | 22.01% | +2.13% |

Table 7: **Prompt Sensitivity in Minority Recovery.** Evaluated on the 170 minority tasks for Gemini 3 Flash Preview. The recovery phenomenon remains highly robust across entirely different instruction paradigms.

| Prompt Paradigm | Total Tasks | Correct | Accuracy |
|---|---|---|---|
| Direct Selection (Zero-Shot) | 170 | 49 | 28.82% |
| Reasoning (Deliberative) | 170 | 57 | 33.53% |
| Neutral (Unconstrained) | 170 | 43 | 25.29% |
| **Mean ± Standard Deviation** | | | **29.21% ± 4.13%** |

- **Direct Selection (Zero-Shot):** Prompted the model to bypass intermediate reasoning and output only the final answer, forcing it to rely on its immediate heuristic priors (Prompt: *"Trust your instinct and pick one answer. Provide ONLY the answer text, not the candidate number or reference."*).

- **Reasoning (Deliberative Analysis):** Forced the model to actively analyze the candidate rationales step-by-step before concluding (Prompt: *"Analyze the candidate answers carefully, reason step-by-step, and pick the correct answer. Provide your reasoning first, and then the final answer prefixed with 'Final Answer:'."*).

- **Neutral (Unconstrained Synthesis):** A baseline instruction asking the model to synthesize the candidates and pick the best one without explicitly forbidding or requiring chain-of-thought (Prompt: *"Synthesize the candidate answers and pick the best one. Provide ONLY the answer text."*).

As detailed in Table 7, the minority recovery phenomenon exhibits strong structural stability, achieving a mean recovery rate of 29.21% with a standard deviation of just 4.13% across entirely different instruction paradigms. This tight distribution empirically demonstrates that the architecture's ability to defeat flawed majority consensus is an intrinsic property of the heterogeneous candidate pool, rather than an artifact of specific prompt formulations. Because the synthesis agent consistently rescues the correct minority answer whether instructed to respond via direct selection or deliberative analysis, the data validates our core architectural design: providing the comparative reasoning traces allows the synthesis layer to actively evaluate the underlying logical rationales rather than merely counting exact textual matches.

### 5.4.3 Evaluating the Mechanics of the Heterogeneous Pipeline via Divergence Bucketing

To further isolate the empirical impact of candidate diversity, we segmented the HLE dataset into four distinct buckets based on the level of heterogeneity—defined by the number of semantically unique answers generated across the four sub-agents (ranging from one unique answer to four completely disjoint answers). For each bucket, we evaluated the standard Chain-of-Thought (CoT) baseline, the Oracle accuracy of the diverse candidate pool, and the final Synthesis accuracy.

As detailed in Table 8, the data reveals a clear relationship between task complexity, candidate divergence, and synthesis efficacy:

Table 8: **Answer Divergence.** Performance of baseline Chain-of-Thought (CoT), Oracle pool, and the Synthesis layer on Gemini 3 Flash Preview, grouped by the number of unique answers generated by the four agents (from complete agreement to very high divergence). Tasks with fewer than 4 successful agent attempts were excluded (N=2464 out of 2500 total).

| Bucket (Unique Answers) | Count | CoT Acc. | Oracle Acc. | Synthesis Acc. |
|---|---|---|---|---|
| **1** (All agreed) | 1217 | 51.7% | 52.6% | 52.4% |
| **2** (2 unique) | 718 | 26.6% | 52.6% | 35.9% |
| **3** (3 unique) | 321 | 11.2% | 38.6% | 23.4% |
| **4** (All disagreed) | 208 | 5.8% | 20.7% | 5.8% |

- **The Collapse of Standard CoT:** In low-divergence scenarios (Bucket 1), where all four agents agree, the standard CoT baseline performs strongly (51.7%). However, as task complexity increases and triggers moderate-to-high divergence (Buckets 2 and 3), single-stream CoT reasoning experiences a catastrophic collapse, dropping to 26.6% and ultimately 11.2% accuracy.

- **Benefits are Maximized under Moderate Heterogeneity:** In these same highly complex tasks (Buckets 2 and 3), the heterogeneous candidate pool successfully captures the correct answer at an elevated rate (38.6% Oracle in Bucket 3). More importantly, the task-adaptive synthesis layer effectively leverages this diversity, achieving 35.9% and 23.4% accuracy across these respective buckets. This more than doubles the baseline CoT performance in Bucket 3, empirically demonstrating that structurally diverse reasoning topologies act as a critical safety net against homogeneous mode collapse.

**The Operational Boundary of Synthesis**  A critical architectural insight emerges when analyzing Bucket 4, where all four agents generate completely disjoint answers (a 1-1-1-1 split). While the Oracle pool still contains the correct answer 20.7% of the time, the synthesis layer's accuracy drops to 5.8%, effectively matching the CoT baseline.

When contrasted with our Minority Recovery analysis—where the synthesis layer successfully recovers correct minority answers against a flawed consensus—Bucket 4 clearly delineates the framework's operational boundary. The synthesis layer thrives on comparative logic, weighing a correct minority argument against a flawed majority. However, when consensus breaks down completely (Bucket 4), the synthesis layer loses its comparative anchors. Consequently, its selection accuracy regresses toward random chance within the oracle pool ($20.7\% \div 4 \approx 5.1\%$). This boundary condition highlights a fundamental limitation: while the synthesis layer effectively resolves conflicting logic, it struggles to isolate the correct path from pure noise when zero consensus exists.

### 5.4.4  Summary of Findings

These analyses empirically confirm that the observed performance gains are driven by our engineered heterogeneous pipeline rather than mere stochastic noise. The diverse candidate pool successfully captures correct logical trajectories on complex tasks that resist single-stream prompting (Table 5 and Table 8), while the synthesis layer actively evaluates underlying reasoning traces to rescue correct solutions from flawed majority consensus (Table 6). However, as demonstrated in Bucket 4, this synthesis mechanism relies on comparative logic and regresses to random chance when consensus breaks down entirely. Altogether, this synergistic mechanism yields a consistent absolute accuracy gain of approximately 2.3% across four distinct frontier model families, validating the architectural robustness of the framework while honestly delineating its operational boundaries.

### 5.5 Rationale for Agent Selection and Complementary Strengths

Our selection of these four distinct agents—the Hierarchical Strategic Planning Agent, Standard Chain Agent, Spectrum Search Agent, and Adversarial Refinement Agent—was motivated by a principled goal: to encompass the diverse problem-solving paradigms required for advanced reasoning tasks. Specifically, the architecture of PoTRE is grounded in a systematic framework designed to mitigate the primary reasoning bottlenecks inherent in LLMs. Rather than selecting agent personas at random or merely scaling the ensemble size, we map each agent's unique topology directly to a well-documented failure mode in LLM reasoning:

- Decomposition Limitation: LLMs frequently struggle to break down long-horizon, complex problems into manageable sub-tasks. We address this with the Hierarchical Strategic Planning Agent, which enforces structured, role-based sub-goal generation and execution to prevent the model from becoming overwhelmed by problem scope.

- Depth and Memory Limitation: Complex logic often exceeds a model's parametric memory retrieval capacity when not sequentially grounded. We counter this with the Standard Chain Agent, forcing rigorous, step-by-step deductive depth and ensuring the model remains anchored to first principles.

- Breadth and Tunnel-Vision Limitation: Standard greedy decoding and homogeneous sampling frequently trap models in local optima, leading to correlated errors. The Spectrum Search Agent mitigates this by enforcing divergent, breadth-first parallel hypothesis exploration, ensuring the model does not prematurely commit to a flawed premise.

- Hallucination and Verification Limitation: Models notoriously struggle to critique their own plausible but flawed logic in a single pass. The Adversarial Refinement Agent introduces an independent dialectic loop to actively identify, challenge, and correct internal verification failures.

By formalizing our agent selection around these specific reasoning vulnerabilities, we ensure that the resulting ensemble achieves partial error orthogonality. This complementary structure helps in covering the structural limitations of one agent by the topological strengths of another.

To determine whether all four architectures are necessary or if a smaller subset drives the majority of performance gains, we conducted a complementary analysis focusing on task exclusivity. While some redundancy inevitably exists on lower-complexity tasks, our analysis of the HLE dataset reveals that at the frontier of difficulty, each agent possesses unique, non-overlapping capabilities. Specifically, we identified multiple instances where a highly complex task was solved exclusively by one agent while the other three failed. Four illustrative examples highlight these complementary strengths:

- **Hierarchical Strategic Planning Agent – Avoiding Distractors and Interpreting Scope:** In a math task requiring the calculation of the number of non-negative integer solutions to a Diophantine equation (specifically $x_1^2 + x_2^2 + x_3^2 + x_4^2 + x_5^2 = 2024$), this agent correctly identified the answer as $29,010$ by recognizing it as the number of partitions ignoring order. The other three methods failed or yielded errors. This suggests that the Hierarchical Strategic Planning Agent's high-level oversight and role-based decomposition are particularly effective at interpreting the problem's intended scope correctly and avoiding common misinterpretations that mislead standard step-by-step reasoning.

- **Standard Chain Agent – Complex Multi-Step Reasoning:** In a chess puzzle requiring the determination of a mate in 2 sequence for black without moving the queens, the Standard Chain Agent was the only method to arrive at the correct sequence of moves (*Rxf3, Rf1#*), while the others failed. This illustrates that for tasks requiring strict, sequential deduction and verification of all opponent replies, explicit step-by-step reasoning remains essential.

- **Spectrum Search Agent – Numerical Precision in Complex Algorithms:** In a complex knot theory problem calculating the difference between braid indices and Seifert circle bounds using HOMFLY polynomials, the Spectrum Search Agent (which leverages parallel generation paths) found

the exact correct integer value (3). The other three methods failed, demonstrating that diverse parallel paths can successfully overcome cascading errors in sequential reasoning during complex, multi-step algebraic manipulations.

- **Adversarial Refinement Agent − Multi-Constraint Resolution and Game Theory:** In a game theory problem determining the product of target sums for which Player B can win under optimal play, the Adversarial Refinement Agent exclusively identified the correct product (7744). This suggests that deliberative, adversarial refinement processes are uniquely suited for weighing complex, competing strategies and constraints in game theory.

Ultimately, these examples are the qualitative manifestations of the Inter-Agent Marginal Complementarity defined mathematically in Section 5.8. As demonstrated quantitatively in our Pareto frontier analysis (Appendix G.2), removing any single agent results in a measurable loss of coverage for specific problem types, which these examples illustrate.

### 5.6 Extension: Open-Book Generalization with Search

To evaluate the universality of the PoTRE framework, we extend the architecture to an "Open-Book" setting by equipping the four reasoning agents with access to Google Search. This allows for a direct comparison against frameworks like ReThinker Tang et al. (2026) and tool-heavy "Deep Research" agents like Yunque DeepResearch Cai et al. (2026) and Tongyi DeepResearch Team et al. (2025).

#### 5.6.1 Comparison to State-of-the-Art

Table 9 presents a comprehensive comparison of PoTRE against state-of-the-art baselines on the HLE text-only benchmark ($N = 2,158$). Three key findings emerge:

- **Scaffolding Lift in Search:** In the open-book setting, the lightweight Gemini-3-Flash-Preview model (55.28%) outperforms the larger Gemini-3-Pro-Preview model (53.85%). This suggests that for tool-use tasks, the *latency-throughput advantage* of Flash (enabling wider parallel search in the Spectrum search agent) outweighs the *parameter-depth advantage* of Pro.

- **Outperforming SOTA:** PoTRE (Flash) achieves 55.28%, explicitly outperforming ReThinker framework (52.2%) and the Yunque DeepResearch baseline (51.7%). This demonstrates a significant "Scaffolding Lift," demonstrating that our engineered heterogeneous pipeline enables lightweight models to surpass frontier proprietary systems.

- **Superiority over Open-Source Agents:** PoTRE significantly outperforms the Tongyi DeepResearch agent (32.9%), establishing a new state-of-the-art for open-architecture research systems on this benchmark.

- **Agent Comparison (Depth vs. Breadth):** We contrast PoTRE's *parallel breadth* (topological diversity) against the *sequential depth* optimized by current SOTA systems. While Yunque DeepResearch relies on long-context centralization Cai et al. (2026) and ReThinker relies on iterative homogeneous refinement Tang et al. (2026), both require frontier-model backbones (Gemini-3-Pro-Preview) to function. Our results (55.28% vs. 51.7%/52.2%) demonstrate that parallelizing reasoning across diverse agents is a highly effective method for decoupling performance from parameter scale, proving to be a superior architectural strategy for maximizing the reasoning potential of lightweight models.

**Full Set Verification and SOTA Comparison.** To ensure these findings are robust, we also evaluated PoTRE on the full HLE validation set ($N = 2,500$), which includes multimodal and complex queries. The Scaffolding Lift remains consistent: Gemini-3-Flash-Preview achieves 53.48% (1337/2500), maintaining its lead over the standalone Gemini-3-Pro-Preview baseline (51.88%). Furthermore, Gemini-3.1-Pro-Preview achieves **58.40%**(1460/2500) maintaining its lead amongst all other models.

Table 9: **Open-Book Performance Comparison.** PoTRE establishes a new state-of-the-art on the HLE text-only subset with Gemini-3.1-Pro-Preview (60.24%). Additionally, PoTRE (Flash) demonstrates significant *Scaffolding Lift*, outperforming the heavier Yunque and ReThinker baselines despite utilizing a highly cost-efficient backbone.

| System | Base Model | Mode | Accuracy |
|---|---|---|---|
| *Baselines* | | | |
| Tongyi DeepResearch Team et al. (2025) | Qwen-30B | Open-Book | 32.90% |
| Yunque DeepResearch Cai et al. (2026) | Gemini-3-Pro-Preview | Open-Book | 51.70% |
| ReThinker Tang et al. (2026) | Gemini-3-Pro-Preview | Open-Book | 52.20% |
| *PoTRE (Ours)* | | | |
| PoTRE (Pro) | Gemini-3-Pro-Preview | Open-Book | 53.85% |
| PoTRE (Flash) | Gemini-3-Flash-Preview | Open-Book | **55.28%** |
| PoTRE (3.1 Pro) | **Gemini-3.1-Pro-Preview** | **Open-Book** | **60.24%** |

### 5.7 Impact of Search Augmentation on Sub-Agents

Figure 3 illustrates the profound impact of search augmentation across our four reasoning agents on the HLE dataset. While external retrieval universally elevates performance, the magnitude of these gains reveals critical insights into the distinct reasoning bottlenecks of each agent structure:

- **Effect on Direct Chain Agent:** The most dramatic improvement is observed in the Direct Chain Agent. Operating under closed-book constraints, this standard reasoning topology performed the poorest (40.72%). However, upon introducing search augmentation, it experienced a massive +14.68% surge, bringing it perfectly in line with the other agents at 55.40%. This indicates that traditional Chain-of-Thought reasoning is disproportionately bottle-necked by parametric memory limits. When forced to rely solely on internal weights for highly specialized HLE queries, the agent confidently hallucinates. Search acts as a massive equalizer, substituting flawed internal retrieval with grounded external facts, allowing the model's pure logical sequencing to shine.

- **Amplifying Exploration:** The Spectrum Search Agent achieved the highest absolute search-augmented accuracy (57.20%), building upon an already strong closed-book foundation (48.40%). Because this agent's core topology is designed for breadth-first exploration and trying different things across diverse hypotheses, injecting real-time search data increases the quality of its candidate generation. Search gives the spectrum search agent a substantially richer, more accurate foundation of raw material to explore.

- **Grounding Complex Structural Topologies:** Both the Adversarial Refinement Agent (47.00% → 55.16%) and the Hierarchical Strategic Planning Agent (45.36% → 55.32%) demonstrate robust, consistent gains of roughly 8% to 10%. For these highly structured topologies, search acts as an factual grounding.
    - *In the Adversarial setting:* External search provides concrete evidence to resolve internal debates, preventing the agents from arguing into theoretical dead ends.
    - *In the Hierarchical setting:* Search supplies the precise, step-by-step data required to successfully execute sub-goals without suffering from cascading knowledge errors.

Ultimately, this analysis demonstrates that search augmentation does more than just "add facts"—it fundamentally repairs the specific structural weaknesses inherent to different reasoning agents, elevating the

baseline quality of the candidates passed to the final Synthesis agent. Overall, the full PoTRE system improves from 49.92% (Closed) to 58.40% (Open) but the theoretical upper bound (Oracle) with Search is **64.6%** (1,615/2,500), indicating significant headroom for improved synthesis strategies in future work.

Figure 3: **Impact of Search Augmentation on HLE.** Performance comparison of the four sub-agents under Closed-Book and Search-Augmented constraints. External retrieval universally elevates accuracy across all reasoning topologies. Notably, the Direct Chain Agent experiences the most dramatic marginal gain (+14.60%), effectively utilizing search to overcome its structural limitations and matching the performance of more complex architectural agents.

### 5.7.1 Robustness and Boundary Conditions

We evaluate PoTRE on two distinct formats to verify the universality of these gains.

**1. Closed-Ended Reasoning (MCQ Comparison).** In Table 10, we specifically evaluated PoTRE on the HLE Multiple-Choice subset ($N = 591$) to enable a direct comparison with MonoScale Shao et al. (2026), a recent study that benchmarks agentic scaling on this exact partition. This allows us to isolate the impact of our reasoning topology against their strategy of simply increasing agent count.

**2. Domain-Specific Expert Reasoning (PRBench Finance).** To test the boundaries of search augmentation, we evaluated PoTRE on the "Hard Subset" of PRBench Finance ($N = 300$) [Table 11].

The minimal gains on PRBench provide a crucial insight: access to the internet does not automatically solve expert-level problems. While search helps significantly on open-domain knowledge tasks (HLE +8.00%), it offers little advantage on tasks requiring synthesized professional judgment. Notably, the Scaffolding Lift persists: the lightweight Flash model (0.3509) continues to outperform the Pro model (0.3351), confirming that our agentic topology is effective even in domain-specific settings.

Table 10: **MCQ Performance.** By testing on the MonoScale subset, we confirm that PoTRE (Open-Book) significantly outperforms the agentic scaling baseline (65.48% vs. 19.90%). This demonstrates that *reasoning architecture* yields far higher gains than merely scaling the number of agents ($N = 10$).

| Method | Model | MCQ Accuracy |
|---|---|---|
| MonoScale ($N = 10$ Agents) | Qwen-3 | 19.90% |
| PoTRE (Ours) | Gemini-3-Flash-Preview | **60.74**% |
| PoTRE (Ours) | Gemini-3-Pro-Preview | 60.07% |
| PoTRE (Ours) | Gemini-3.1-Pro-Preview | **65.48%** |

Table 11: **PRBench Control Experiment.** Search yields minimal gains ($< 0.01$) on this expert reasoning task. This confirms that for high-level professional domains, the bottleneck is *judgment* (reasoning), not just *information access* (retrieval).

| Model | Mode | Avg Clip Score |
|---|---|---|
| Gemini-3-Pro-Preview | Open-Book | 0.3351 (+0.0032) |
| Gemini-3-Flash-Preview | Open-Book | **0.3509 (+0.0023)** |

## 5.8 Orthogonality Analysis

To understand the efficacy of our poly-topological ensemble, we analyzed the inter-agent orthogonality and marginal utility of our four primary reasoning agents. The analysis revealed divergence in the failure modes of the individual agents, demonstrating that their unique topological structures actively navigate the language model toward distinct, valid regions of the solution space.

Rather than relying on mere prompt variations or homogeneous sampling, PoTRE deploys a set of structurally diverse reasoning agents: the Adversarial Refinement Agent performs debate, the Hierarchical Strategic Planning Agent utilizes structured diagnostic planning, and the Spectrum Search Agent synthesizes parallel, independent perspectives. As illustrated in Table 12, the Spectrum Search Agent establishes a strong individual accuracy of 57.20%. However, the sequential aggregation of the remaining agents yields a monotonically increasing cumulative accuracy. Notably, the *Direct Chain Agent* successfully resolves an additional 100 unique tasks that the primary spectrum search agent failed to solve. In a sub-agent level analysis evaluating the union of successful predictions—where a task is considered solved if *any* of the four sub-agents produces the correct answer—this aggregate coverage reaches 64.60%, representing a substantial absolute gain of 7.40% over the best individual constituent.

To formalize this reasoning orthogonality, we computed an Inter-Agent Marginal Complementarity matrix (Table 13), defined mathematically as the conditional probability $P$(Agent B solves | Agent A fails). These off-diagonal probabilities remain robust across all pairings. While the reasoning agents share a substantial baseline of knowledge—resulting in redundancy on easier tasks—their failure boundaries remain only loosely correlated. For example, when the *Spectrum Search Agent* fails, the *Hierarchical Strategic Planning Agent* independently complementing 9.9% of those failures, while the *Direct Chain Agent* simultaneously complements 9.4%. Furthermore, our sequential ablation demonstrates that every individual agent contributed a meaningful margin on top of the growing ensemble (*Direct Chain*: +100, *Hierarchical*: +58, *Adversarial*: +27). This marginal complementarity capability empirically confirms that the reasoning manifolds of each agents are fundamentally distinct, definitively validating the necessity of a functionally diverse, "width-over-depth" architecture to prevent homogeneous mode collapse.

**ARC AGI 2 Benchmark Validations**   To evaluate cross-domain robustness, we identically computed the marginal impact (Table: 14) and marginal complementarity matrix (Table 15) on the ARC-AGI 2 dataset

Table 12: **Cumulative Marginal Impact on HLE.** Sequential addition of each reasoning topology demonstrates the value of topological diversity. Each distinct agent successfully solves a unique subset of problems that prior agents failed on, culminating in an upper-bound ensemble accuracy of 64.60% ($N = 2500$).

| Sub-Agents | New Tasks Added | Cumulative Total | Ensemble Accuracy |
|---|---|---|---|
| Spectrum Search Agent | +1430 | 1430 | 57.20% |
| Direct Chain Agent | +100 | 1530 | 61.20% |
| Hierarchical Strategic Planning Agent | +58 | 1588 | 63.52% |
| Adversarial Refinement Agent | +27 | 1615 | 64.60% |

Table 13: **Inter-Agent Marginal Complementarity on the HLE.** This table displays the marginal complementarity between parallel reasoning agents. Each cell indicates the percentage of questions failed by the row's agent but successfully solved by the column's complementing agent. The complementarity rates across all combinations demonstrate they are loosely correlated of our chosen reasoning topologies.

| Failed Agent \ Complementing Agent | Hierarchical Strategic Planning Agent | Adversarial Refinement Agent | Spectrum Search Agent | Direct Chain Agent |
|---|---|---|---|---|
| Hierarchical Strategic Planning Agent | – | 11.7% | 13.7% | 12.8% |
| Adversarial Refinement Agent | 12.1% | – | 13.0% | 12.6% |
| Spectrum Search Agent | 9.9% | 8.8% | – | 9.4% |
| Direct Chain Agent | 12.7% | 12.1% | 13.0% | – |

($N = 120$ tasks). The architectural heterogeneity of the four agents is not restricted to linguistic tasks; the architecture demonstrates equivalent efficacy when applied to the spatial grid reasoning of ARC-AGI. generating incredibly high Marginal Complementarity rates (e.g. the *Spectrum search* agent successfully complementing nearly a third, 29.9%, of tasks failed by *Adversarial refinement agent*).

### 5.8.1 Calibration against a Null Model

To formally quantify the degree of independence between our agents, we calibrate our findings against a null model of complete statistical independence. We define the *Relative Complementarity Ratio* as:

$$\text{Complementarity Ratio} = \frac{P(\text{Agent B solves} \mid \text{Agent A fails})}{P(\text{Agent B solves})}$$

Under a null model of complete independence, this ratio equals 1.0. A ratio $< 1.0$ indicates correlation in failure modes (sharing blind spots), while a ratio $> 1.0$ indicates positive complementarity. As detailed in Table 16, we analyzed the average ratios across model families for both HLE and ARC-AGI-2.

The data reveals a stark contrast dependent on the problem domain. On the exceptionally difficult HLE benchmark, the ratios fall below 1.0 (ranging from 0.25 to 0.57), indicating that agents are partially correlated. Because HLE presses the absolute limits of the base models' factual knowledge, agents inevitably share fundamental parametric blind spots. However, on the ARC-AGI dataset—which evaluates pure spatial and visual fluid intelligence without relying on retrieved facts—the ratio successfully reaches or exceeds 1.0 for the Gemini Flash and Pro models. This demonstrates that our multi-agent architecture is fully capable of achieving positive complementarity. Consequently, this formal calibration justifies framing PoTRE's agents as *loosely correlated* rather than strictly orthogonal: the degree of inter-agent independence achievable is heavily modulated by the intrinsic complexity and factual demands of the target domain.

Table 14: **Cumulative Marginal Impact on ARC-AGI-2 (Gemini-3-Flash-Preview).** Sequential addition of each reasoning topology further demonstrates the value of topological diversity on spatial logic tasks, peaking at an ensemble accuracy of 44.2%.

| Sub-Agents | New Tasks Added | Cumulative Total | Ensemble Accuracy |
|---|---|---|---|
| Spectrum Search Agent | +35 | 35 | 29.2% |
| Chain-of-Thought | +10 | 45 | 37.5% |
| Hierarchical Strategic Planning Agent | +5 | 50 | 41.7% |
| Adversarial Refinement Agent | +3 | 53 | 44.2% |

Table 15: **Inter-Method Marginal Complementarity on ARC-AGI-2.** This table displays the marginal complementarity across all parallel reasoning agents. Each cell indicates the percentage of questions failed by the row's agent but successfully solved by the column's complementing agent. Noticeably, the Spectrum Search Agent serves as a massive rescuer for the other topologies on this spatial benchmark.

| Failed Agent \ Complementing Agent | Hierarchical Strategic Planning Agent | Direct Chain Agent | Spectrum Search Agent | Adversarial Refinement Agent |
|---|---|---|---|---|
| Hierarchical Strategic Planning Agent | – | 24.3% | 30.4% | 2.6% |
| Direct Chain Agent | 5.4% | – | 18.5% | 3.3% |
| Spectrum Search Agent | 5.9% | 11.8% | – | 3.5% |
| Adversarial Refinement Agent | 4.3% | 23.9% | 29.9% | – |

## 5.9 The Verification Bottleneck

Our ablation studies reveal a consistent gap between the "Oracle" potential and final performance across domains. On HLE (Open-Book), the system generated correct candidates for 64.60% of tasks, yet recovered 58.40%. Similarly, on ARC-AGI-2, the Oracle score reached 44.16% (53/120) compared to the actual score of 38.30%. This indicates that while PoTRE excels at *generating* diverse valid reasoning paths, the Synthesizer's ability to *discriminate* truth from plausible distractors remains the primary constraint.

## 6 Computational Cost

A critical finding of this study is the consistent "Scaffolding Lift" observed across all evaluated benchmarks (ARC-AGI-2, HLE, and PRBench). Our results demonstrate that the smaller, more efficient Gemini-3-Flash-Preview, when equipped with the PoTRE agentic framework, consistently outperforms the larger Gemini-3-Pro-Preview operating with single-pass prior work prompts Zhang et al. (2025b) and Zhang et al. (2025a). Consistent with recent literature Snell et al. (2024), this confirms that PoTRE's structural scaffolding provides a highly effective mechanism for structuring test-time compute, allowing a smaller model to systematically surpass the single-inference limits of a larger foundation model, rather than the gains being merely a result of prompting disparities.

These findings challenge the assumption that reasoning performance scales strictly with model size. Instead, they suggest that for complex tasks—ranging from the visual abstractions of ARC-AGI-2 to the semantic logic of HLE—architectural scaffolding is a more decisive performance driver than raw parameter count. By distributing the reasoning process among specialized agentic roles (e.g., adversarial verification, spectrum search, hierarchical planning and direct chain), PoTRE enables smaller models to exceed their parameter-bound capabilities, effectively trading internal scale for external test-time compute.

Rather than relying on massive token inflation, PoTRE strategically optimizes inference-time compute. By distributing this compute across four distinct reasoning agents, the framework enables standard LLMs to

Table 16: **Relative Complementarity Ratio.** Calibration of agent independence against a null model (Ratio = 1.0). Ratios vary significantly by domain, reflecting the relationship between task complexity and inter-agent independence.

| Model Family | Avg. Relative Complementarity (HLE) | Avg. Relative Complementarity (ARC) |
|---|---|---|
| Gemini 3 Flash Preview | 0.27 | **1.10** |
| Gemini 3 Pro Preview | 0.26 | **1.10** |
| Gemini 3.1 Pro Preview | 0.25 | 0.80 |
| DeepSeek V3.2 | 0.57 | N/A |
| Claude 4.5 Sonnet | 0.54 | 0.65 |

achieve state-of-the-art performance using similar or fewer tokens than heavily scaled homogeneous ensembles. While this agentic approach incurs a 15× token overhead compared to baseline CoT, the impact on user experience is minimized through asynchronous execution. Because the agents run in parallel, the total latency is bounded only by the slowest single component, delivering high-fidelity reasoning without prohibitive delays.

To rigorously evaluate the computational efficiency of our framework against existing literature, we calculated the inference costs based on the official Google Gemini API pricing tiers (Standard API, $\leq$ 200k context window). Because modern reasoning models utilize hidden "thinking" budgets, output costs were calculated by subtracting the logged prompt tokens from the total tokens to capture the true billed output volume (which encompasses both visible candidate tokens and hidden reasoning tokens).

We logged exact token consumption for our evaluations across the 2,158-question text-only subset of HLE.

- **Gemini 3.1 Pro Preview Variant:** Priced[3] at \$2.00 per 1M input tokens and \$12.00 per 1M output tokens. The evaluation consumed 28.6M input tokens and 463.2M output tokens (comprising 18.8M candidate tokens and 444.4M reasoning tokens). This resulted in an input cost of \$57.13 and an output cost of \$5,558.38, for a total empirical cost of \$5,615.51.

- **Gemini 3 Flash Preview Variant:** Priced at \$0.50 per 1M input tokens and \$3.00 per 1M output tokens. The evaluation consumed 28.3M input tokens and 680.5M output tokens (19.4M candidate tokens and 661.1M reasoning tokens). Despite utilizing a larger reasoning budget to achieve its performance, the cost-efficient architecture resulted in an input cost of \$14.13 and an output cost of \$2,041.55, for a total empirical cost of \$2,055.68.

**ReThinker Baseline:** Because the exact token consumption of the ReThinker framework is unpublished, we established an empirically-aligned cost baseline grounded in their reported architectural constraints. ReThinker relies on the Gemini 3 Pro model (which shares the exact \$2.00/\$12.00 pricing tier as Gemini 3.1 Pro Preview) and utilizes a 3-stage agentic loop (Solver, Critic, Selector). Crucially, their methodology explicitly defines a 128K maximum context window limit for their generation process.

To ensure fair comparison, we anchored the baseline to our empirical observations for the Gemini 3.1 Pro Preview model. We matched their input token volume exactly to our empirical run (28.6M total input tokens), as both frameworks process the identical dataset text. Similarly, we established a baseline of 18.8M visible candidate tokens. To account for the generative overhead of their multi-round iterative synthesis, we allocated a highly conservative budget of ~197M thinking tokens, yielding a total estimated output volume of 215.8M tokens (100,000 output tokens per question). This models a scenario where their agentic loop utilizes ~113,236 total tokens per question, keeping it comfortably within their stated 128K context window limit. Applying the standard pricing model, this utilization incurs an estimated input cost of \$57.13 and an output cost of \$2,589.60, yielding an estimated total evaluation cost of \$2,646.73. As detailed in Table

---

[3]Official Pricing: `https://ai.google.dev/gemini-api/docs/pricing`

Table 17: **Accuracy, Token Consumption, and Estimated Cost Comparison.** Evaluated on the HLE Text-Only Subset ($N = 2,158$). The Flash variant of our framework demonstrates exceptional cost-efficiency, outperforming the baseline in accuracy while costing significantly less to run.

| Metric | Baseline (Estimated) | Our Framework (Empirical) | |
| | ReThinker[†] (Gemini-3-Pro-Preview) | Flash Variant (Gemini-3-Flash-Preview) | Pro Variant (Gemini-3.1-Pro-Preview) |
| --- | --- | --- | --- |
| **Accuracy** | 52.20% | 55.28% | **60.24%** |
| **Input Tokens** | ∼28.6M | 28.3M | 28.6M |
| **Output Tokens**[*] | ∼215.8M | 680.5M | 463.2M |
| **Total Tokens** | ∼244.4M | 708.8M | 491.8M |
| **Estimated Cost** | ∼$2,646.73 | **$2,055.68** | $5,615.51 |

[*] Output encompasses both visible candidate tokens and model-generated reasoning/thinking tokens.

[†] ReThinker tokens are estimated by matching our empirical input volume and utilizing their stated 128K context limit.

17, our Gemini 3 Flash Preview variant surpasses the baseline's reported accuracy (55.28% vs. 52.20%). Crucially, despite generating nearly triple the volume of raw reasoning tokens to achieve this performance (680.5M vs. an estimated 215.8M), our framework's total cost remains modest at $2,055.68 vs. an estimated $2,646.73. This demonstrates that leveraging a high-throughput model like Gemini 3 Flash Preview allows for significantly deeper, more exhaustive reasoning trajectories—ultimately outperforming complex multi-agent Pro-level baselines—while maintaining strict cost superiority.

# 7 Conclusion

We presented PoTRE, an empirical framework that systematically leverages distinct reasoning topologies-adversarial, hierarchical, spectrum search and direct chain agents to solve complex general intelligence tasks. Through extensive experimentation across three distinct frontier benchmarks, we demonstrated that architectural topology is as critical as model scale for high-order problem solving.

PoTRE establishes a new state-of-the-art on *Humanity's Last Exam*, achieving **49.92%** in closed-book settings and **58.40%** in open-book mode on full set where it outperforms all the existing models and agentic frameworks. Furthermore, PoTRE achieves 60.24% on text only subset, outperforming frameworks like Yunque DeepResearch (51.7%) and ReThinker (52.2%). We observe a robust "Scaffolding Lift" across all three benchmarks: the lightweight Gemini-3-Flash-Preview backbone, when augmented with PoTRE, consistently outperforms the larger Gemini-3-Pro-Preview model—achieving absolute superiority on ARC-AGI-2 (38.30% vs. 21.66%) and PRBench (0.3486 vs. 0.2603)—and effectively matches or exceeds even high-cost Self-Consistency ensembles ($N = 16$). This demonstrates that structuring test-time compute through our engineered heterogeneous pipeline allows efficient models to consistently outperform heavier frontier baselines and homogeneous sampling strategies while operating under comparable or reduced token budgets.

Two critical objectives are highlighted for future investigation. First, priority is placed on closing the "The Verification Bottleneck" (64.60% Oracle vs. 58.40% Actual), where correct candidates currently generated are frequently overlooked by the Synthesis agent. To resolve this, a transition moves beyond black-box evaluation by incorporating process-level analysis, ensuring that underlying generative logic is verified rather than relying on surface-level plausibility. Second, the implementation of adaptive agent routing is identified as a necessary advancement. By utilizing a lightweight dispatcher to dynamically activate reasoning agents based on problem difficulty, computational efficiency can be optimized while preserving ensemble robustness—essential steps in the shift from simple prompting to structured, self-correcting multi-agent architectures.

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

# A    Appendix A: Architectural Decisions & Prompt Engineering

This appendix details the exact system prompts and interaction protocols used in PoTRE.

## A.1    Note on Spectrum Search Agent Strategy

*Architectural Note:* The prompts for the spectrum search agent differ significantly between domains. This is a deliberate "Task-Adaptive" design choice to address the specific constraints of verification:

- **For ARC-AGI-2 (Exact Logic):** We generate $N^4$ parallel independent candidates and employ an Iterative Refinement Loop (Hypothesis → Critique → Code).
  - *Reasoning:* Since the solution space is deterministic and executable, the agent must verify its logic against training examples before final implementation.
  - *Selection:* We use output majority voting on the executed grids, as consensus is a robust signal for objective correctness.

- **For HLE & PRBench (Open-Ended):** We generate $N$ parallel independent candidates.

---

$^4$For all datasets in our implementation, we set $N = 8$.

- *Reasoning:* Because these tasks lack verifiable input-output examples at inference time (unlike ARC), we cannot programmatically test and refine the answers. Instead, we rely entirely on the breadth and diversity of the generated reasoning paths.
- *Selection:* We use an LLM judge.

The following sections provide the prompts for these components.

## A.2 Humanity's Last Exam

### A.2.1 Adversarial Refinement Agent via Debate (HLE)

This section contains the exact system prompts and interaction templates used in the adversarial debate loop. The temperature settings was left default / not set for this agent.

---

**System Prompt: Proposer Agent**

```
PROPOSER_SYSTEM = """You are an expert Proposer in a refined debate.
Your goal is to answer the user's question as accurately as possible.
You will receive a Question (and optional Image).
You must provide a tentative Answer and a detailed Rationale.
Later, you might receive critique from a Verifier.
You must update your answer if the critique is valid.

Output JSON:
{
"rationale": "...",
"answer": "..."
}
"""
```

---

**System Prompt: Verifier Agent**

```
VERIFIER_SYSTEM = """You are an expert Verifier. You will receive a Question,
an Image (optional), and a Candidate Answer from a Proposer.
Your goal is to find flaws, logical gaps, or factual errors in the
Candidate Answer.
If the answer is correct, you should APPROVED it.
If incorrect, provide specific, constructive critique to help the Proposer fix it.

Output JSON:
{
"critique": "...",
"status": "APPROVED" or "REJECTED"
}
"""
```

---

Round 1: Initial Proposal, The Proposer is fed the question (and optionally the image).

**Input Template: Initial Proposal**

```
Question: {Question Text}
[Image] (Optional)
```

Round N: Verification, The Verifier receives the original context plus the Proposer's latest candidate.

---

**Input Template: Verification**

```
[Image] (Optional)
Question: {Question Text}

Candidate Answer: {Answer}
Candidate Rationale: {Rationale}

Critique this.
```

---

Round N: Refinement (Rebuttal), If the Verifier rejects the answer, the Proposer receives the critique to guide refinement.

---

**Input Template: Refinement**

```
The Verifier provided this critique:
{Critique}

Please refine your answer and rationale.
```

---

### A.2.2 The Hierarchical Strategic Planning Agent (HLE)

This section contains the exact system prompts and interaction templates used in the Hierarchical Strategic planning agent.

- Temperature for Architect = 0.7.

- Temperature for Engineer = 0.7.

- Temperature for Supervisor = 0.

---

**System Prompt: Architect Agent**

```
ARCHITECT_SYSTEM = """You are a Lead Architect for complex reasoning tasks.
Your goal is to break down the user's question into 2-3 logical sub-steps or key aspects that
    need to be analyzed to reach the correct answer.
Do NOT try to answer the question directly effectively. Instead, Plan.

Output JSON:
{
  "plan_steps": ["Step 1...", "Step 2..."],
  "reasoning_strategy": "..."
}
"""
```

---

**System Prompt: Engineer Agent**

```
ENGINEER_SYSTEM = """You are a Senior Engineer.
You will receive a plan from the Architect and the original Question (and optional Image).
Execute the plan. detailed.

Output JSON:
{
```

---

```
  "execution_trace": "...",
  "draft_answer": "..."
}
"""
```

## System Prompt: Supervisor Agent

```
SUPERVISOR_SYSTEM = """You are a Chief Supervisor.
You have the Architect's Plan and the Engineer's Execution/Draft.
Your job is to specific the final answer and rationale, ensuring strict adherence to the
    facts and logic.
Refine the Engineer's draft if needed.

Output JSON:
{
  "rationale": "...",
  "answer": "..."
}
"""
```

## System Prompt: Meta-Reviewer (Diagnosis)

```
You are a Senior Meta-Reviewer for an AI Reasoning System.
The Agent is stuck in a loop trying to solve a complex reasoning task.

HISTORY OF ATTEMPTS:
{History Summary}

YOUR DIAGNOSIS TASKS:
1. **Pattern Recognition:** Is the agent repeating the same failed logic or plan?
2. **Error Analysis:** - Is the Architect proposing plans that are too vague?
   - Is the Engineer failing to execute specific steps?
   - Is the Supervisor missing critical constraints in the final answer?
3. **Strategic Pivot:** Dynamically synthesize a high-level strategic directive to re-orient
    the Architect's approach.

OUTPUT FORMAT (JSON):
{
    "diagnosis": "The agent keeps trying to calculate X without considering Y...",
    "custom_advice": "You are ignoring the constraint about 'without moving queens'. Focus on
        knight moves."
}
```

### Interaction Templates

The following templates define how data flows between the agents.

## 1. Architect Step Input

```
[Image] (Optional)
Question: {Question Text}
```

```
(Note: If Meta-Reviewer intervenes, a "CRITICAL INTERVENTION" block is appended here)
```

**2. Engineer Step Input**

```
[Image] (Optional)
Question: {Question Text}

Architect's Plan:
{Plan JSON}

Execute this plan detailed.
```

**3. Supervisor Step Input**

```
[Image] (Optional)
Question: {Question Text}

Architect Plan: {Plan JSON}
Engineer Draft: {Draft Answer JSON}

Provide the Final Answer.
```

### A.2.3    Spectrum Search Agent (HLE)

This section contains the exact system prompts and interaction templates used in the Spectrum Search Agent. Temperature for generation was set to 0.7 and temperature for Judge was set to 0.

**System Prompt: Candidate Generator**

```
CANDIDATE_SYSTEM = """You are a Candidate Generator.
You will receive a Question (and optional Image).
Propose a detailed, logical answer. Think step by step.
Output JSON: { "rationale": "...", "answer": "..." }
"""
```

**System Prompt: Expert Judge**

```
JUDGE_SYSTEM = """You are an Expert Judge.
You have a Question and multiple Candidate Answers.
Your job is to select the BEST answer based on accuracy, logical soundness, and completeness.
Output JSON: { "best_candidate_index": int, "reason": "..." }
"""
```

**Interaction Templates**

**Candidate Generation Step**    Each worker receives the same input independently.

**Input: Candidate Generation**

```
[Image] (Optional)
Question: {Question Text}
```

**Judging Step**   The Judge receives the original question and the aggregated list of candidates.

**Input: Judging Step**

```
Question: {Question Text}

--- Candidate 0 ---
Rationale: {Rationale Text}
Answer: {Answer Text}

--- Candidate 1 ---
Rationale: {Rationale Text}
Answer: {Answer Text}

...

Select the index (0-based) of the best candidate.
```

### A.2.4   Direct Chain Agent (HLE)

Temperature $= 0.7$

**System Prompt: CoT Instruction**

```
SYSTEM_INSTRUCTION = """"You are an expert in humanities, law, and logical reasoning.
Answer the following question.
You must output your answer in valid JSON format:
{
  "rationale": "Detailed step-by-step reasoning...",
  "answer": "The final answer (e.g. 'A', 'B', '42', 'summary')"
}
"""
```

**Interaction Template**   The agent performs a single-turn generation.

**Input Template**

```
[Image] (Optional)
Question: {Question Text}

Provide your detailed rationale and final answer in JSON.
```

### A.3 Financial Domain Prompts

#### A.3.1 Adversarial Refinement Agent (Finance)

This section contains the exact system prompts and interaction templates used in the financial adversarial debate loop. The temperature settings was left default / not set for this agent.

- **Proposer Agent**

---

**System Prompt: Financial Architect**

```
You are a Senior Financial Architect.
Your goal is to answer the user's finance question as accurately as possible.
You will receive a conversation history or a question.

Provide a comprehensive, accurate, and professional answer.
```

---

- **Verifier Agent**

---

**System Prompt: Financial Auditor**

```
You are a Senior Financial Auditor.
You will receive a Question and a Candidate Answer from a Proposer.

Your goal is to verify the answer for:
1. Accuracy: Are the financial facts and calculations correct?
2. Completeness: Did the answer address the user's question fully?

If the answer is accurate and complete, APPROVED it.
If not, provide specific critique demanding the missing or incorrect elements.

Output JSON:
{
  "critique": "...",
  "status": "APPROVED" or "REJECTED"
}
```

---

- **Dynamic Context & Flow** The context is constructed linearly. The agent sees the following sequence:

  **1. System Instruction** (Defined above for Proposer/Verifier)

  **2. Conversation History (Injected First)** All previous turns are passed to the model's history before the current question.

---

**Context Structure**

```
[User]: {prompt_0}
[Assistant]: {response_0}
...
[User]: {prompt_N}
[Assistant]: {response_N}
```

---

  **3. Current Turn** After the history, the current tasks are appended.

**Input: Proposer View**

```
{current_question}
[Optional Image Attachment]
```

The verifier sees the history, then receives the Current Question bundled with the Candidate Answer in a single turn.

**Input: Verifier View**

```
{current_question}
[Optional Image Attachment]

Candidate Answer: {current_answer}
Candidate Rationale: {current_rationale}

Critique this.
```

If the Verifier rejects the answer, the Proposer receives the critique to guide the refinement.

**Input: Refinement Instruction**

```
The Verifier provided this critique:
{critique}

Please refine your answer and rationale.
```

### A.3.2 Hierarchical Strategic Planning Agent (Finance)

This section contains the prompts for the hierarchical financial reasoning agents.

- **Architect Agent**
  - **Role:** Principal Financial Architect
  - **Goal:** Break down the problem into a logical plan.
  - **Temperature:** 0.7.

**System Prompt: Architect**

```
You are a Principal Financial Architect.
Your goal is to break down the user's finance problem into 2-3 logical sub-steps (e.g
    . Regulatory Analysis, Quantity Calculation, Risk Assessment).
Do NOT answer the question directly. Instead, Plan.

Output JSON:
{
  "plan_steps": ["Step 1...", "Step 2..."],
  "reasoning_strategy": "..."
}
```

*Context: Receives the full conversation history (User/Assistant turns).*

- **Engineer Agent**

- **Role:** Senior Financial Analyst
- **Goal:** Execute the Architect's plan rigorously.
- **Temperature:** 0.7.

**System Prompt: Engineer**

```
You are a Senior Financial Analyst.
You will receive a plan from the Architect and the original Question.
Execute the plan rigorously. Use financial formulas and regulations where applicable.

Output JSON:
{
  "execution_trace": "Detailed work...",
  "draft_answer": "..."
}
```

**Appended Instruction: Engineer Input**

```
Architect's Plan: {plan_json}

Execute this plan to answer the previous question.
```

*Context: Receives the full conversation history.*

- **Supervisor Agent**
  - **Role:** Chief Risk Officer (CRO)
  - **Goal:** Synthesize the final answer from the plan and execution draft.
  - **Temperature:** 0.

**System Prompt: Supervisor**

```
You are a Chief Risk Officer (CRO).
You have the Architect's Plan and the Analyst's Execution/Draft.
Your job is to synthesize the Final Answer.

Ensure the answer is comprehensive, accurate, and professional.
Refine the draft to be technically precise and practically actionable.

Output JSON:
{
  "answer": "..."
}
```

**Appended Instruction: Supervisor Input**

```
Architect Plan: {plan_json}
Engineer Draft: {eng_json}

Review the draft and provide the Final Answer.
```

*Context: Receives the full conversation history.*

**System Prompt: Meta-Reviewer (Diagnosis)**

```
You are a Senior Meta-Reviewer for an AI Reasoning System.
The Agent is stuck in a loop trying to solve a complex reasoning task.

HISTORY OF ATTEMPTS:
{History Summary}

YOUR DIAGNOSIS TASKS:
1. **Pattern Recognition:** Is the agent repeating the same failed logic or plan?
2. **Error Analysis:** - Is the Architect proposing plans that are too vague?
   - Is the Engineer failing to execute specific steps?
   - Is the Supervisor missing critical constraints in the final answer?
3. **Strategic Pivot:** Dynamically synthesize a high-level strategic directive to re-orient
    the Architect's approach.

OUTPUT FORMAT (JSON):
{
    "diagnosis": "The agent keeps trying to calculate X without considering Y...",
    "custom_advice": "You are ignoring the constraint about 'without moving queens'. Focus on
        knight moves."
}
```

### A.3.3 Spectrum Search Agent (Finance)

This section details the parallel generation and judging prompts used for financial queries. Temperature for generation was set to 0.7 and temperature for Judge was set to 0.

- Candidate Agent
    - Role: Quantitative Financial Analyst
    - Goal: Generate a top-tier answer. (Count: 8 parallel candidates).

**System Prompt: Candidate**

```
You are a Quantitative Financial Analyst candidate.
You will receive a Finance Question or Context.

Your goal is to provide a "Top Tier" answer that is accurate, specific, and professional.
Focus on explaining the concepts clearly and citing relevant examples.
```

*Context: Receives the full conversation history. Ends with the current User Question (and optional image).*

- Judge Agent
    - Role: Chief Investment Officer (CIO)
    - Goal: Select the best answer from the candidates.

**System Prompt: Judge (CIO)**

```
You are a Chief Investment Officer (CIO) and Portfolio Manager.
You have a Question and multiple Candidate Answers from your analysts.
```

```
Your job is to select the BEST answer based on:
1. **Mechanistic Depth**: Does it explain the root cause/driver?
2. **Specificity**: Does it use real-world examples/regulations?
3. **Risk Awareness**: Does it cover downside risks?
4. **Clarity**: Is it easy to understand?

Output JSON: { "best_candidate_index": int }
```

**Input Template: Judge's View**  The Judge receives the full conversation history (User turns + final question). Then, the candidate answers are appended to the final user turn.

**Input: Judge Context**

```
[History...]

[User]: {current_question}
[Optional Image Attachment]

Here are the candidate answers from your analysts:

--- Candidate 0 ---
Answer: {answer_0}

--- Candidate 1 ---
Answer: {answer_1}

...

--- Candidate n ---
Answer: {answer_n}

Select the index (0-based) of the best candidate.
```

### A.3.4 Direct Chain Agent (Finance)

Temperature = 0.7

- Role: Distinguished Professor of Finance

- Goal: Derive answers from first principles using logical steps.

**System Prompt: Direct Chain Agent**

```
You are a Distinguished Professor of Finance.
The user will provide a Finance Question or Context.
You must derive the answer from first principles.

Break down complex problems into simple, logical steps. Use analogies where helpful.
```

**Dynamic Context & Flow**  The context is constructed linearly. The agent sees the following sequence:

**A. Conversation History (Injected First)**    All previous turns are passed to the model's history *before* the current question.

---
**Context: Conversation History**

```
[User]: {prompt_0}
[Assistant]: {response_0}
...
[User]: {prompt_N}
[Assistant]: {response_N}
```
---

**B. Current Turn**    After the history, the current tasks are appended.

---
**Context: Current Turn**

```
{current_question}
[Optional Image Attachment]
```
---

### A.4  Neuro-Symbolic (ARC-AGI-2)

### A.4.1  Adversarial Refinement Agent (ARC-AGI)

Temperature was not set for this agent / left to default.

- Vision Agent (Proposer) Prompts
  - Initial Observation (First Training Pair)
  - Context: The agent sees the first input/output pair.

---
**Input: Initial Observation**

```
Here is the first training pair for an ARC task.
Input Grid:
[NUMPY_ARRAY]
Output Grid:
[NUMPY_ARRAY]
[ATTACHED IMAGE: Input/Output rendered]

What do you see? Describe the objects, potential transformation, and pattern.
```
---

- Subsequent Observations
  - Context: The agent sees pairs $2 \ldots N$.

---
**Input: Subsequent Observation**

```
Here is pair [INDEX].
Input Grid:
[NUMPY_ARRAY]
Output Grid:
[NUMPY_ARRAY]
```
---

```
[ATTACHED IMAGE: Input/Output rendered]

What do you see now? Does this confirm or change your previous hypothesis?
```

- Rule Consolidation (Pre-Debate)
    - Context: After observing all pairs, the agent must commit to a rule.

**Instruction: Rule Consolidation**

```
Based on all the examples you have seen, strictly state your
FINAL TRANSFORMATION RULE.
```

- Verifier Agent (Critic) Prompts
    - Initial Verification
    - Context: Agent 2 receives all training data and Agent 1's proposed rule.

**System Prompt: Initial Verification**

```
I am sharing all the training inputs for this ARC task with you.
--- Pair 1 ---
Input: [GRID]
Output: [GRID]
[ATTACHED IMAGE]
... (Repeated for all pairs) ...

Another pro agent analyzed these pairs individually and proposed
this TRANSFORMATION RULE:

[AGENT_1_PROPOSED_RULE]

What do you think? Is it correct? If yes, say 'YES' or 'PASS'. I
f no, say 'NO' or 'FAIL' and provide the reason.
```

- Debate Step (Counter-Argument)
    - Context: If Agent 1 provides a rebuttal, Agent 2 is asked to re-evaluate.

**Input: Re-evaluation**

```
Agent 1 replied:
[REBUTTAL_TEXT]

What do you think now? PASS or FAIL?
```

- Final Code Verification
    - Context: After Agent 1 generates python code that passes training data, Agent 2 gives a final confidence vote.

**Input: Final Code Check**

```
Here is my final code, it passed on all train pairs.

[GENERATED_CODE]

Are you also confident that it will pass hidden test case too?
Please reply YES or NO and give your explanation.
```

- Feedback & Refinement Loops
  - Rebuttal Request (Agent 1)
  - Context: When Agent 2 rejects the rule.

**Instruction: Rebuttal**

```
The Verifier analyzed your rule and said:
[VERIFIER_CRITIQUE]

Please revise your thought or explain why you are correct.
```

- Code Generation Request
  - Context: Converting the agreed-upon rule to code.

**Instruction: Code Generation**

```
[CONTEXT_MSG or "Your transformation rule is correct..."]

Provide python code.
If you want to run an adhoc test/diagnosis, start with:
# TYPE: DIAGNOSIS

If you are confident and providing the solution, start with:
# TYPE: SOLUTION

IMPORTANT: Your 'solve(grid)' function MUST return a 'numpy.ndarray', NOT a list.

Return ONLY the code block.
```

- Visual Feedback (Execution Failure)
  - Context: If the code runs but produces incorrect output on training data.

**Feedback: Execution Failure**

```
Execution Results:
Pair [N]: FAIL - [REASON]
...

See the attached visualization of your failure (Input | Prediction | Expected).
```

```
[ATTACHED IMAGE: Side-by-side comparison of failure]
```

### A.4.2  Hierarchical Strategic Planning Agent (ARC-AGI)

This section details the meta-reasoning layer used when the primary solver fails. Temperature was set to default here / not set explicitly.

**1. The Diagnosis Prompt**   This is the core prompt sent to the Meta-Reviewer when the primary solver gets stuck. It aggregates the history of failures and asks for a strategic intervention.

---

**System Prompt: Meta-Reviewer Diagnosis**

```
You are a Senior Meta-Reviewer for an AI Autonomous Coding Agent.
The Agent is stuck in a loop trying to solve an ARC task.

HISTORY OF ATTEMPTS:
[HISTORY_LOG_SUMMARY]
(e.g., "Turn 1: Failed - IndexError", "Turn 2: Failed - Robustness Error")

YOUR DIAGNOSIS TASKS:
1. **Pattern Recognition:** Is the agent repeating the same failed logic?
2. **Error Analysis (CRITICAL):** - Look for "ROBUSTNESS FAILURE" or "Color Invariance" in
    the logs.
  - If present, the Agent is hardcoding colors (e.g. 'grid[mask] = 3'). You MUST stop this.

OUTPUT FORMAT (JSON):
{
    "diagnosis": "The agent is failing Color Invariance checks by hardcoding '3'.",
    "custom_advice": "Stop using magic numbers. Infer the target color from the example pairs
        ."
}
```

---

**2. Role-Based System Prompts**   The Hierarchical stratrgic planning agent orchestrates three specialized sub-agents.

**2.1. Lead Architect**   Responsible for high-level reasoning and hypothesis generation.

---

**System Prompt: Lead Architect**

```
You are the Lead Architect. Your goal is to find the Single Abstract Rule for an ARC task.

CRITICAL SUCCESS CRITERIA:
1. UNIVERSALITY: The rule must apply to ALL training pairs exactly the same way.
  - REJECT: "If Pair 1 do X, If Pair 2 do Y."
  - ACCEPT: "For ALL inputs, find the object with the most unique colors and rotate it 90 deg
      ."

2. VISUAL SCAN PRIORITY: LEGENDS & KEYS
  - Sometimes the rule is explicitly written in the grid as a "Legend" or "Key".
  - SCAN THE CORNERS: Look specifically at the corners ... for small, isolated clusters of
      pixels...
  - IF FOUND: Flag this immediately. This is likely a "Recoloring Key" or "Symbol Map."
```

---

```
3. REFINE LOOP: If your hypothesis requires "special cases" for different inputs, it is WRONG
    . You must discard it and REFINE the hypothesis to find the deeper, shared pattern.

Instructions:
1. Analyze Visuals: Look for 'Compass', 'Alignment', or 'Counting' patterns in the images
    first.
2. Write Universal Code: Provide a 'solve(grid)' function. Always start with 'grid = np.array
    (grid)'.
3. Dynamic Derivation: No magic numbers. Use 'np.unique' or border scanning to find values.

Output Format (JSON): {
  "visual_observation": "(REQUIRED) Describe the object/grid alignment patterns.",
  "thought_process": "Reasoning...",
  "hypothesis": "The rule is...",
  "code": "Full python code..."
}
```

**2.2. Technical Engineer**    Responsible for cleaning code and fixing syntax before execution.

**System Prompt: Technical Engineer**

```
You are the Technical Validator (Engineer). Your role is to take the python code provided by
    the Architect and prepare it for execution.

Your Responsibilities:
1. Code Assembly: Ensure the code is a complete, self-contained python block. Include 'import
    numpy as np'.
2. Technical Correction: Fix syntax errors (like missing colons) and ensure 'grid = np.array(
    grid)' is the first line.
3. Validation: If code fails, provide a specific diagnostic report.

Output Format: Return ONLY the cleaned python code block inside triple backticks.
```

**2.3. Supervisor (Loop Prevention)**    Ensures the Architect doesn't propose the same failed hypothesis twice.

**System Prompt: Supervisor**

```
You are the SUPERVISOR for an ARC-AGI Solver System.
Your goal is to prevent the "Architect" agent from getting stuck in a feedback loop.

### INPUT DATA
1. HISTORY: A list of previously attempted hypotheses and their outcomes (Pass/Fail).
2. CURRENT_PROPOSAL: The new hypothesis the Architect just generated.

### YOUR TASK
Compare the CURRENT_PROPOSAL against the HISTORY.
Determine if the Architect is proposing a "Functionally Identical" strategy to one that
    already failed.

### DEFINITION OF "FUNCTIONALLY IDENTICAL"
Two strategies are identical if they result in the same pixel manipulation, even if the
    wording is different.
```

```
* **Example of a LOOP (REJECT THIS):**
    * History: "Crop the object and rotate 90 degrees CCW." (Failed: Pixel Mismatch)
    * Current: "Identify the red shape, extract the subgrid, and turn it Left."
    * *Verdict:* REJECT. "Turn Left" is "Rotate 90 CCW". This is the same logic.

### OUTPUT FORMAT
Return a JSON object:
{
  "status": "APPROVED" | "REJECTED",
  "reasoning": "Brief explanation of why this is new or a repeat.",
  "feedback_to_architect": "If REJECTED, write a strict instruction forcing a pivot..."
}
```

**3. Interaction & Runtime Inputs**  These are the inputs sent to agents during specific phases of the execution loop.

**3.1. Architect Input**  Sent at the start of Turn 1.

**Input: Architect Task Data**

```
TASK DATA (Images and Grids):

--- PAIR 1 ---
INPUT GRID (Visual): [IMAGE_BLOB]
INPUT GRID (Numeric):
[NUMPY_ARRAY]

OUTPUT GRID (Visual): [IMAGE_BLOB]
OUTPUT GRID (Numeric):
[NUMPY_ARRAY]

... (Repeated for all training pairs) ...
```

**3.2. Supervisor Input (Loop Check)**  Sent every time the Architect proposes a new hypothesis.

**Input: Supervisor Check**

```
HISTORY:
- Attempt 1: [HYPOTHESIS] (Result: [ERROR_TYPE])
- Attempt 2: [HYPOTHESIS] (Result: [ERROR_TYPE])

CURRENT_PROPOSAL:
[NEW_HYPOTHESIS_TEXT]
```

**3.3. Architect Input (Supervisor Rejection)**  Feedback sent if the Supervisor detects a loop.

**Feedback: Supervisor Rejection**

```
SYSTEM ALERT: Your plan was REJECTED by the Supervisor.
REASON: [REASONING]
INSTRUCTION: [FEEDBACK_TO_ARCHITECT]
```

```
ACTION: Propose a DIFFERENT strategy.
```

### 3.4. Engineer Input (Code Assembly)  Sent to the Engineer to finalize the code.

**Input: Engineer Code Assembly**

```
ARCHITECT CODE:
[CODE_FROM_ARCHITECT]

TASK: Assemble into a robust 'solve(grid)' function.
REMINDER: Start with 'grid = np.array(grid)'.

CRITICAL: If the Critic suggests a specific algorithmic approach (e.g., 'Use a mask', 'Use
    flood fill'), YOU MUST ATTEMPT IT. The Critic's technical insight overrides the
    Architect's initial hypothesis if the original code is failing.
```

### 3.5 Code Critique and Refinement

**Critique**

```
    You are a Logic Rigor Critic for the ARC Challenge.

        Review the Python code below.
        Your Goal: Ensure the code implements a **Single Abstract Rule** that works for ALL
            inputs universally.
```

**Refinement**

```
        The previous code was REJECTED by the Logic Critic.

        CRITIC FEEDBACK:
        "{msg}"

        OFFENDING CODE:
        '''python
        {current_code}
        '''

        TASK:
        Rewrite the 'solve(grid)' function to fix the violation above.
        1. Address the CRITIC FEEDBACK directly.
        2. If the Critic says the logic is flawed (e.g., 'Use different logic'), YOU MAY
            CHANGE THE APPROACH.
        3. If the feedback is about style (e.g., 'Remove patching'), fix the
            implementation style.
        4. Return the full corrected Python code.
```

**Debug**

```
        DEBUG REPORT (Attempt {debug_count}/{max_retries}):
        The code executed but produced the WRONG output.
```

```
        VISUAL EVIDENCE:
        Review the attached image (Left=Input, Middle=Your Prediction, Right=Expected).

        DATA MISMATCH:
        {log_msg}

        HINT:
        {hint}

        PREVIOUS FAILED CODE (Do NOT repeat this logic):
        '''python
        {current_code}
        '''

        TASK: Write the CORRECTED Python code.
```

*Hint* comes from checks like comparing shapes from predicted and expected train pairs, for example, if rotated version matches expected then corresponding hint is injected.

**Debug code crash**

```
        CRITICAL EXCEPTION (Attempt {debug_count}/{max_retries}):
        The code crashed or failed robustness checks.

        ERROR TRACEBACK:
        {log_msg}

        PREVIOUS FAILED CODE:
        '''python
        {current_code}
        '''

        TASK: Rewrite the code to handle this edge case or error.
        Ensure color invariance and boundary checks.
```

### A.4.3   Spectrum Search Agent

This section outlines the prompt sequence for the perception-induction-coding loop.

- Phase 1: Hypothesis Generation (Perception & Induction)
    - Goal: Induce the abstract transformation rule from multimodal data (Text Grids + Images + Vision JSON (computed using skimage module)).
    - Temperature: 0.7

**System Prompt: Hypothesis Generation**

```
You are an expert ARC-AGI Solver and python Programmer.
GOAL: Induce the transformation rule from the training examples and write a python function '
    transform(input_grid)'.

--- DATA SOURCES ---
1. RAW GRID: Use for syntax (indexing, shapes).
```

```
2. VISION JSON: Use for logic (objects, colors, counting). TRUST THIS DATA.
3. IMAGE: Use for gestalt patterns (symmetry, containment).

--- VISUAL ANALYSIS REPORT (Pre-computed JSON) ---
TRUST THIS JSON OVER YOUR OWN EYES.
[VISUAL_DESCRIPTION]

--- TRAINING EXAMPLES ---
[MULTIMODAL_DATA: Text + Image for each pair]

[ERROR_CONTEXT_IF_RETRY]

--- TEST TASK ---
INPUT GRID (RAW): ...
INPUT VISION JSON: ...
INPUT IMAGE: [IMAGE_BLOB]

--- ANALYSIS INSTRUCTIONS ---
1. ANALYZE CHANGE: Compare Input JSON vs Output JSON.
2. ASK 'WHY?':
   - Why did this object move here? (Gravity? Attraction? Alignment?)
   - Why did this color change? (Intersection? Size? Enclosure?)
3. BI-DIRECTIONAL CHECK (Mental):
   - Can you reconstruct the Input from the Output? If not, is information lost? (e.g.
       projection)
   - Describe the transformation as a precise algorithm.
   - DO NOT just predict pixels. Predict the FUNCTION.

--- RESPONSE GUIDELINES ---
1. DO NOT WRITE python CODE YET.
2. FOLLOW THE FORMAT BELOW EXACTLY.
3. Your Hypothesis must be natural language.

Format:
[Analysis]
1. Observation: ...
2. Why: ...
3. Reversibility: ...
[Hypothesis]
...
```

- Phase 2: The Logic Critic (Self-Correction & Refinement)

    - Goal: Critically evaluate the initial hypothesis and produce a "Refined Algorithm". This strictly
      separates reasoning from coding.
    - Temperature: 0.7

- Iteration 1 Prompt

**Prompt: Critique & Refinement**

```
YOUR TASK:
1. CRITIQUE: What is WRONG with your Turn 1 Hypothesis? (Edge cases? Ambiguity?)
2. SIMPLICITY CHECK: ARC tasks are usually solved by 1-3 simple core rules. If your logic is
     a page long, IT IS LIKELY WRONG. Can you simplify?
```

```
3. REFINE: precise natural language algorithm.
4. MENTAL VERIFICATION: Walk your hypothesis through ALL training examples.
   - If it fails even one pixel on Example 1, IT IS WRONG. Discard it.
4. STATUS CHECK: Are you confident this algorithm is perfect?

Format:
[Critique]
...
[Refined Algorithm]
...
[Status]
CONTINUING (or READY)
```

- Subsequent Iteration Prompt

**Prompt: Critique & Refinement (Turn 2, Iteration N)**

```
PREVIOUS REFINEMENT:
[CONTEXT]

YOUR TASK:
1. CRITIQUE the Previous Refinement. Did it fix the issues? Any new edge cases?
2. SIMPLICITY CHECK: Is it still too complex? Try to cut 50% of the steps. Think in terms of
     Objects and Physics (Gravity, Collision, etc).
3. REFINE again.
3. STATUS CHECK: Are you confident now?
```

- Phase 3: The Solver (Implementation)
  - Goal: Translate the validated "Refined Algorithm" into a self-contained python function.
  - Temperature: Uses a "temperature ladder" ranging from 0.2 to 1.0, distributed linearly across $k = 8$ parallel candidates to ensure a diverse mix of conservative and creative code implementations.

**System Prompt: The Solver**

```
--- INSTRUCTIONS ---
1. Implement the solution using your Validated Hypothesis and Algorithm.
2. Simplify your logic. Focus on: Extract Object -> Transform (Rotate/Flip/Align) -> Result.
3. Write the python function 'transform(grid)'.
4. The code MUST be self-contained. DO NOT assume any external helper functions serve you.
5. Output ONLY valid python code in a code block.
6. Use the 'transform' function signature.
```

Temperature: Fixed at 0.2 to ensure precise, non-random modifications when fixing code errors.

**Repair Code**

```
    You are an expert in solving Abstract Reasoning Corpus tasks.,
    GOAL: Fix the Python Code which fails on the Training Examples.,
    "--- VISUAL CONTEXT ---",
```

```
        visual_description
        --- TRAINING EXAMPLES (VISUAL) ---,
        *image_parts,
        --- CURRENT CODE (BUGGY) ---,
        ‘‘‘python\n{code}\n‘‘‘,
        --- ERROR FEEDBACK ---,
        error_feedback,
        --- INSTRUCTIONS ---,
        1. LOOK at the images and the error feedback.,
        2. THINK STEP-BY-STEP: Analyze the Error. Is it a Logic Error, Shape Mismatch, or
             Pixel Mismatch? Why did it happen?",
        3. GENERALIZATION STRATEGY: How can you fix this logic so it works for ALL examples,
             not just this specific failure?",
        4. IMPLEMENTATION: Return the fixed Python code.,
        5. Output Format:\n[Analysis]\n...\n[Code]\n‘‘‘python\n...\n‘‘‘
```

### A.4.4 Direct Chain Agent (ARC-AGI)

- Turn 1: Analysis & Verifiable Hypothesis

  - Goal: Force the model to "show its work" by interacting with the python environment to verify hypotheses *before* committing to a solution. The model must implement a testing loop where it discards wrong hypotheses.
  - Temperature: 1.

  > **System Prompt: Analysis & Verification**
  >
  > ```
  >         I have uploaded a JSON file containing train pairs (Input/Output). I need you
  >             to reverse-engineer the exact transformation logic.
  >         Do NOT guess the logic yet. Follow this strict process using python:
  >
  >         Analyze the Object Differences: Write a script to compare the Input grid vs
  >             the Output grid for the first 3 examples.
  >         Which pixels changed?
  >         Which pixels stayed the same?
  >         Is there a 'separator' line (a unique column)?
  >
  >         Generate Hypotheses: Based on step 1, list 3 possible mathematical rules (e.g
  >             ., 'Mirroring', 'Periodicity based on count', 'Color mapping').
  >
  >         Test & Verify (Crucial):
  >         Write a function transform(input_grid) for each of your 3 hypotheses.
  >         Run these functions on the train inputs.
  >         Compare your generated output to the actual train output.
  >         If the match is not 100%, the hypothesis is WRONG. Discard it and try another.
  >
  >         Final Answer: Only output the logic that achieves a 100% match on all training
  >             examples.
  >
  >         --- TRAIN PAIRS ---
  >         [JSON_DATA]
  > ```

- Turn 2: Test Case Execution

  - Goal: If the model found a verifiable rule in Turn 1, this prompt asks it to apply that rule to the hidden test cases. It explicitly allows for a final code fix if the previous step wasn't perfect.

– Temperature: 1.

---

**System Prompt: Test Execution**

```
        Here are the test cases. I need you to run your code on ALL of them.
        Your code must handle BOTH the Training examples (perfectly) and these Test
            examples.
        If your previous code was correct, reuse it. If not, fix it so it works on all
             cases.
        Return the Final Code that solves these cases.

        --- Test Inputs ---
        [JSON_DATA]
```

---

## A.5 Synthesis Agent Prompts

Across all datasets (HLE, PRBench, and ARC-AGI-2), the temperature for the synthesis agents was set to 0.0 to ensure deterministic and reproducible aggregation of results. For the ARC-AGI-2 benchmark specifically, we also generated a second attempt using a temperature of 0.7. This aligns with the official ARC Prize submission rules, which permit up to two attempts per task. Note that this second attempt was **not** factored into our primary accuracy metrics and was used solely to observe the performance gains achievable when utilizing both allowed submissions. Furthermore, candidates_text contains its answer and rationale both.

---

**System Prompt: Humanity's Last Exam**

```
"""Problem:
{problem_context}

Candidates:
{candidates_text}

Trust your instinct and pick one answer. Provide ONLY the answer text, not the candidate
    number or reference.

Final Answer:
"""
```

---

**System Prompt: PRBench Finance**

```
You are an Expert Finance Judge and Senior Auditor.
You are given multiple candidate answers to a complex finance task.
Your goal is to SYNTHESIZE a single "Super-Answer" that combines the best parts of all
    candidates.

Problem:
{problem_context}

Candidates:
{candidates_text}

Synthesis Instructions:
```

```
1. **Enrichment**: Dynamically select the best elements from ANY candidate. For example, if
      one candidate has better calculations, use them. If another has better citations or
      formatting, use those. Do not assume specific candidates have specific strengths.
2. **Completeness**: Ensure every single user question is answered with maximum depth.
3. **Accuracy**: If candidates have conflicting numbers, perform a sanity check and use the
      most plausible/consistent one.
4. **Tone**: Professional, authoritative, and precise.
5. **Output**: Provide ONLY the final merged answer. Do not explain your process.

Final Answer:
```

**System Prompt: ARC-AGI-2**

```
You are a Principal Logic Expert and ARC-AGI Judge.
You are tasked with determining the Single Correct Output Grid for a given Test Input.
You have access to:
1. **Problem Description**: Training examples demonstrating the rule.
2. **Candidates**: Proposed solutions from other expert agents.

GOAL: Synthesize the perfect solution.
- The candidates might be correct, partially correct, or completely wrong.
- Use the Training Examples to VERIFY the rule.
- If a candidate follows the rule perfectly, select it.
- If candidates diverge, use your reasoning to correct the errors.
- If all candidates are wrong, derive the solution from scratch.

Problem:
{problem_context}

Candidates:
{candidates_text}

OUTPUT FORMAT:
Provide ONLY the final output grid as a raw JSON list of lists (e.g. [[1,0],[0,1]]).
Do not include markdown blocks ('''json) or explanations. Just the raw JSON string.

Final Answer:
```

## A.6   Baseline Prompts

Temperature was set to 0 for all the baseline runs.

**Humanity's Last Exam**

```
    # Role and Objective
You are an expert reasoning assistant helping to solve complex problems. Your goal is to
    provide the correct answer by strictly following a rigorous thinking process.
# General Rules
1. You MUST plan extensively before generating the final answer.
2. Reflect on your reasoning at each step to catch potential errors.
3. Do not just guess; derive the answer through logical deduction.
# Task Execution Rules:
1. Analysis:
  - Break down the problem into key components.
```

```
     - Identify what information is given and what needs to be inferred.
2. Step-by-Step Reasoning:
   - Execute your plan step by step.
   - For each step, explicitly state your reasoning and the conclusion of that step.
   - If a step involves calculation or logic, double-check it.
3. Reflection:
   - Before concluding, ask yourself: "Does this make sense?", "Is there a counter-example?",
       "Did I miss any constraints?"
4. Final Output:
   - Provide the final answer clearly at the end.
   - If the question asks for a specific format (e.g., a number, a code snippet), strictly
       follow it.
# Environment Information
- You are running in a restricted reasoning environment without external tools (no web search
    , no code execution).
- Rely solely on your internal knowledge and logical capabilities.

IMPORTANT: You must output your answer in valid JSON format as follows:
{
  "rationale": "Your extensive planning, step-by-step reasoning, and assertions...",
  "answer": "The final answer (e.g. 'A', 'B', '42', 'summary')"
}
```

## PRBench Finance

```
You are a Chief Financial Officer and Expert Financial Analyst.
The user will provide a Finance Question or Context.
Your goal is to provide the most ACCURATE and PRECISE answer possible.

Guidelines:
1. PRECISION: Perform all calculations with high precision. Double-check your arithmetic.
2. FIRST PRINCIPLES: Derive answers from fundamental financial concepts and regulations.
3. CONSTRAINTS: Pay close attention to every constraint and detail in the prompt.
4. COMPLETENESS: Ensure the final answer addresses the specific question asked without
    ambiguity.
5. REASONING: Briefly explain your steps to ensure correctness before stating the final
    answer.

Output JSON:
{
  "answer": "The final answer..."
}
```

## ARC AGI 2

```
I will provide you with several input and output matrices about 2D grids. You need to
    summarize the grid-changing rule from it.

Step 1: Rule Extraction First, find the matrix-changing rule from the examples.
Step 2: Rule Verification Check, the correctness of the rule based on the examples. If the
    rule is correct, apply it to the new input. Otherwise, summarize a new rule and apply it
     to the new input.
Step 3: Apply the finalized rule to the new input. Put the output matrix within \boxed{}.
```

```
Example input 1 [Matrix Data] [Image of Matrix]
Example output 1 [Matrix Data] [Image of Matrix]
... [Repeats for all training examples] ...

New Input [Matrix Data] [Image of Matrix]
```

## A.7 Self-Consistency Implementation

For the Self-Consistency (SC) baselines, we utilized the same generation prompts as the standard baseline but increased the sampling budget to $N = 8$ and $N = 16$ independent responses per problem. To encourage diversity in the reasoning paths during generation, we set the temperature to 0.7.

For tasks with structured outputs (HLE and ARC-AGI), the final prediction was selected via standard majority voting based on exact string matching. However, as PRBench Finance requires complex, free-form reasoning, exact matching is insufficient for determining consensus. To address this, we employed an additional LLM-based aggregation step (with temperature set to 0.0 for deterministic selection) to identify the semantically most common answer among the generated samples. The specific prompt used for this consensus extraction is provided below:

---

**Prompt for Majority vote**

```
You are an expert consensus judge.
You will be provided with a Question/Context, and 16 different candidate answers generated by
    an AI.
Your task is to analyze these candidate answers, determine the "majority vote" or most common
    distinct answer, and output the final consensus answer.

Think carefully about which answer is truly the most common conceptually, accounting for
    minor formatting differences or rounding.
Provide your reasoning, and then output your final majority consensus answer in the specified
    JSON format.

Output JSON:
{
  "reasoning": "Explanation of the voting...",
  "answer": "The final consensus majority answer..."
}
```

---

# B  Multi-Agent Consensus (CoScientist) Ablation

To investigate whether inter-agent collaboration could bridge the The Verification Bottleneck, we implemented a "CoScientist" consensus pipeline Gottweis et al. (2025). This architecture decomposed the selection process into five specialized roles: *Proximity* (clustering), *Ranking* (tournament comparison), *Reflection* (peer review), *Evolution* (refinement), and *Meta-Review* (final polish).

We evaluated this pipeline by feeding the raw PoTRE candidate outputs directly into the consensus agent. Empirical results indicate that this added complexity did not yield improvements over existing Synthesizer, and in some cases degraded performance:

- **Humanity's Last Exam (HLE):** The consensus pipeline achieved an accuracy of **39.60%** (990/2500), under performing the standard PoTRE 3.1 Pro result (49.92%).

- **PRBench Finance:** The pipeline attained an Average Clipped Score of **0.3185** (vs. 0.3486 for PoTRE).

- **ARC-AGI-2:** Accuracy dropped to **19.16%** (23/120), significantly lower than the PoTRE performance (38.30%).

These findings suggest that while iterative debate is intuitively appealing, it may introduce "consensus collapse," where agents converge on plausible-sounding but incorrect answers, particularly in abstract reasoning tasks. For transparency, the exact system prompts used for each role are provided below.

**Note on Co-Scientist Implementation**: In this ablation study, the Co-Scientist workflow was implemented primarily as a consensus and synthesis mechanism to aggregate candidate answers, rather than as a full generation pipeline. We followed the structural design of the original work, utilizing Proximity, Ranking, Reflection, Evolution, and Meta-Review agents. Because the task was restricted to comparing and refining existing solutions, the prompts were naturally more concise (focusing on pairwise comparisons and peer review) compared to the elaborate persona and planning prompts used in PoTRE's generation phase. We acknowledge that further prompt engineering tailored specifically to the reasoning tasks could improve its performance, and we leave this prompt optimization as an area for future exploration.

## B.1 Prompts for HLE and PRBench Finance

**Proximity Agent (Data Clustering)**

```
system_instruction = "You are a Data Clustering Agent."

prompt = f"""Analyze the following candidate answers. Group IDs that represent the
    SUBSTANTIALLY SAME answer or logic.
Output a JSON list of lists, where each inner list contains IDs of identical answers.
Example: [[0, 2], [1], [3, 4]]

Candidates:
{candidates_text}
"""
```

**Ranking Agent (Tournament logic)**

```
system_instruction = "You are a Tournament Judge. Pick the best answer."

prompt = f"""Compare Option A and Option B for the following problem.
Which is more accurate, rigorous, and complete?
Output ONLY 'A' or 'B'.

Problem:
{self.context}

Option A:
{ans_a[1]}

Option B:
{ans_b[1]}
"""
```

**Reflection Agent (Peer Reviewer)**

```
system_instruction = "You are a Scientific Peer Reviewer."
```

```
prompt = f"""Review the following solution to the problem below.
Identify:
1. Correctness errors
2. Missing details or derivations
3. Logical gaps or weak reasoning
Be extremely critical and rigorous.

Problem Context:
{self.context}

Proposed Solution:
{candidate_answer}
"""
```

## Evolution Agent (Refinement)

```
system_instruction = "You are a Research Evolution Agent."

prompt = f"""You are the Evolution Agent.
Refine the following 'Best Answer' based on the 'Critique'.
Synthesize existing knowledge, fix errors, and clarify concepts.
Ensure the final answer is complete and solves the original problem perfectly.

Problem:
{self.context}

Best Answer So Far:
{best_candidate[1]}

Critique from Peer Review:
{critique}

Task: Rewrite the answer to be perfect.
"""
```

## Meta-Review Agent (Final Polish)

```
system_instruction = "You are a Meta-Review Editor."

prompt = f"""Finalize this answer.
Ensure the tone is professional, authoritative, and precise.
Output ONLY the final answer content. Do not include "Here is the answer" or similar
    preambles.

Draft:
{final_draft}
"""
```

## B.2 Prompts for ARC AGI 2

### Proximity Agent (Data Clustering)

```
Analyze the following candidate answers. Group IDs that represent the SUBSTANTIALLY SAME
    answer, logic, or grid structure.
Output a JSON list of lists. Example: [[0, 2], [1], [3]]

Candidates:
[List of candidate IDs and their answers]
```

### Ranking Agent (Tournament Logic)

```
Compare Option A and Option B for the following task.
Which is more accurate, correct, and follows the pattern?
For ARC/Grid tasks, check which output accurately follows the transformation rules seen in
    training for ALL test inputs.
Output ONLY 'A' or 'B'.

Task Context:
[Task Context string containing training and test grids]

Option A:
[Candidate A string]

Option B:
[Candidate B string]
```

### Reflection Agent (Peer Reviewer

```
Review the following solution.
Identify:
1. Pattern violations (if ARC/Grid task)
2. Logical gaps
3. Correctness errors
Be extremely critical.

Context:
[Task Context string containing training and test grids]

Proposed Solution:
[The winning candidate answer from the Ranking Agent]
```

### Evolution Agent (Refinement)

```
You are the Evolution Agent.
Refine the 'Best Answer' based on the 'Critique'.
If this is an ARC Grid task, ensure the final output contains perfectly valid grids for ALL
    test inputs and follows the training pattern.
Output the FINAL, CORRECTED answer.

Context:
```

```
[Task Context string containing training and test grids]

Best Answer So Far:
[The winning candidate answer from the Ranking Agent]

Critique:
[The critique generated by the Reflection Agent]
```

**Meta-Review Agent (Final Polish)**

```
Finalize this answer.
Output ONLY the final answer content as a SINGLE valid JSON array containing ALL the test
    output grids.
For example, if there are two test inputs, output:
[
  [[0, 0], [0, 0]],
  [[1, 1], [1, 1]]
]
Do NOT include any explanations, markdown formatting, or multiple code blocks. Return ONLY
    the JSON array.

Draft:
[The refined final draft from the Evolution agent]
```

## C  Algorithms for each Sub-Agent

All algorithms for, Adversarial Refinement agent, Hierarchical Strategic Planning Agent, Spectrum Search Agent and Direct Chain Agent are provided as Algorithm 2, Algorithm 3, Algorithm 4 and Algorithm 5 respectively.

---

**Algorithm 2** Adversarial Refinement Agent

---

**Require:** Task Context $\mathcal{T}$ (text, images, or grid pairs), Proposer Agent $\mathcal{A}_P$, Verifier Agent $\mathcal{A}_V$, Max Debate Turns $N_{max}$
**Ensure:** Final Consensus Solution $S_{final}$ (Code or JSON structure)
    % **Step 1: Initialization & Initial Proposal**
1: $S_{cand} \leftarrow \mathcal{A}_P.\text{GenerateProposal}(\mathcal{T})$         ▷ Initial generation (hypothesis, logic, or code)
2: $turn \leftarrow 0$
    % **Step 2: Debate & Verification Loop**
3: **while** $turn < N_{max}$ **do**
4:     $Critique, Status \leftarrow \mathcal{A}_V.\text{Evaluate}(S_{cand}, \mathcal{T})$         ▷ Verifier analyzes the candidate
5:     **if** $Status = \text{APPROVED}$ **then**
6:         **break**         ▷ Consensus reached
7:     **end if**
8:     $Feedback \leftarrow \text{FORMATREBUTTAL}(Critique)$
9:     $S_{cand} \leftarrow \mathcal{A}_P.\text{RefineProposal}(S_{cand}, Feedback)$         ▷ Proposer incorporates critique
10:     $turn \leftarrow turn + 1$
11: **end while**
    % **Step 3: Final Output**
12: $S_{final} \leftarrow S_{cand}$
13: **return** $S_{final}$         ▷ Returns agreed solution, or best-effort if max turns reached

---

---

**Algorithm 3** Hierarchical Strategic Planning Agent

---

**Require:** Task Context $\mathcal{T}$, Agent Ensemble $\{\mathcal{A}_{Arch}, \mathcal{A}_{Eng}, \mathcal{A}_{Sup}, \mathcal{A}_{Meta}\}$, Max Turns $T_{max}$, Max Debug Retries $D_{max}$

**Ensure:** Final Answer $A_{final}$

1: $L_{history} \leftarrow [\,]$                        ▷ Log of execution trajectories
2: $P_{Arch} \leftarrow$ Base System Prompt
3: **for** $t = 1$ **to** $T_{max}$ **do**
       % **Phase 1: Meta-Strategic Intervention**
4:     **if** SHOULDINTERVENE$(t, L_{history})$ **then**
5:         $Review \leftarrow \mathcal{A}_{Meta}.\text{AnalyzeTrajectory}(L_{history}, \mathcal{T})$
6:         $P_{Arch} \leftarrow$ INJECTHEURISTIC$(P_{Arch}, Review)$        ▷ Force strategic pivot
7:     **end if**
       % **Phase 2: Hypothesis Generation**
8:     $Plan \leftarrow \mathcal{A}_{Arch}.\text{GeneratePlan}(\mathcal{T}, P_{Arch})$
       % **Phase 3: Concept Verification (For Formal/Strict Tasks)**
9:     **if** $\mathcal{T}$ requires strict logical constraints **then**
10:         $Valid, Feedback \leftarrow \mathcal{A}_{Sup}.\text{EvaluateLogic}(Plan)$
11:         **if not** $Valid$ **then**
12:             $P_{Arch} \leftarrow \text{Update}(P_{Arch}, Feedback)$        ▷ Reject flawed logic immediately
13:             **continue**
14:         **end if**
15:     **end if**
       % **Phase 4: Implementation**
16:     $Draft \leftarrow \mathcal{A}_{Eng}.\text{GenerateDraft}(\mathcal{T}, Plan)$
       % **Phase 5: Domain-Adaptive Validation & Engineer Auto-Debugging**
17:     **if** $\mathcal{T}$ requires environment execution (e.g., ARC) **then**
18:         $d \leftarrow 0, Success \leftarrow$ False
19:         **while** $d < D_{max}$ **and not** $Success$ **do**        ▷ Internal Engineer Repair Loop
20:             $Valid, ErrorMsg \leftarrow$ EXECUTEANDREVIEW$(Draft, \mathcal{T})$
21:             **if** $Valid$ **then**
22:                 $Success \leftarrow$ True
23:             **else**
24:                 $Draft \leftarrow \mathcal{A}_{Eng}.\text{RefineDraft}(Draft, ErrorMsg)$        ▷ Pass tracebacks to Engineer
25:                 $d \leftarrow d + 1$
26:             **end if**
27:         **end while**
28:         **if not** $Success$ **then**
29:             $P_{Arch} \leftarrow \text{Update}(P_{Arch}, \text{"Execution Failed: "} + ErrorMsg)$        ▷ Propagate to Architect
30:             **continue**
31:         **end if**
32:         $A_{final} \leftarrow Draft$
33:     **else**                           ▷ Factual / reasoning tasks (e.g., HLE)
34:         $A_{final}, Rationale \leftarrow \mathcal{A}_{Sup}.\text{ReviewAndSynthesize}(Draft, \mathcal{T}, Plan)$
35:         $Success \leftarrow$ True
36:     **end if**
37:     $L_{history}.\text{append}(\{Plan, Draft, Success\})$
38:     **if** $Success$ **then**
39:         **break**                   ▷ Task solved, exit macro-loop
40:     **end if**
41: **end for**
42: **return** $A_{final}$

---

---

**Algorithm 4** Spectrum Search Agent

---

**Require:** Task Context $\mathcal{T}$, Number of generative agents $N$
**Ensure:** Optimal verified solution $s^*$

    % **Initialization**
1: $\mathcal{C} \leftarrow \emptyset$                                                        ▷ Candidate solution pool
    % **Phase 1: High-Concurrency Hypothesis Generation**
2: **for** $i = 1, \ldots, N$ **in parallel do**
3:     $\tau_i \leftarrow \text{ScaleTemperature}(i, N)$                    ▷ Linearly scale temperature $T \in [0.2, 1.0]$ for diversity
4:     $c_i \leftarrow \text{GeneratorAgent}_\theta(\text{prompt} = \mathcal{T}, \text{temp} = \tau_i)$
5:     **if** $\mathcal{T}$ requires Code Execution (e.g., ARC) **then**
6:         $c_i \leftarrow \text{RefinementLoop}(c_i, \mathcal{T}, k_{\max} = 4)$            ▷ Iterative self-critique prior to execution
7:     **end if**
8:     $\mathcal{C} \leftarrow \mathcal{C} \cup \{c_i\}$
9: **end for**
    % **Phase 2: Domain-Specific Verification & Consensus**
10: **if** $\mathcal{T}$ requires Code Execution (e.g., ARC) **then**
       % **Execution-Based Verification**
11:     $\mathcal{D}_{\text{train}} \leftarrow \text{ExtractConstraints}(\mathcal{T})$
12:     $\mathcal{C}_{\text{valid}} \leftarrow \{c \in \mathcal{C} \mid c(x) = y, \forall (x, y) \in \mathcal{D}_{\text{train}}\}$
13:     **if** $\mathcal{C}_{\text{valid}} = \emptyset$ **then**
14:         $\mathcal{C}_{\text{valid}} \leftarrow \text{ExecutionFeedbackRepair}(\mathcal{C}, \mathcal{D}_{\text{train}})$           ▷ Traceback-guided repair
15:     **end if**
16:     $Y_{\text{consensus}} \leftarrow \text{MajorityVote}(\{c(\mathcal{T}_{\text{test}}) \mid c \in \mathcal{C}_{\text{valid}}\})$
17:     $s^* \leftarrow \arg\min_{c \in \mathcal{C}_{\text{valid}}, c(\mathcal{T}_{\text{test}}) = Y_{\text{consensus}}} \text{Complexity}(c)$
18: **else**
       % **Semantic/Factual Verification (e.g., HLE)**
19:     $\mathcal{P}_{\text{eval}} \leftarrow \mathcal{T} \cup \text{Serialize}(\mathcal{C})$
20:     $idx^*, \text{rationale}^* \leftarrow \text{ExpertJudgeAgent}_\psi(\text{prompt} = \mathcal{P}_{\text{eval}}, \text{temp} = 0.0)$
21:     $s^* \leftarrow \mathcal{C}[idx^*]$                              ▷ Select highest fidelity solution candidate
22: **end if**
23: **return** $s^*$

---

---

**Algorithm 5** Direct Chain Agent

---

**Require:** Task instance $\mathcal{T}$, Target model $\mathcal{M}$, Max API retries $R_{max}$
**Ensure:** Structured output $O$ (e.g., Parsed JSON or Executable Code)

    **% Step 1: Initialization & Configuration**
1:  $P \leftarrow \textsc{ConstructPrompt}(\mathcal{T})$            ▷ Domain-specific prompt, incl. images/text/examples
2:  $config \leftarrow \{\text{temperature} : \tau\}$
3:  $attempt \leftarrow 0$
4:  $wait\_time \leftarrow W_{init}$            ▷ Initial delay penalty
    **% Step 2: Robust Execution Loop**
5:  **while** $attempt < R_{max}$ **do**
6:     **try**
7:        $Response \leftarrow \textsc{GenerateContent}(\mathcal{M}, P, config)$
8:        **if** $Response$ is **null or** $Response.text$ is empty **then**
9:           **raise** ResponseContentError
10:       **end if**
      **% Step 3: Parsing & Validation**
11:      $O \leftarrow \textsc{RobustParse}(Response.text)$         ▷ Regex extraction & fallback logic
12:      **if** $O$ is valid **and** meets structural constraints **then**
13:         **return** $O$
14:      **else**
15:         **raise** StructuralParsingError
16:      **end if**
17:    **catch** APIError $E$          ▷ e.g., 429 Rate Limit, 503 Service Unavailable
      **% Step 4: Exponential Backoff with Jitter**
18:      $jitter \leftarrow \text{Uniform}(0.8, 1.2)$         ▷ Prevent synchronized retry cascades
19:      $delay \leftarrow \min(W_{max}, wait\_time \times jitter)$
20:      $\text{Sleep}(delay)$
21:      $wait\_time \leftarrow \min(W_{max}, wait\_time \times \alpha)$    ▷ Exponential multiplier, e.g., $\alpha = 1.5$ or $2.0$
22:      $attempt \leftarrow attempt + 1$
23:    **end try**
24:  **end while**
25:  **raise** ExceededMaxRetriesError

---

Table 18: **Claude-4.5-Sonnet Thinking Results.** Performance evaluation of the PoTRE framework using the Claude-4.5-Sonnet Thinking model across three distinct benchmarks. The results demonstrate consistent improvements over both baseline and Self-Consistency (SC).

| | Baseline Evaluation | | | PoTRE Evaluation | |
|---|---|---|---|---|---|
| Benchmark | Prior Work Prompts | SC ($N = 8$) | SC ($N = 16$) | Score | Improv.[‡] |
| Humanity's Last Exam | 12.08% | 12.39% | 12.61% | **15.20**% | +3.12 |
| ARC-AGI 2 | 5.00% | 5.00% | 6.66% | **10.83**% | +5.83 |
| PRBench Finance Hard | 0.2625 | 0.2222 | 0.2343 | **0.5196** | **+0.2571** |

[‡]Improvement calculated vs. Baseline (Prior work prompts).

## D    Experiments with Claude Sonnet 4.5 Thinking

To verify that PoTRE's architectural gains are fundamentally model-agnostic and not overfit to the Gemini family, we extended our evaluation to Claude-4.5-Sonnet Thinking. The comprehensive results across all three benchmarks are detailed in Table 18. On the PRBench Finance - Hard subset, this model achieved an average clipped score of **0.5196**, surpassing the baseline of 0.2713 and nearly doubling the margin by **0.2571** points. These results represent a new state-of-the-art on this benchmark[5].

*Note on ARC-AGI-2 Performance:* The official task page[6] indicates that Claude-4.5-Sonnet Thinking (across 8K, 16K, and 32K configurations) failed all tasks within the public evaluation sets. This baseline deficiency highlights the model's inherent struggle with this specific spatial reasoning benchmark, effectively contextualizing the low absolute accuracy observed in our results.

### D.1    Ablation Study: Contribution of Individual agents

We conducted same ablation study we've performed for other models to understand the contributions of each sub-agent.

#### D.1.1    Humanity's Last Exam (HLE)

To validate the necessity of a diverse multi-agent ensemble, we conducted a component ablation on the HLE dataset, tracking both total and uniquely solved tasks across all 2500 problems (Table 19). While the Adversarial Refinement Agent achieved the highest independent performance (319 solved, 12.76% accuracy), the most critical finding lies in the "Exclusive" column. Every distinct reasoning topology successfully resolved a substantial number of tasks that all other agents failed to answer, ranging from 57 to 92 uniquely solved problems per agent. Because these diverse reasoning paths somewhat cover each other's structural blind spots, the theoretical Oracle upper bound—representing tasks where at least one agent produced the correct answer—reaches 592 solved tasks (23.7%). This nearly doubles the highest single-agent baseline, empirically demonstrating that topological diversity is essential for maximizing performance on highly constrained, complex reasoning benchmarks.

#### D.1.2    PRBench Finanace Hard

To further understand the topological preferences of different frontier models, we conducted a component ablation on the PRBench Finance Hard subset (Table 20). The analysis reveals a distinct architectural preference: Claude-4.5-Sonnet Thinking strongly favors the Hierarchical Strategic Planning Agent as its primary reasoning engine, achieving a leading individual Average Clipped score of 0.4038. However, the true strength of the PoTRE framework emerges at the synthesis agent. By integrating the diverse reasoning trajectories

---

[5]Official Leaderboard: https://scale.com/leaderboard/prbench-finance
[6]ARC-AGI-2 Tasks: https://arcprize.org/tasks/?dataset=arc-agi-2&model=claude-sonnet-4-5-20250929-thinking-32k

Table 19: **Component analysis on HLE.** Performance breakdown of individual reasoning agents across 2500 tasks. The "Exclusive" column indicates the number of tasks solved uniquely by that specific agent. The combined Oracle upper bound (where at least one agent produced the correct answer) reached 592 solved tasks (23.7%).

| Sub-Agents | Solves | Exclusive | Accuracy |
|---|---|---|---|
| Adversarial Refinement Agent | **319** | **92** | **12.76%** |
| Hierarchical Strategic Planning Agent | 283 | 66 | 11.32% |
| Direct Chain Agent (CoT) | 283 | 60 | 11.32% |
| Spectrum Search Agent | 287 | 57 | 11.48% |
| **Oracle Upper Bound** | **592** | – | **23.7%** |

Table 20: **Component Analysis on PRBench Finance (Hard).** Performance breakdown of individual reasoning agents compared to the final PoTRE synthesis. While the Hierarchical Strategic Planning Agent serves as the strongest individual topology, the final synthesis provides a massive lift, confirming the value of multi-agent aggregation.

| Sub-Agents | Avg Norm | Avg Clip |
|---|---|---|
| Adversarial Refinement Agent | 0.4095 | 0.3613 |
| Hierarchical Strategic Planning Agent | **0.4507** | **0.4038** |
| Spectrum Search Agent | 0.4346 | 0.3882 |
| Direct Chain Agent | 0.4069 | 0.3602 |
| PoTRE Final Synthesis | **0.5565** | **0.5196** |

generated by the parallel sub-agents, the final synthesis unlocks a massive domain-specific performance gain for Claude, elevating its Average Clipped score from 0.4038 to a state-of-the-art 0.5196.

### D.1.3 ARC-AGI 2

Turning to spatial reasoning on ARC-AGI-2 (Table 21), this model demonstrates a distinct reliance on the Spectrum Search Agent, which leads the individual sub-agents with 9 solved tasks. The PoTRE final synthesis successfully integrates these diverse reasoning paths, elevating the overall performance to 13 solved tasks. This demonstrates a clear improvement over the best single-agent baseline, though it remains below the theoretical Oracle upper bound of 15.

**Summary of Ablation** Across three highly distinct domains—scientific knowledge (HLE), complex financial reasoning (PRBench), and strict spatial logic (ARC-AGI-2)—our ablation studies consistently demonstrate that the PoTRE framework is fundamentally greater than the sum of its individual components. Rather than relying on a single dominant reasoning topology, the architecture actively exploits topological diversity. On HLE, this diversity proved essential, as the agents solved distinct, mutually exclusive subsets of problems, effectively doubling the best single-agent baseline. Conversely, on PRBench and ARC-AGI-2, while the base models exhibited clear topological preferences for specific domains (favoring Hierarchical Planning for finance and Spectrum Search for spatial tasks), the final synthesis agent consistently extracted marginal value from the parallel agents. By successfully integrating these diverse reasoning trajectories, PoTRE systematically narrows the gap toward theoretical Oracle upper bounds, confirming that structured, multi-agent aggregation is a robust and domain-agnostic mechanism for scaling inference-time compute.

Table 21: **Component Analysis on ARC-AGI-2.** Performance breakdown of individual reasoning agents compared to the PoTRE final synthesis. The Spectrum Search Agent serves as the primary reasoning engine, while the synthesis agent further elevates the final score toward the Oracle upper bound.

| Sub-Agents | Solved | Exclusive |
|---|---|---|
| Adversarial Refinement Agent | 7 | 1 |
| Hierarchical Strategic Planning Agent | 3 | 0 |
| Spectrum Search Agent | **9** | **7** |
| Direct Chain Agent | 3 | 1 |
| Oracle Upper Bound | **15** | – |
| **PoTRE Final Synthesis** | **13** | – |

Table 22: **DeepSeek Results.** Performance evaluation of the PoTRE framework using the DeepSeek model across distinct reasoning benchmarks. The results highlight the multi-agent synthesis gains compared to standard baselines and Self-Consistency (SC).

| Benchmark | Baseline Evaluation | | | PoTRE Evaluation | |
|---|---|---|---|---|---|
| | Prior Work Prompts | SC ($N = 8$) | SC ($N = 16$) | Score | Improv.[‡] |
| Humanity's Last Exam | 18.16% | 19.46% | 19.83% | **20.20**% | +2.04 |
| PRBench Finance Hard | 0.2299 | 0.0717 | 0.0991 | **0.3499** | **+12.00** |

[‡]Improvement calculated vs. Baseline (Prior work prompts).

# E    Experiments with Deepseek-V3.2-maas

To further establish the model-agnostic versatility of the PoTRE framework, we expanded our evaluation to include the DeepSeek model. As given in table 22, the PoTRE-enhanced DeepSeek achieved an accuracy of 20.20% on the Humanity's Last Exam (HLE) text-only subset. Furthermore, on the complex financial reasoning tasks of the PRBench Finance - Hard subset, the model delivered an average clipped score of 0.3499. These results reinforce that our multi-agent synthesis effectively extracts structural reasoning gains across diverse frontier architectures, proving the framework's viability extends well beyond the Gemini and Claude models.

## E.1    Ablation Study: Contribution of Individual agents

We conducted same ablation study we've performed for other models to understand the contributions of each sub-agent.

### E.1.1    Humanity's Last Exam (Text-Only)

To unpack the structural advantages of the PoTRE framework when applied to the DeepSeek architecture, we conducted a component ablation on the HLE text-only subset (Table 23). The Spectrum Search Agent emerges as the dominant individual topology for this model, independently solving 441 tasks (20.4% accuracy) and contributing a massive 128 exclusive solves. The ensemble's topological diversity remains robust, with the remaining agents successfully resolving between 64 and 66 exclusive tasks each, expanding the theoretical oracle upper bound to 696 solved tasks (32.3%). However, the actual PoTRE synthesis achieves only 20.20% accuracy, slightly trailing the best single-agent baseline. This substantial gap between the realized performance and the oracle bound exposes a critical limitation: while the parallel agents successfully generate the correct reasoning path, the final synthesis agent struggles to consistently filter out the noise from

Table 23: **Component Analysis of DeepSeek on HLE (Text-Only).** Performance breakdown of individual reasoning agents across 2158 tasks. The "Exclusive" column indicates the number of tasks solved uniquely by that specific agent. The combined Oracle upper bound (where at least one agent produced the correct answer) reaches 696 solved tasks (32.3%).

| Sub-Agents | Solves | Exclusive | Accuracy |
|---|---|---|---|
| Spectrum Search Agent | **441** | **128** | **20.4%** (441/2158) |
| Hierarchical Strategic Planning Agent | 335 | 66 | 15.5% (335/2158) |
| Direct Chain Agent | 332 | 66 | 15.4% (332/2158) |
| Adversarial Refinement Agent | 245 | 64 | 11.4% (245/2158) |
| **Oracle Upper Bound** | **696** | – | **32.3%** |

Table 24: **Component Analysis of DeepSeek on PRBench Finance (Hard).** Performance breakdown of individual reasoning agents compared to the final PoTRE synthesis. The synthesis agent successfully integrates diverse reasoning paths to achieve a final Average Clipped score of 0.3499, significantly outperforming the best individual agent.

| Sub-Agents | Avg Norm | Avg Clip |
|---|---|---|
| Spectrum Search Agent | **0.3577** | **0.3053** |
| Adversarial Refinement Agent | 0.3509 | 0.2993 |
| Direct Chain Agent | 0.3277 | 0.2730 |
| Hierarchical Strategic Planning Agent | 0.3260 | 0.2701 |
| **PoTRE Final Synthesis** | **0.3987** | **0.3499** |

incorrect agents. Rather than extracting these unique correct answers, the synthesis bottleneck occasionally degrades the optimal yield, highlighting a primary area for future routing and consensus optimization.

### E.2 PRBench Finance (Hard)

To evaluate DeepSeek's performance on complex financial reasoning, we conducted a component ablation on the PRBench Finance Hard subset (Table 24). The DeepSeek model strongly favors the Spectrum Search Agent, which achieves a leading individual average clipped score of 0.3053. Notably, unlike the synthesis bottleneck observed in the HLE text-only evaluation, the PoTRE framework demonstrates high effectiveness in this domain. By successfully synthesizing the diverse reasoning trajectories generated by the parallel sub-agents, the architecture elevates the final Average Clipped score to 0.3499. This substantial gain over the best single-agent baseline confirms that structured, multi-agent aggregation effectively extracts and amplifies the underlying model's financial reasoning capabilities.

## F   LLM-as-a-Judge vs. Exact Match Evaluation for HLE

Our evaluation methodology strictly adhered to the standardized protocol established for the HLE benchmark. Because strict string matching is fundamentally inadequate for assessing the multidisciplinary complexity of these tasks, the official evaluation employs an LLM-as-a-judge equipped with a specialized extraction prompt.

To empirically validate the reliability of the automated judge and quantify the limitations of exact string matching, we conducted a human-subject evaluation using a randomized sample of 100 tasks. Expert human annotators established a ground-truth accuracy of 55.0 % on this subset. This human baseline underscored

the strictness of case-insensitive exact matching, which yielded a significantly lower accuracy of 34.0 %. In contrast, the LLM-based judge achieved a 57.0 % accuracy. Crucially, an item-level concordance analysis revealed a 98.0 % agreement rate between the automated judge and the human evaluators. This high degree of inter-rater reliability demonstrates that the automated judge successfully mitigates the false negatives inherent in exact matching, establishing it as a robust surrogate for expert human assessment.

## F.1 Analysis of the Discrepancy (False Negatives in Exact Match)

Our sample analysis revealed that strict exact matching fails primarily due to formatting and representation differences in correct answers, which the LLM judge successfully normalizes and identifies. Common edge cases included:

- **Equivalent Mathematical Expressions:** For example, a ground truth of $\frac{n(6L^2+4-4L)+L^3+2L(1-L)}{48n^3}$ versus a prediction of $\frac{3L^2-2L+2}{24n^2} + \frac{L^3-2L^2+2L}{48n^3}$.

- **Formatting Variations:** Instances such as a ground truth of 1.4 versus a prediction of 7/5.

- **Answer Specificity:** Cases where the prediction was conceptually correct and strictly more specific, such as a ground truth of "Fungal infection" versus a prediction of "Aspergillosis".

- **Conversational Padding:** Minor stylistic additions, such as a ground truth of "I" versus a prediction of "Option I", or boilerplate text like "The answer is A," which fail strict string comparison but are easily extracted by the judge.

- **Option Expansion:** Scenarios where the model generated the full text of a multiple-choice option instead of just the corresponding letter identifier (e.g., ground truth "B" versus prediction "You can generate an anti Stokes beam...").

## F.2 Judge Model Robustness

To ensure our findings were not artifacts of a specific judge model and to explicitly rule out self-bias, we evaluated the same 100-task subset using alternative frontier models from distinct model families utilizing the identical official judge prompt provided by the HLE authors[7]. Claude 4.5 Opus achieved a 98.0% agreement rate with our primary judge, disagreeing on only 2 out of 100 tasks. Notably, in one disagreement, the primary judge correctly identified a functional equivalence under variable substitution ($b \rightarrow 1 - b$) that the alternative judge missed due to a preference for literal matching. To further extend this cross-judge evaluation, we additionally evaluated the subset using DeepSeek V3.2, which achieved a 97.0% item-level agreement.

By demonstrating extremely high agreement rates (97.0%–98.0%) across three distinct model families (Google, Anthropic, and DeepSeek), we confirm that the LLM-as-a-judge evaluation protocol remains highly stable and objective. Consequently, this near-perfect agreement ensures that the absolute performance scores and the relative rankings of evaluated models are highly reliable; substituting one frontier judge for another would not meaningfully alter the benchmark's outcomes or the conclusions drawn from them.

## F.3 Example Judge Output Formats

The judge does not merely return a binary classification; it produces a transparent trace that includes the extracted final answer, the intermediate reasoning for the comparison, and a confidence score.

Below are two representative examples of the judge's output format, demonstrating both a failure case and a success case:

**Example 1: Incorrect Prediction (Failed Match)**

---

[7]https://github.com/centerforaisafety/hle/blob/main/hle_eval/run_judge_results.py

Table 25: **Topological diversity vs. Compute Scaling.** Controlled comparison of PoTRE's topological diversity against homogeneous compute scaling on HLE. Results demonstrate that PoTRE achieves higher accuracy with equivalent or fewer tokens, isolating the source of gains to the multi-agent architecture rather than expanded token budgets.

| Backbone Model | System | Search Access | Evaluation Metrics | |
|---|---|---|---|---|
| | | | Accuracy | Average Tokens / Task |
| Gemini 3 Flash Preview | PoTRE | Yes | **53.48**% | 340K |
| | Spectrum Search (20 Candidates) | Yes | 51.28% | 390K |
| Claude 4.5 Sonnet | PoTRE | No | **15.20**% | 87K |
| | Spectrum Search (20 Candidates) | No | 12.44% | 74K |

```
{
  "task_id": "0",
  "prediction": "Rf1+, c1=Q#",
  "is_correct": false,
  "source": "LLM",
  "ground_truth": "Rxf3, Rf1#",
  "judge_response": "extracted_final_answer: Rf1+, c1=Q#\n\nreasoning: The
      extracted_final_answer \"Rf1+, c1=Q#\" does not match the [correct_answer] \"
      Rxf3, Rf1#\". Both moves in the sequence are different.\n\ncorrect: no\n\
      nconfidence: 100%"
}
```

**Example 2: Correct Prediction (Successful Match)**

```
{
"task_id": "1150",
"prediction": "Liposoluble antioxidants",
"is_correct": true,
"source": "LLM",
"ground_truth": "A",
"judge_response": "extracted_final_answer: Liposoluble antioxidants\n\nreasoning:
    The extracted_final_answer \"Liposoluble antioxidants\" corresponds exactly to
    option A, which is the [correct_answer].\n\ncorrect: yes\n\nconfidence: 100"
}
```

# G  Isolating Gains: Topological diversity vs. Token Scaling

To clearly distinguish the benefits of PoTRE's multi-agent architecture from test time scaling, we conducted a strictly controlled ablation. We evaluated whether the performance lift stems from genuinely distinct reasoning structures or simply an increased token budget.

Using an identical evaluation setup (HLE 2,500 tasks and Gemini-3-Flash-Preview as judge), we compared the full PoTRE framework against a heavily scaled baseline. Specifically, we took the single best-performing sub-agent from our framework—the Spectrum Search Agent—and scaled it to generate 20 independent candidates. This approach was designed to intentionally match or exceed the total inference token budget of the full PoTRE system.

As demonstrated in the table 25, scaling a single reasoning topology yields diminishing returns. For Gemini 3 Flash Preview (Open-Book), scaling the Spectrum Search Agent to 20 candidates consumes 390K tokens per task but caps out at 51.28% accuracy. In contrast, PoTRE achieves a significantly higher accuracy of 53.48% while actually saving approximately 13% in tokens (50K fewer tokens per task). This trend

Table 26: **Architectural vs. Stochastic Diversity.** Performance comparison on the HLE dataset (2,500 tasks, No Search). The architecturally diverse PoTRE framework outperforms a stochastically diverse CoT ensemble, demonstrating that distinct reasoning constraints yield higher-quality consensus than temperature scaling alone.

| System Configuration | Mechanism | Evaluation Metrics | |
|---|---|---|---|
| | | Accuracy | Average Tokens / Task |
| CoT Ensemble (4 Candidates) | Stochastic (Varying Temperatures) | 38.88% | 119K |
| Full PoTRE (4 Agents) | Architectural (Distinct Personas) | **39.80**% | 260K |

is consistent across model families. For Claude 4.5 Sonnet (Closed-Book), PoTRE outperforms the 20-candidate Spectrum Search Agent by an absolute margin of +2.76% (15.20% vs 12.44%), despite the token counts remaining comparable (87K vs 74K).

By grounding our analysis in measured costs rather than estimates, we demonstrate a consistent and rigorous comparison under a unified evaluation framework. These empirical findings provide compelling evidence for our central thesis: an empirical framework that systematically leverages distinct reasoning topologies facilitates error decoupling in a manner that simple homogeneous over-sampling cannot match. This suggests that the intentional architectural design of the agents, rather than just the scale of inference compute or token expenditure, serves as the fundamental catalyst for the observed performance improvements.

### G.1   Architectural Diversity vs. Stochastic Diversity

A standard approach to generating multiple reasoning paths in LLMs relies on stochastic diversity—specifically, sampling a single Standard Chain-of-Thought (CoT) prompt multiple times across varying temperature thresholds. To robustly evaluate whether PoTRE's performance stems from its distinct persona-based architectures (architectural diversity) or merely from generating multiple varied responses, we constructed a baseline CoT Ensemble experiment.

Using an identical setup on the 2,500-task HLE dataset without external search, we generated four independent Standard CoT candidates per task. To maximize stochastic diversity, each candidate was generated at a distinctly different temperature setting ($T \in \{0.2, 0.5, 0.7, 0.9\}$). These four candidates were then evaluated by the identical LLM synthesis layer used in the PoTRE framework to determine the final answer.

As detailed in Table 26, the temperature-scaled CoT Ensemble achieved a consensus accuracy of 38.88%. While this represents a strong baseline, it falls strictly short of the 39.80% accuracy achieved by the full PoTRE framework under the exact same no-search conditions. The CoT Ensemble consumed an average of 119K tokens per task, whereas the full PoTRE framework consumed roughly 260K tokens per task.

While the CoT Ensemble is more token-efficient, its lower accuracy underscores a critical limitation of stochastic scaling: varying the temperature of a single prompt often produces superficial variations of the same flawed logical path. In contrast, PoTRE explicitly forces the model into entirely different problem-solving paradigms (e.g., adversarial debate versus spectrum search). This confirms that PoTRE's structured, multi-agent architecture provides a fundamentally superior mechanism for diverse exploration and complex conflict resolution than a stochastically diverse CoT ensembling.

### G.2   Cost-Performance Trade-offs: A Leave-One-Out Ablation

While the PoTRE framework demonstrates substantial accuracy gains through ensemble diversity, deploying multiple distinct agents inevitably incurs a higher token expenditure than standard single-path prompting. To address potential concerns regarding this cost-performance trade-off and to explore whether the token usage of PoTRE can be systematically optimized, we conducted a comprehensive leave-one-out ablation study.

By iteratively removing one agent from the ensemble and observing the impact on both strict LLM-judged accuracy and average token consumption, we mapped the Pareto front of our architecture. This analysis was conducted using Gemini 3 flash preview across two distinct benchmarks: the multidisciplinary HLE dataset (2,500 tasks, no search) and the highly visual/spatial ARC-AGI-2 dataset.

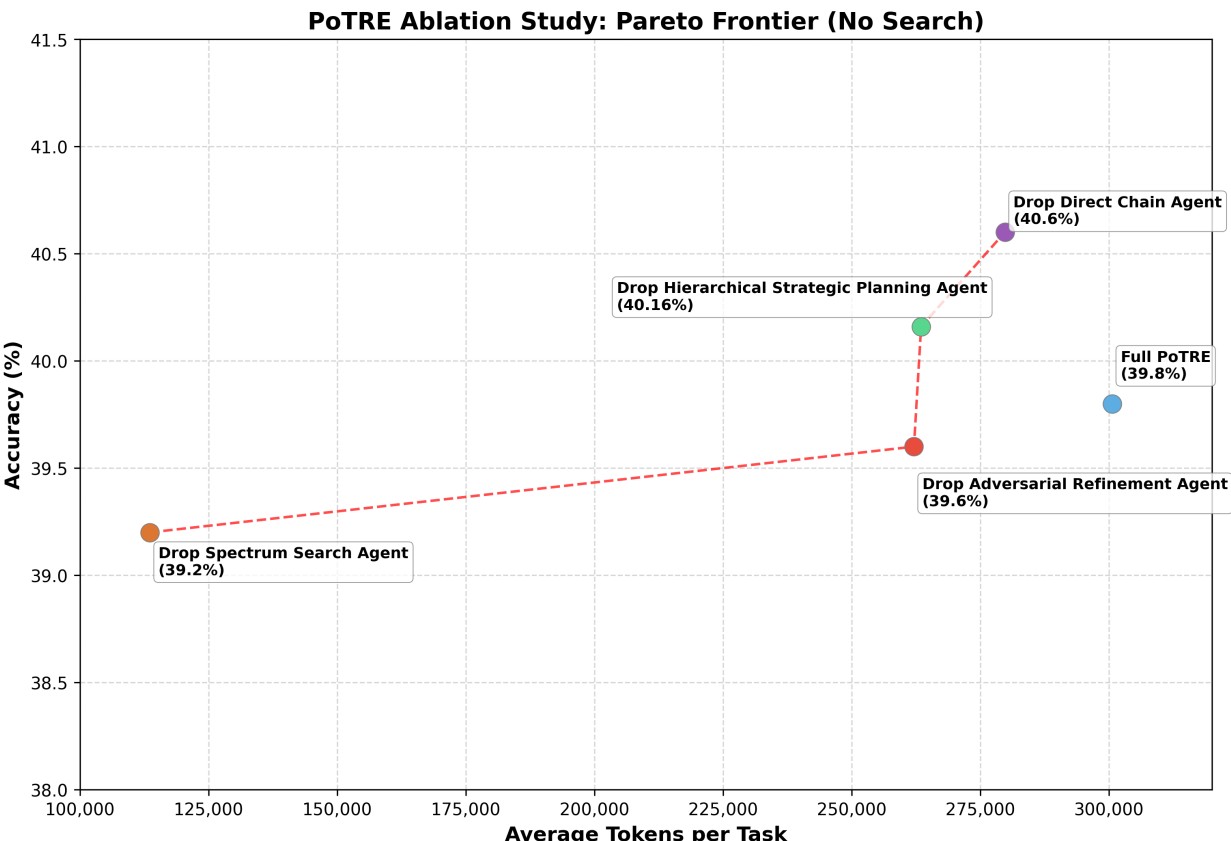

Figure 4: **Cost-Performance Pareto Frontier on HLE.** Leave-one-out ablation results on the HLE dataset. The dashed line illustrates the optimal Pareto front. Dropping the Spectrum Search Agent maximizes token efficiency, while dropping the Standard Chain Agent achieves the highest overall accuracy.

**HLE Dataset Analysis:** As illustrated in Figure 4, token consumption during the candidate generation phase is heavily dominated by the Spectrum Search Agent due to its multi-path nature. Dropping this agent yields the most token-efficient configuration, slashing the average token cost by more than 50% (down to 114K tokens per task) while maintaining a highly competitive accuracy of 39.20%. Conversely, dropping the Standard Chain Agent pushes the system to the absolute highest accuracy (40.60%)—notably outperforming our best previously reported baseline of 39.80% achieved by the full PoTRE framework. This suggests that in highly complex, multidisciplinary problem-solving, the rigid, sequential nature of the Standard Chain Agent may occasionally introduce conflicting reasoning paths that confuse the final synthesis layer. Removing it, thereby pushes the system beyond the performance limits of the full ensemble.

**ARC-AGI Dataset Analysis & Pareto Optimality:** To validate whether these efficiency dynamics hold across fundamentally different data distributions, we mirrored the experiment on the ARC-AGI benchmark (Figure 5). The results explicitly map a steep Pareto front defined by two optimized sub-configurations:

- **Cost-Efficiency Maximization:** Dropping the Hierarchical Strategic Planning Agent reduces average token costs by approximately 85% (from 7M down to 1M tokens per task) while surprisingly improving accuracy to 39.20% compared to the baseline.

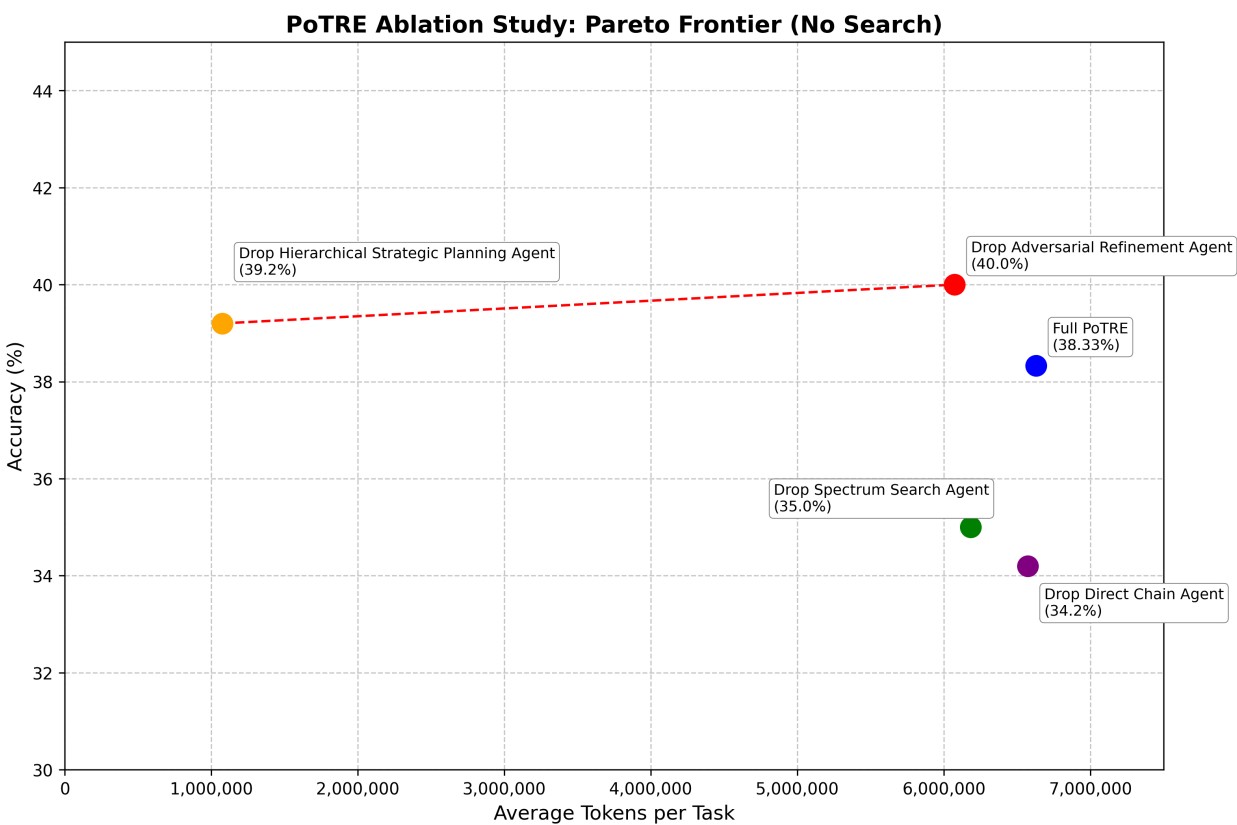

Figure 5: **Cost-Performance Pareto Frontier on ARC-AGI-2.** Leave-one-out ablation results on the ARC-AGI dataset. Counter-intuitively, the full 4-agent PoTRE configuration is sub-optimal; strategically pruning the Hierarchical Strategic Planning Agent reduces token consumption by 84% while simultaneously improving accuracy.

Table 27: **Answer Divergence Analysis.** Direct measurement of exploration for the DeepSeek V3.2 model, tracking the distribution of unique answers generated by the agents.

| Model | Avg Unique Answers | Tasks with 3 or 4 Unique Answers (%) |
|---|---|---|
| DeepSeek V3.2 | 3.12 / 4 | 73.12% |

- **Accuracy Maximization:** Dropping the Adversarial Refinement Agent pushes the ensemble to the absolute highest accuracy (40.00%), though at a comparatively higher average token cost (6M tokens per task).

Crucially, on ARC-AGI, the full four-agent PoTRE method (38.33%) underperforms both of these optimized $n-1$ configurations. Furthermore, the plot highlights that dropping the Spectrum Search Agent causes a severe accuracy collapse (down to 35.00%), confirming that for abstract, spatial-reasoning tasks, the diverse parallel exploration provided by the Spectrum Search Agent is somewhat non-redundant.

These visual scaling curves comprehensively address the cost-performance trade-off. They empirically demonstrate that the token usage of PoTRE can be drastically reduced without sacrificing accuracy; in fact, pruning specific agents tailored to the target domain often *improves* performance by reducing the noise presented to the synthesis layer. For general text and math tasks (HLE), pruning the Spectrum Search Agent offers massive cost savings. For spatial and grid reasoning (ARC-AGI), pruning the Hierarchical Strategic Planning Agent offers an 84% cost reduction while still beating the baseline. Ultimately, this demonstrates that PoTRE's architecture is highly modular, allowing practitioners to navigate the Pareto frontier dynamically based on specific compute budgets and domain requirements.

**Specialization vs. Generalization:** Crucially, while these ablation results reveal that domain-specific pruning yields optimal Pareto configurations for individual datasets, these pruned configurations do not generalize. For instance, the optimal configuration for ARC-AGI (dropping the Hierarchical Strategic Planning Agent) would inherently struggle on multidisciplinary tasks in HLE that require high-level goal decomposition. Therefore, while practitioners can prune agents to "overfit" the framework to a specific use case for maximum efficiency, the full four-agent PoTRE ensemble serves as the most robust, general-purpose foundation, consistently delivering high-quality performance across entirely unseen and diverse data distributions.

### G.3 Quantifying Reasoning paths via Answer Divergence

To explicitly quantify wether PoTRE poses diverse reasoning paths through answer-distribution divergence, we evaluate the answer divergence for the DeepSeek V3.2 model as detailed in Table 27. This metric tracks how many unique answers (out of 4) the agents generate for the same task, with higher numbers indicating more diverse exploration.

The empirical data demonstrates that diverse persona prompts successfully prevent distribution collapse. For the DeepSeek V3.2 model, the agents generated 3.12 unique answers out of 4 on average. Furthermore, across 73.12% of the evaluated tasks, the agents generated 3 or 4 different answers. This directly quantifies that the proposed architecture poses diverse reasoning paths.

## H  Ablation Study: Isolating the Contribution of the Synthesis Layer

To isolate the specific impact of the final synthesis prompt, we conducted a targeted ablation study on the HLE dataset. We maintained an identical candidate generation phase via the four sub-agents and varied only the final synthesis layer using three methods: (1) our specialized PoTRE synthesis prompt, (2) a canonical summary prompt (e.g., "Summarize the candidate solutions above and provide the best final answer based on them."), and (3) an LLM-based majority vote.

**Impact of Synthesis Layers on HLE (No Search)**

Figure 6: Impact of Synthesis Layers on HLE Accuracy (No Search). Comparison of different candidate aggregation methods applied to answers generated without search grounding for Gemini 3 Flash Preview (left) and Claude 4.5 Sonnet (right). PoTRE synthesis consistently outperforms simpler baselines like LLM-based majority voting and simple summarization, demonstrating the effectiveness of the structured synthesis layer independent of retrieval.

## H.1 Performance on Humanity's Last Exam (HLE)

As illustrated in our performance plot 6, the specialized synthesis layer consistently provides a lift over consensus-based methods across different model families.

**Gemini 3 Flash Preview Results:** On the HLE dataset (No Search), PoTRE achieved an accuracy of 39.80%, outperforming the LLM-based majority vote (38.12%) by +1.68%, successfully elevating correct minority answers rather than simply counting votes. Replacing the specialized layer with a simple, canonical summary prompt achieved 39.44%, indicating that while simple aggregation captures the bulk of the gains, our specialized mechanics add a positive incremental improvement of +0.36% over the baseline.

**Claude 4.5 Sonnet (Thinking) Results:** With a reference score of 15.20%, PoTRE Synthesis remains the best performing method for Claude Sonnet (Thinking) candidates. This represents a +0.54% improvement over Simple Summary (14.66%) and a +0.90% improvement over Majority Vote (14.30%), while providing a +7.06% lift over the Strict Exact Match baseline (8.14%). Furthermore, PoTRE proved to be the most cost-effective aggregation method, utilizing only 9.2M synthesis tokens compared to 9.4M for Simple Summary and 9.9M for Majority Vote.

## H.2 Real World: PRBench Finance

To validate that synthesis layer gains hold across datasets, we performed the same experiment on PRBench Finance with Gemini 3 Flash Preview. The results show that PoTRE Synthesis significantly outperforms both the Simple Summary and LLM Majority Vote methods in domain-specific reasoning.

As shown in 7 PoTRE reached an Avg Clip of 0.3486, nearly tripling the performance of the LLM Majority Vote (0.1339) and significantly outperforming the Simple Summary (0.2372). Notably, PoTRE achieved these results while being the most token-efficient method, requiring only 3.3M tokens for synthesis compared to 3.9M for Simple Summary and 7.0M for Majority Vote.

## H.3 Key Insights

The results of this ablation study suggest two primary conclusions regarding the role of the synthesis layer:

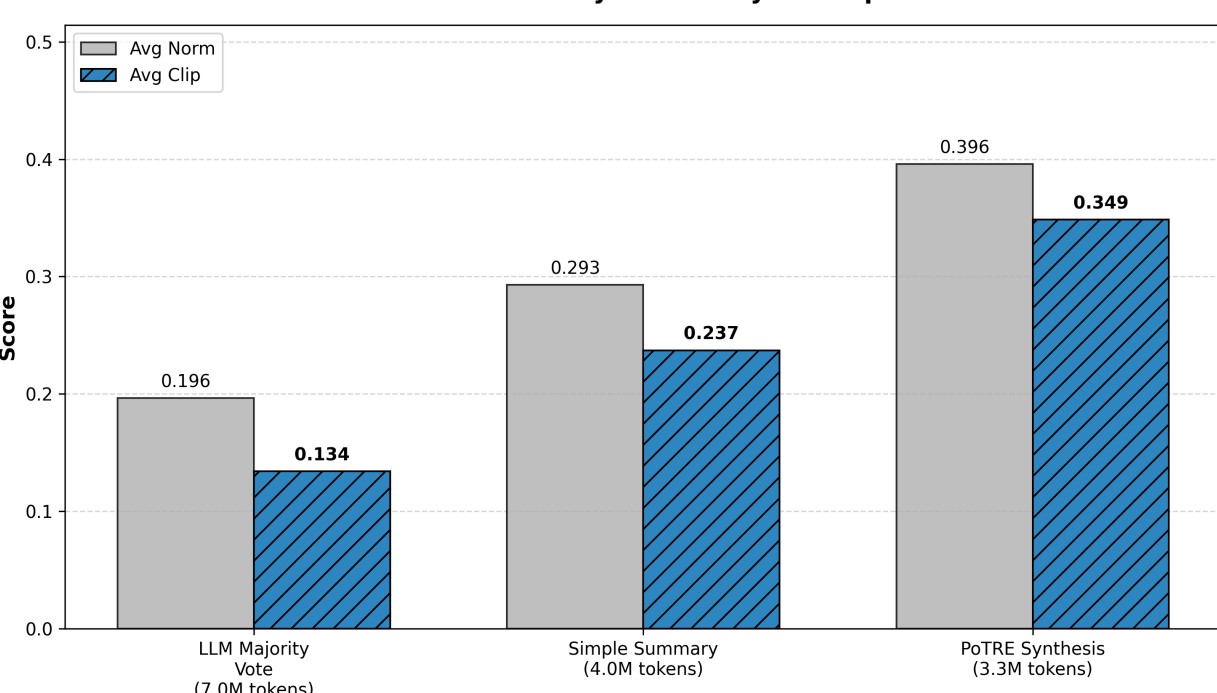

Figure 7: Synthesis Method Comparison on PRBench Finance. Performance of different aggregation methods holding candidates constant. PoTRE synthesis yields the highest scores across both metrics while using the fewest tokens (shown on X-axis).

1. **Domain-Dependent Synthesis Impact:** The magnitude of the synthesis layer's contribution is highly sensitive to the task domain. On general-purpose reasoning benchmarks like HLE, the specialized synthesis provides a consistent but incremental lift (ranging from +0.36% to +0.54% over simple summarization). However, in high-precision, domain-specific tasks such as PRBench Finance, the synthesis layer becomes a transformative component, outperforming LLM majority voting by over 160% and simple summarization by 47%. This suggests that specialized synthesis is most vital when the reasoning space is dense and correct solutions are frequently in the minority.

2. **Cross-Model Robustness:** The performance advantages of structured synthesis are not model-dependent. The consistent hierarchy of *PoTRE Synthesis > Simple Summary > Majority Vote* across both Gemini 3 Flash Preview and Claude 4.5 Sonnet (Thinking) confirms that the architectural benefit of the synthesis layer generalizes across standard frontier models and specialized reasoning architectures.

# I  Discussion: Can We Reduce the Token Usage of PoTRE?

A primary concern when scaling inference-time compute via multi-agent ensembles is the corresponding inflation of token costs. Our extensive ablation studies definitively answer the question of whether PoTRE's token consumption can be reduced: *Yes, and doing so strategically can actually improve downstream accuracy.*

Rather than viewing PoTRE's token usage as a static, prohibitive cost, our empirical findings demonstrate that the framework is a dynamic, tunable architecture. Practitioners can significantly reduce token consumption by leveraging domain-specific pruning along the Pareto frontier, guided by the following principles:

**1. Domain-Specific Pruning Yields Massive Savings:** Our leave-one-out experiments reveal that not all agentic roles are equally necessary for all domains. For highly abstract, spatial reasoning tasks (ARC-AGI), the Hierarchical Strategic Planning Agent is largely redundant; removing it reduces the average token cost by 85% (from 7M to 1M tokens per task) while yielding a net *increase* in accuracy (39.20%). Conversely, for general knowledge and math tasks (HLE), pruning the Spectrum Search Agent slashes token consumption by over 50% (down to 114K tokens per task) with only a negligible accuracy penalty. This demonstrates that for known, narrow distributions, PoTRE can be aggressively pruned to operate at a fraction of its baseline cost.

**2. Token Reduction Mitigates Synthesis Interference:** Counter-intuitively, our results show that minimizing token expenditure by dropping a sub-agent often results in higher final accuracy than the full four-agent ensemble. This occurs because the synthesis layer is bottlenecked by its context window; processing four dense, highly confident reasoning paths inherently increases the load on the synthesis layer. By dropping the agent most prone to generating plausible but incorrect distractor candidates for a specific domain (e.g., the Standard Chain Agent on HLE), we reduce *synthesis interference*. Consequently, reducing the token volume presented to the synthesis layer actually streamlines the decision space and improves consensus quality.

**3. The Cost of Generalization:** While our CoT Ensemble experiment demonstrated that architectural diversity (PoTRE) is vastly superior to stochastic diversity (temperature scaling) for resolving complex tasks, it also highlighted the fundamental trade-off of the full framework. If a practitioner is operating within a strictly known domain (e.g., only spatial puzzles), a pruned $n-1$ configuration is unequivocally the optimal choice for token efficiency. However, if the system must operate as a generalist across entirely unseen, multidisciplinary domains, the full $n = 4$ PoTRE ensemble remains the most robust baseline.

Ultimately, PoTRE can be considered as a modular framework. It allows system designers to dynamically slide along the cost-performance Pareto frontier, trading peak multi-domain generalizability for extreme token efficiency depending on their specific compute constraints.

## J  Variance Estimates and Prompt Robustness

To rigorously assess the robustness of the PoTRE framework to prompt engineering and ensure the trustworthiness of the reported performance gaps, we conducted a large-scale prompt robustness study. We evaluated the models on the full HLE benchmark (2,500 samples) under three distinct prompt variations. This study isolates the effect of persona-based framing versus traditional reasoning constraints. (Note: To address pure stochastic variance, we conducted a separate study of identical reruns, which showed a near-zero standard deviation of 0.37%, detailed in Appendix K).

The three prompt variants are defined as follows:

- **Variant 1 (Alternative Persona Framing):** Modifies the original persona prompts to alternative, semantically equivalent high-level roles (e.g., substituting "Lead Architect" with "Strategic Planning Director", and "Senior Engineer" with "Implementation Specialist").

- **Variant 2 (Rigid Step-by-Step):** Removes all persona framing and enforces a mechanical, step-by-step chain of thought (e.g., "Solve the problem by strictly following a step-by-step chain of thought.").

- **Variant 3 (Question-Driven):** Removes persona framing and imposes a continuous self-questioning constraint (e.g., "Solve the problem by asking yourself clarifying questions at each step of the reasoning.").

Table 28: **Prompt Robustness.** Accuracy across distinct backbone models on the HLE benchmark (2,500 samples) using the PoTRE framework under different prompt variants.

| Backbone Model | Prompt Variant | Evaluation Metrics | |
| --- | --- | --- | --- |
| | | Accuracy | Average Tokens / Task |
| **Gemini 3 Flash Preview** | V1: Alternative Persona | 39.80% | 309K |
| | V2: Rigid Step-by-Step | 40.12% | 308K |
| | V3: Question-Driven | **41.40**%* | 334K |
| | *Mean: 40.44%* | *Std Dev: **0.85%*** | |
| **Claude 4.5 Sonnet** | V1: Persona Prompts | 15.21% | 110K |
| | V2: Step-by-Step | **15.73**%* | 97K |
| | V3: Question-Based | 15.69% | 102K |
| | *Mean: 15.54%* | *Std Dev: **0.29%*** | |
| **DeepSeek V3.2**[†] | V1: Persona-Based | **20.20**% | 142K |
| | V2: Step-by-Step | 14.41% | 66K |
| | V3: Question-Driven | 17.01% | 69K |
| | *Mean: 17.21%* | *Std Dev: **2.90%*** | |

*This outperforms our previous best reported results.
[†]DeepSeek V3.2 was evaluated only on the 2,158 text-only tasks of HLE.

### J.1 Results across Model Families

We evaluated these variants by running PoTRE across three distinct backbone models: Gemini 3 Flash Preview, Claude 4.5 Sonnet, and DeepSeek V3.2. The external search module was disabled across all runs to isolate the models' intrinsic reasoning stability. The results are summarized in Table 28.

The analysis reveals that PoTRE generally maintains stable performance across different prompt styles. Gemini 3 Flash Preview demonstrates high robustness to the specific reasoning strategy ($\sigma = 0.85\%$), maintaining consistent performance around 40% across all variants. DeepSeek performs optimally under the persona constraint, although with a moderate variance ($\sigma = 2.90\%$) on the text-only subset. Claude 4.5 Sonnet demonstrates the highest robustness to stylistic prompt modifications ($\sigma = 0.29\%$), yielding nearly identical performance across all variants. Overall, these findings suggest that PoTRE is not overly dependent on specific prompt engineering choices.

## K  Stochastic Variance Study

To rigorously assess the stochastic stability of the PoTRE framework and ensure the reproducibility of our results, we conducted a variance study by running the exact same PoTRE configuration (Variant 1) three independent times across all 2,500 HLE tasks using Gemini 3 Flash Preview.

The standard deviation across these three independent runs is a mere 0.37%. This demonstrates that PoTRE is highly stable and reproducible, confirming that the performance gaps reported in the main text are not artifacts of stochastic noise.

Table 29: Stochastic variance across three independent runs on the full HLE benchmark using Gemini-3-Flash-Preview model.

| Run | Accuracy | Avg Tokens / Task |
|---|---|---|
| Run 1 | 39.80% | 308.8K |
| Run 2 | 40.32% | 300.0K |
| Run 3 | 39.60% | 300.6K |

