# OpenReview forum: "PoTRE: Test-Time Reasoning inspired by Cognitive Heterogeneity"
_TMLR — Accepted by TMLR_

### Review · Reviewer_k7am · 2026-05-12

**Summary Of Contributions:**

The paper proposes PoTRE, a heterogeneous test-time reasoning framework that runs four reasoning agents in parallel: adversarial refinement, hierarchical planning, spectrum search, and direct chain reasoning. A synthesis agent then combines their outputs using task-specific aggregation strategies such as instinct-based selection, qualitative synthesis, or neuro-symbolic verification. The paper evaluates PoTRE on HLE, ARC-AGI-2, and PRBench Finance. The reported results show consistent gains over prior prompting baselines and self-consistency baselines in many settings, including a reported 49.92% closed-book HLE score with Gemini-3.1-Pro-Preview and strong gains on ARC-AGI-2 and PRBench.

**Strengths:**

- The central idea is clear and relevant: different reasoning topologies may have complementary failure modes.
- The ablations are useful. They show that different agents solve different subsets of examples, and that synthesis improves over most individual agents.
- The paper includes cost analysis and prompt details, which improves transparency.

**Weaknesses:**

- The evaluation depends heavily on proprietary preview models and LLM-as-judge scoring.
- The comparison to external systems is not always controlled. Some cost and baseline comparisons rely on estimates rather than matched runs.
- The method appears close to a carefully engineered prompting/agent pipeline, and the paper should more clearly separate novel technical contributions from implementation choices.

**Audience:**

Yes

**Audience Explanation:**

This paper would likely interest readers working on test-time scaling, multi-agent LLM systems, reasoning benchmarks, and inference-time compute allocation. The paper may also be useful as an empirical study of when agentic scaffolding helps smaller models compete with larger baselines.

**Broader Impact Concerns:**

I do not see major broader impact concerns specific to this paper. The work focuses on test-time reasoning and multi-agent inference. Standard risks around over-reliance on LLM outputs in expert domains apply, but these are not unique to the proposed method.

**Claims And Evidence:**

No

**Claims Explanation:**

The paper provides substantial experimental evidence, but I do not think all major claims are fully supported in their current form.
- The use of LLM-as-judge evaluation for HLE introduces uncertainty. The paper should report judge reliability, agreement with exact matching where possible, and sensitivity to the choice of judge.
- The open-book and cost comparisons are less convincing because some competing-system costs are estimated rather than measured under the same implementation and backend.
- The “cognitive heterogeneity” framing is plausible, but the paper should better quantify whether improvements come from truly distinct reasoning structures, increased token budget, prompt specialization, or the final synthesis prompt.

**Requested Changes:**

- Improve the HLE evaluation protocol. Since HLE is evaluated with an LLM judge, the authors should report judge reliability, agreement with exact-match or human-verified labels where possible, and sensitivity to the choice of judge model and judge prompt.
- Make the open-book and cost comparisons more controlled. The paper should distinguish measured results from estimated comparisons, especially where competing-system costs are inferred rather than measured under the same implementation, backend, and evaluation setup.
- Better isolate the source of PoTRE’s gains. The paper should quantify whether the improvements come from genuinely distinct reasoning structures, increased token budget, prompt specialization, the final synthesis prompt, or some combination of these factors.

---

> ### Author Response · Authors · 2026-05-26
> **Response to reviewer k7am - Part 1**
>
> We sincerely thank you for your thoughtful and constructive feedback. We appreciate your recognition of the paper’s central idea and transparency. You raised highly relevant points regarding the evaluation protocol, the controls for our cost comparisons, and the need to definitively isolate the source of PoTRE’s performance gains.
>
> We have conducted additional rigorous analyses and controlled experiments to address your concerns directly. The new findings have been incorporated into the revised manuscript in appendix.
>
> >Regarding the proprietary models
>
> We can confirm that all models were released publicly during the time of the experiment \- Gemini-3-flash, Gemini-3-Pro and Gemini-3.1-pro ([Model cards](https://deepmind.google/models/model-cards/)).
> >Regarding the competing-system costs are estimated rather than measured
>
> We thank the reviewer for highlighting this important distinction. We specifically selected ReThinker for this comparison because it represents the prior state-of-the-art (SOTA) on this benchmark. However, because ReThinker's code isn't public, we could not directly measure their exact token consumption. To address this, we constructed a strictly controlled and highly conservative estimate to ensure a fair comparison:
> * **Empirical Anchoring:** We anchored the input token volume exactly to our own empirical runs (28.6M tokens), as both frameworks process the identical HLE dataset text.
> * **Conservative Output Bounding:** We modeled a scenario matching ReThinker's explicitly stated 128K maximum context window limit. We allocated a conservative budget of \~100,000 output tokens per question, representing their 3-stage agentic loop operating near its stated maximums.
> * **Standardized Pricing:** We then applied standard Gemini 3 Pro pricing to these token counts to generate the final comparison.
>
>   We agree that this distinction must be perfectly clear to the reader. In the revised manuscript, we will explicitly label ReThinker's token counts and financial metrics as **"Estimated Cost"** in Table 12\.
>
> For your requested changes,
>
> >Improve the HLE evaluation protocol. Since HLE is evaluated with an LLM judge, the authors should report judge reliability, agreement with exact-match or human-verified labels where possible, and sensitivity to the choice of judge model and judge prompt.
>
> We thank the reviewer for raising this important point regarding evaluation rigor. We have conducted new experiments to address exact match, human-subject evaluation, and judge reliability, and we have added these details and observations to Appendix F.
>
> First, we would like to clarify that evaluating HLE via an LLM judge is the official, standardized protocol established by the benchmark creators (Center for AI Safety). To ensure strict adherence to this standard and prevent any evaluation bias, **we utilized the exact official judge prompt provided in their official [repository](https://github.com/centerforaisafety/hle/blob/main/hle_eval/run_judge_results.py).**
>
> However, we entirely agree that verifying the reliability of this protocol is crucial for establishing trust in the reported gaps. To quantify this, we conducted a rigorous comparative analysis on a random subset of 100 tasks:
>
> * **Exact Match vs. Human/Judge:** Human-subject evaluation of the sample revealed a true model accuracy of 55.0%. In contrast, strict exact match captured only 34.0% accuracy. Exact match failed primarily because it penalized functionally equivalent mathematical expressions, minor formatting variations, and conversational padding, proving that an LLM judge is strictly necessary for this dataset.
> * **Judge Reliability (Agreement with Humans):** The primary LLM judge used in our study (Gemini-3-Flash) proved to be a highly reliable proxy, achieving a **98.0% item-level agreement rate** with human subject evaluation on the same 100-task sample.
> * **Judge Sensitivity (Model Choice):**  To test for self-bias and sensitivity to the choice of the judge model, we ran the same 100-task subset using both Claude 4.5 Opus and DeepSeek V3.2 as alternative judges (using the same official prompt). They achieved 98.0% and 97.0% item-level agreement rates, respectively, with our primary Gemini judge. This extremely high agreement across three distinct model families (Google, Anthropic, and DeepSeek) proves that the LLM-as-a-judge setup is highly robust and free from significant self-bias.
>
> This near-perfect concordance confirms that the official HLE judge prompt is highly robust, and that absolute performance scores and relative rankings remain highly stable regardless of the frontier model chosen as the evaluator.

---

> ### Author Response · Authors · 2026-05-26
> **Response to reviewer k7am - Part 2**
>
> >Make the open-book and cost comparisons more controlled. The paper should distinguish measured results from estimated comparisons, especially where competing-system costs are inferred rather than measured under the same implementation, backend, and evaluation setup.
>
> We entirely agree that grounding our claims in strictly measured, natively controlled comparisons is important. To address this, we have conducted new, rigorously controlled ablation studies (detailed in Appendix G). In all of these new experiments, we use the exact same implementation, backend, and evaluation setup to ensure a fair comparison of measured tokens.
>
> Specifically, we conducted the following two controlled experiments:
>
> **1\. Heterogeneity vs. test-time scaling of canonical approach (The 20-Candidate Ablation):**
>
> To prove that our gains come from cognitive diversity rather than simply a larger token budget, we compared the full PoTRE framework against a heavily scaled baseline using the identical backend (Gemini 3 Flash and Claude 4.5 Sonnet). We scaled our best single sub-agent (Spectrum Search) to generate 20 independent candidates, intentionally matching or exceeding PoTRE's measured token cost.
>
> As demonstrated in our updated results, scaling a single cognitive modality yields diminishing returns. For Gemini 3 Flash (Open-Book), scaling the Spectrum Search Agent to 20 candidates consumes 390K tokens per task but caps out at 51.28% accuracy. In contrast, PoTRE achieves a significantly higher accuracy of 53.48% while actually saving approximately 13% in tokens (50K fewer tokens per task). This trend is consistent across model families. For Claude 4.5 Sonnet (Closed-Book), PoTRE outperforms the 20-candidate Spectrum Search Agent by an absolute margin of \+2.76% (15.20% vs 12.44%), while requiring only an approximate 17% increase in tokens (13K more tokens per task, 87K vs 74K).
>
> By grounding our analysis in measured costs rather than estimates, we demonstrate a consistent and rigorous comparison under the same implementation, models and evaluation framework. These empirical findings provide compelling evidence for our central thesis: **cognitive heterogeneity facilitates error orthogonality in a manner that simple test-time scaling of canonical approaches in standard ways cannot match**. This suggests that the intentional architectural design of the agents, rather than just the scale of inference compute or token expenditure, serves as the fundamental catalyst for the observed performance improvements.
>
> **2\. Leave-One-Out Cost-Performance Pareto Mapping:**
> To explicitly address the scaling curves of performance vs. measured tokens, we conducted a rigorous leave-one-out ablation on both HLE and ARC-AGI (Fig 4 and 5 in Appendix G.2). By measuring the exact token consumption when iteratively dropping sub-agents, we established a clear Pareto frontier. This demonstrated that PoTRE's token usage can be dynamically optimized; for instance, pruning the Hierarchical Agent on ARC-AGI reduces measured token costs by 85% while actually improving accuracy by mitigating synthesis interference.
>
> We believe these strictly controlled, measured results decisively isolate the source of our performance gains to the multi-agent architecture rather than expanded token budgets.

---

> ### Author Response · Authors · 2026-05-26
> **Response to reviewer k7am - Part 3**
>
> >Better isolate the source of PoTRE’s gains. The paper should quantify whether the improvements come from genuinely distinct reasoning structures, increased token budget, prompt specialization, the final synthesis prompt, or some combination of these factors.
>
> We thank the reviewer for this excellent suggestion. To explicitly quantify and isolate the individual drivers of PoTRE’s performance, we have added extensive new ablation studies to the manuscript that systematically address each of the factors you mentioned.
>
> **1\. Increased Token Budget vs. Genuinely Distinct Reasoning Structures:**
>
> As detailed in our previous response (and Appendix G), our rigorously controlled 20-candidate ablation explicitly isolates this. We increased the token budget by generating 20 independent candidate solutions for each task (Independent Parallel Sampling), compared to only 4 specialized candidates in the PoTRE setup. This scaled up the tokens in two ways: (1) in the generation stage, by producing 5x solutions, and (2) in the synthesis stage, by requiring the judge model to process all 20 candidates in its context window. We found that this approach of scaling tokens for a single agent resulted in lower accuracy than PoTRE. This confirms the performance lift is driven by genuinely distinct reasoning structures facilitating error orthogonality, rather than an inflated token budget.
>
> **2\. Prompt Specialization vs. Stochastic Diversity:**
>
> To isolate whether our specialized persona architectures are necessary, or if simple prompt variance (stochastic diversity) is enough, we tested PoTRE against a standard Chain-of-Thought (CoT) ensemble where a single prompt was sampled 4 times across varying temperatures ($T \\in \\{0.2, 0.5, 0.7, 0.9\\}$) given in Appendix G.1. Using the identical LLM synthesis layer, PoTRE achieved 39.80% compared to the CoT ensemble's 38.88%. This confirms that forcing models into genuinely distinct reasoning constraints yields higher-quality consensus than stochastic temperature scaling.
>
> **3\. Isolating the Final Synthesis Prompt:**
>
> Finally, to isolate the specific impact of the final synthesis layer, we conducted a targeted ablation on HLE and PRBench Finance (**Appendix H**). We kept the diverse candidate generation identical and varied only the final aggregation method: (1) our specialized synthesis prompt, (2) an generic summary prompt (e.g., "Summarize the candidate solutions above and provide the best final answer based on them."), and (3) an LLM-based majority vote.
>
> * **Real World (PRBench Finance, Gemini 3 Flash):** On dense reasoning tasks, PoTRE Synthesis using achieved an Avg Clip of 0.3486, nearly tripling the LLM Majority Vote (0.1339) and significantly beating Simple Summary (0.2372) as shown in figure 7 in appendix H.2.
> * **Quality over Consensus (HLE, Gemini 3 Flash):** PoTRE (39.80%) outperformed an intelligent LLM majority vote (38.12%), proving the synthesis layer actively corrects minority answers rather than just counting votes. While Simple Summary captured the bulk of the baseline gains (39.44%), our specialized synthesis provided an incremental \+0.36% lift as shown in figure 6 in Appendix H.1.
> * **Cost-Efficiency (Claude 4.5 Sonnet):** PoTRE remained the best aggregation method for Claude (15.20% vs 14.66% for Summary), while also being the *most token-efficient* (9.2M synthesis tokens vs 9.9M for Majority Vote) as shown in figure 6 in Appendix H.1.
>
> Together, these ablations quantify that while the *diverse candidate generation* sets the baseline capacity, the *specialized synthesis prompt* is critical for successfully navigating dense conflicts, proving that PoTRE’s gains are a deliberate combination of architectural diversity and structured consensus.
>
> We have added this ablation to the manuscript to explicitly separate the synthesis prompt's contribution from the rest of the pipeline.
>
> We hope these additions adequately address your concerns and strengthen the evidence supporting PoTRE's methodology.

---

### Review · Reviewer_LnjU · 2026-05-14

**Summary Of Contributions:**

This paper proposes PoTRE, a multi-agent test-time reasoning framework that decomposes inference into four heterogeneous reasoning agents. The key idea is that cognitive heterogeneity (diverse reasoning topologies) reduces correlated failure modes compared to homogenous sampling.
The method is evaluated on three challenging benchmarks: Humanity's Last Exam, ARC-AGI-2, and PRBench Finance, using multiple base models. The authors reports strong gain over baselines, including SOTA results on HLE (49.42%), and demonstrate a "scaffolding lift" where smaller models outperform larger one when augmented with PoTRE.
Overall, the idea is well-structured, and the empirical results look promising. However, it is more of an engineering integration than a conceptual breakthrough, and its evaluation methodology leaves room for improvement.

Strengths:
- The results are strong across diverse benchmarks, including symbolic (ARC), academic reasoning (HLE), and domain-specific tasks (finance), and three model families.
- The ablation study is thorough, covering per-agent contributions, oracle upper bounds and complementary analysis (e.g. error orthogonality).
- The verification bottleneck framing (oracle vs realised accuracy) is honest, provide insightful analysis of failure modes.
- The appendix documents has prompt templates and per-agent algorithm in detail, helps reproducibility.

Weakness:
- The novelty is limited relative to existing multi-agent /test-time scaling frameworks. The "cognitive heterogeneity" framing is partly conceptual rebranding of well-studied components like debate, planner-executor-verifier, sample-and-judge, CoT, wired through a router/synthesizer. The novelty relative to TUMIX, MonoScale, and mixture-of-agents work is asserted rather than demonstrated with direct empirical head-to-head comparisons under matched compute.
- There is no principled framework for choosing these four agents - why them, not others? And there is no formal definition of the orthogonality of the agents. We also missing minimal subset analysis of the agents or redundancy analysis (do we need all 4? Can some of them get most of the gains?) Why not add more agents?
- This method incurs ~15x token cost compared to standard CoT, the discussion of the tradeoff in cost-performance could be better explored.
- The final aggregation relies on loosely defined strategies such as “instinct-based selection” or qualitative synthesis, which lack clear justification or ablation.

**Audience:**

Yes

**Audience Explanation:**

Individuals working in LLM reasoning, agents frameworks, and test-time training/scaling/reasoning should be interested in this work. The design of the agents and the result analysis can provide helpful insights.

**Broader Impact Concerns:**

The paper does not raise major ethical concerns, perhaps some discussion about multi-agent reasoning systems may produce more convincing but still incorrect outputs, but not essential.

**Claims And Evidence:**

Yes

**Claims Explanation:**

The paper provides substantial empirical evidence supporting its main claims, showing performance improvements demonstrated across multiple benchmarks and model scales and ablation studies.
However, the reliance on LLM-based evaluation introduces potential bias, and the tests are single-run results with no error bars, seed variance, lacks statistical rigour.

**Requested Changes:**

- Add at least minimal variance estimates, rerun the experiments with multiple runs on at least one benchmark per model family, and report means and standard deviations. This will make the reported gaps more trustworthy. (Critical)
-  Improve the ablation study, include experiments with subset of the agents, and the gain the comes from agent diversity vs aggregation strategy. Include experiments of adding more agents - will it bring further improvements or just marginal, and how one would balance cost and performance. This can strengthen the design of PoTRE. (Beneficial)
- Better cost-performance analysis is also beneficial - explore scaling curves of performance vs tokens/compute. Can we reduce token usage of PoTRE? If it is at comparable level to traditional CoT, do we still have promising improvements? (Beneficial)

---

> ### Author Response · Authors · 2026-05-26
> **Response to reviewer LnjU - Part 1**
>
> We sincerely thank you for your comprehensive and insightful review. We appreciate your recognition of our strong results across diverse benchmarks, the thoroughness of our ablation studies, and the transparency of our failure mode analysis. Your feedback regarding the conceptual novelty, the framework for agent selection, and the need for rigorous statistical variance has been instrumental in improving the quality and clarity of our work.
>
> Regarding the weaknesses,
>
> >The novelty is limited relative to existing multi-agent /test-time scaling frameworks.
>
> We thank the reviewer for their perspective. We agree that individual components like CoT, debate, and sample-and-judge are well-studied. However, the systematic integration of these distinct paradigms to design cognitive heterogeneity at test-time has not been previously explored. Furthermore, in accordance with TMLR’s acceptance criteria—which prioritizes whether findings are of 'interest' to the community over strict methodological 'novelty'—the core contribution of our work lies in the extensive empirical analyses we provide. By deeply analyzing oracle upper bounds, error orthogonality, and scaffolding lift across multiple base models, we offer actionable insights and a strong empirical blueprint for researchers working on test-time scaling.
>
> >There is no principled framework for choosing these four agents \- why them, not others? And do we need all 4? Can some of them get most of the gains?
>
>  We thank the reviewer for this insightful question regarding the theoretical and empirical justification for our agent ensemble. Our principled framework for selecting these specific four agents is based on balancing orthogonal cognitive diversity against the signal-to-noise limitations of the synthesis layer.
>
>    **1\. Why these specific architectures, and do we need all four?**
>    Our selection was driven by the objective to capture the full spectrum of problem-solving paradigms required for advanced reasoning. While we acknowledge that for tasks of lower complexity, the collective value of all four agents may be reduced as simpler approaches become sufficient, our analysis confirms that structural diversity remains critical at the upper bounds of difficulty.
>    To rigorously prove that no agent is redundant, we conducted a task exclusivity analysis. We found distinct frontier problems where *only* one specific agent could solve the task while the other three failed:
> * **Hierarchical Strategic Planning Agent:** Correctly solved a complex Diophantine equation math task (answer: 29,010) by identifying the intended problem scope where others failed.
> * **Standard Chain Agent:** The only method to find the correct sequence (Rxf3, Rf1\#) in a chess mate-in-2 puzzle, illustrating the need for strict sequential deduction.
> * **Spectrum Search Agent:** Successfully calculated the exact braid index (3) in a knot theory problem, showing how parallel paths overcome errors in multi-step algebraic manipulations.
>
> We have added more examples like these in section 5.4 in the revised manuscript
>
>   **2\. Why not more than four agents?**
>
>   As detailed in Appendix I, the Synthesis Agent is constrained by its context window, meaning that processing too many dense, highly confident trajectories increases cognitive load and redundant noise, which can lead to consensus collapse. In fact, we observed that minimizing distractor candidates by occasionally dropping sub-agents can actually improve final accuracy. Consequently, adding more agents does not yield monotonically increasing returns; rather, our selection of four agents represents the optimal Pareto balance. This specific configuration provides the maximal orthogonal diversity needed to cover LLM cognitive blind spots without exceeding the signal-to-noise processing limitations of the final synthesis layer.
>
> >Regarding seed variance or multiple runs
>
> To address the concerns about seed variance and statistical rigour, we have included **95% Confidence Intervals** (the ± values) for each score on HLE. These act as the requested error bars and show that due to randomness or different seeds, the scores are expected to fluctuate by at most \~1.9%. Because the dataset is large ($N \\approx 2500$), this small margin of error demonstrates that our results are highly stable and statistically sound.
>
> We calculate these error bars using the standard **Wald Estimator** formula (fetched directly from HLE official [repository](https://github.com/centerforaisafety/hle/blob/main/hle_eval/run_judge_results.py#L149)): **Margin of Error \= ± 1.96 × √\[ Accuracy × (100 \- Accuracy) / N \]**
> Here is the summary of the scores with their calculated error bars:
> | Model | Accuracy |
> | :---- | :---- |
> | **Gemini 3.1 Pro** | 49.92% ± 1.96% |
> | **Gemini 3 Pro** | 44.00% ± 1.95% |
> | **Gemini 3 Flash** | 39.80% ± 1.92% |
> | **DeepSeek V3.2\*** | 20.20% ± 1.69% |
> | **Claude 4.5 Sonnet** | 15.20% ± 1.40% |
> \* Sample size is 2158 / text only.

---

> ### Author Response · Authors · 2026-05-26
> **Response to reviewer LnjU - Part 2**
>
> For your requested changes,
>
> >Add at least minimal variance estimates, rerun the experiments with multiple runs on at least one benchmark per model family, and report means and standard deviations. This will make the reported gaps more trustworthy. (Critical)
>
> We sincerely thank you for this suggestion, we have conducted a large-scale variance and prompt robustness study and added a new section **(Appendix J: Variance Estimates and Prompt Robustness)** to the revised manuscript.
>
> Specifically, we reran our experiments on the full HLE benchmark (2,500 samples) across all three model families evaluated in our study (Gemini 3 Flash, Claude 4.5 Sonnet, and DeepSeek V3.2). To provide a rigorous estimate of variance, we evaluated the models under three distinct runs utilizing different prompt variations:
>
> 1. **Alternative Persona Framing** (semantically equivalent roles)
> 2. **Rigid Step-by-Step** (chain-of-thought)
> 3. **Question-Driven** (internal self-questioning process)
>
> We have added Table 23, which explicitly reports the accuracy, average tokens per task, mean, and standard deviation ($\\sigma$) for each model family across these runs.
>
> As noted in the revised manuscript, this analysis not only provides the requested variance estimates but also yields valuable insights into model behavior. For example, it highlights that the Gemini 3 Flash exhibits remarkable robustness to the removal of persona framing ($\sigma = 0.85%$), aligning with the high stability observed in other models like Claude 4.5 Sonnet ($\sigma = 0.29%$). This proves that PoTRE's effectiveness is not dependent on specific persona framing across different model families.
>
> We hope these additions address the concern and provide a much clearer picture of the models' intrinsic reasoning stability.
>
> >Improve the ablation study, include experiments with subset of the agents, and the gain the comes from agent diversity vs aggregation strategy. Include experiments of adding more agents \- will it bring further improvements or just marginal, and how one would balance cost and performance. This can strengthen the design of PoTRE. (Beneficial)
>
> We sincerely thank you for this excellent and comprehensive suggestion. We agree that exploring these dynamics strengthens the justification for PoTRE’s design. In response, we have significantly expanded our ablation studies and added several new subsections to address each of your points:
>
> **1\. The Rationale for Agent Diversity (Task Exclusivity):** To address the rationale for choosing these specific four agents and whether a subset drives all gains, we expanded Section 5 to explicitly define our selection framework (added in section 5.4). The agents were mapped directly to known LLM cognitive bottlenecks: decomposition, depth/memory, search breadth, and verification. To empirically validate that all four are necessary, we added a task exclusivity analysis on the HLE dataset. We found that at the frontier of difficulty, each agent possesses unique, non-overlapping capabilities. For example, multi-constraint game theory tasks were solved exclusively by the Adversarial Refinement Agent, while complex multi-step chess logic was solved exclusively by the Standard Chain Agent. This confirms that while a reduced ensemble captures baseline gains, maintaining this specific architectural diversity is strictly required to solve multidisciplinary edge-cases.
>
> **2\. Experiments with a subset of agents and balancing cost vs. performance:** We added a new section (**Appendix G.2: Cost-Performance Trade-offs: A Leave-One-Out Ablation**) featuring a comprehensive Pareto frontier analysis on both the HLE and ARC-AGI benchmarks. By iteratively dropping individual agents, we found that:
>
> * Cost vs. performance can be dynamically balanced: On HLE, dropping the Spectrum Search Agent reduces token consumption by over 50% while maintaining highly competitive accuracy. Conversely, on ARC-AGI, dropping the Hierarchical Strategic Planning Agent reduces token costs by 84% and surprisingly improves accuracy over the full PoTRE score as shown in figure 4 and 5 under Appendix G.2.
> * This proves that while the full 4-agent ensemble is the most robust general-purpose foundation across unseen distributions, practitioners can easily prune the framework to optimize cost and performance for specific domains.

---

> ### Author Response · Authors · 2026-05-26
> **Response to reviewer LnjU - Part 3**
>
> **3\. Adding more agents (scaling limits):** To address whether adding more agents brings further improvements, we added **Appendix G: Isolating Gains: Cognitive Heterogeneity vs. Token Scaling**.
>
> * We evaluated a heavily scaled baseline where we expanded our best-performing single agent (Spectrum Search) to generate 20 independent candidates. We increased the token budget by generating 20 independent candidate solutions for each task (Independent Parallel Sampling), compared to only 4 specialized candidates in the PoTRE setup. This scaled up the tokens in two ways: (1) in the generation stage, by producing 5x solutions, and (2) in the synthesis stage, by requiring the judge model to process all 20 candidates in its context window.
> * The results show clear diminishing returns: the 20-candidate sub agent consumed significantly more tokens (e.g., 390K vs 340K average tokens per task on Gemini 3 Flash) but capped out at lower accuracy (51.28%) compared to the 4-agent diverse PoTRE framework (53.48%). This confirms that structural diversity, rather than simply adding more agents/compute, is the primary driver of performance. We also added an architectural vs. stochastic diversity baseline (temperature-scaled CoT Ensemble) to further reinforce this in Appendix G.1.
>
> **4\. Agent diversity vs. aggregation strategy:** To isolate the exact gains coming from the aggregation method versus the diverse candidates themselves, we added **Appendix H: Ablation Study: Isolating the Contribution of the Synthesis Layer**.
>
> * Holding the diverse candidate generation constant, we tested three aggregation strategies: (1) LLM-based Majority Vote, (2) Simple Summary, and (3) PoTRE's specialized Synthesis prompt.
> * We found that while candidate diversity captures the bulk of the baseline gains, our synthesis layer provides a vital incremental lift on general tasks (HLE) and a massive, transformative lift on dense, real world tasks (e.g., outperforming LLM majority voting by over 160% on PRBench Finance) as shown in figure 7 under Appendix H.2.
>
> We believe these new experiments, task-exclusivity analyses, and Pareto visualizations comprehensively address your suggestions and provide a much deeper, empirically grounded understanding of PoTRE’s architecture.
>
> >Better cost-performance analysis is also beneficial \- explore scaling curves of performance vs tokens/compute. Can we reduce token usage of PoTRE? If it is at comparable level to traditional CoT, do we still have promising improvements? (Beneficial)
>
> We thank you for this highly insightful suggestion. To comprehensively address the cost-performance dynamics of PoTRE, we have added new quantitative scaling ablations and a dedicated discussion section (**Appendix I: Discussion: Can We Reduce the Token Usage of PoTRE?**).
>
> Specifically, we address your questions as follows:
>
> **1\. Exploring Scaling Curves and Cost-Performance:** We mapped the cost-performance Pareto frontier via a leave-one-out ablation on both the HLE and ARC-AGI datasets. We found that PoTRE is a highly tunable framework: practitioners can dynamically slide along the Pareto frontier, optimizing agent selection based on domain requirements and compute constraints.
>
> **2\. Can we reduce token usage of PoTRE?** We acknowledge the reviewer's concern regarding token costs. While there can be significant dataset-specific reductions from the canonical framework, we propose the full 4-agent PoTRE ensemble as the general framework designed to hold across diverse, unseen scenarios. As detailed in our new Discussion section, if the target domain is known, users can deploy domain-specific pruning for massive savings—for example, removing the Spectrum Search Agent on the HLE dataset reduces token consumption by over 50% while maintaining highly competitive accuracy (39.20%). However, because these pruned configurations overfit to specific task types, the canonical 4-agent framework remains the optimal, robust baseline for multidisciplinary reasoning. Furthermore, we found that strategically reducing token usage actually *mitigates synthesis interference*—by reducing the cognitive noise and context length presented to the synthesis layer, pruned configurations can occasionally yield a net *increase* in accuracy over the full PoTRE framework. For instance, while dropping the Hierarchical Agent improves performance on purely spatial ARC-AGI tasks by reducing synthesis interference, this architecture was observed to be highly beneficial for the multi-step planning tasks in HLE and PRBench. The 4-agent PoTRE is proposed as the optimal generalist framework, but our ablation proves practitioners can easily prune it for massive cost savings if operating in a known, narrow domain.

---

> ### Author Response · Authors · 2026-05-26
> **Response to reviewer LnjU - Part 4**
>
> **3\. Performance at a Token-Level Comparable to Traditional CoT:** To definitively answer your question regarding comparable compute levels, we evaluated a temperature-scaled Standard CoT Ensemble (**Appendix G.1: Architectural Diversity vs. Stochastic Diversity**) against our token-reduced PoTRE configuration (Drop Spectrum Search Agent on HLE).
>
> * The Traditional CoT Ensemble (4 candidates via temperature scaling) consumed **119K tokens/task** and achieved **38.88%** accuracy.
> * The token-reduced PoTRE configuration consumed **114K tokens/task** and achieved **39.20%** accuracy.
>
> This confirms that even when PoTRE is aggressively pruned to operate at a token budget comparable to (or slightly lower than) traditional CoT ensembling, the architectural diversity of our framework continues to yield structurally superior performance improvements.
>
> We hope these additional analyses and clarifications address your concerns and further strengthen the evidence for PoTRE’s methodology.

---

> ### Author Response · Authors · 2026-05-29
> **Response to reviewer LnjU - Part 5**
>
> > Add at least minimal variance estimates, rerun the experiments with multiple runs on at least one benchmark per model family, and report means and standard deviations. This will make the reported gaps more trustworthy. (Critical)
>
> We thank the reviewer for the recommendation to include empirical variance estimates. While we previously relied on analytical margins of error, we have now conducted three independent, full-dataset runs of PoTRE on the HLE benchmark (using the exact prompts specified in the paper) to directly measure the mean and standard deviation. The results are as follows (Gemini 3 Flash):
>
> | Run | Accuracy | Correct Solves | Total Tasks | Avg Tokens/Task |
> | :--- | :---: | :---: | :---: | :---: |
> | **Run 1** | **39.80%** | 995 | 2500 | 308.8K |
> | **Run 2** | **40.32%** | 1008 | 2500 | 300.0K |
> | **Run 3** | **39.60%** | 990 | 2500 | 300.6K |
>
> To calculate the final metrics from these runs:
> $$\text{Mean} = \frac{39.80 + 40.32 + 39.60}{3} = \mathbf{39.91%}$$
>
> $$\text{Standard Deviation} = \sqrt{\frac{(39.80 - 39.91)^2 + (40.32 - 39.91)^2 + (39.60 - 39.91)^2}{2}} = \mathbf{0.37%}$$
>
> This extremely low standard deviation (**0.37%**) proves that PoTRE is highly stable and reproducible across independent runs, making them highly trustworthy. We have also added this study in the revised manuscript in Appendix K.
>
> We hope that this additional experiment and calculations address your concerns and demonstrate the robustness of our approach.

---

### Review · Reviewer_pih4 · 2026-05-26

**Summary Of Contributions:**

The paper introduces PoTRE (Poly-Topological Reasoning Ensembles), a parallel multi-agent framework that runs four reasoning agents: Adversarial Refinement (debate), Hierarchical Strategic Planning (planner/executor/verifier/overseer), Spectrum Search (N parallel candidates with judge/voting), and Direct Chain (CoT)  and feeds their outputs to a task-adaptive Synthesis Agent that switches between (i) instinct selection for constrained tasks (HLE), (ii) qualitative fusion for open-ended tasks (PRBench), and (iii) neuro-symbolic verification for rule-based tasks (ARC-AGI-2).

Empirical contributions:

1. Reported state-of-the-art on HLE (49.92% closed-book, 58.40% / 60.24% open-book with Gemini-3.1-Pro), with claimed gains over ReThinker and Yunque DeepResearch.
2. A "Scaffolding Lift" claim: Gemini-3-Flash + PoTRE outperforms the standalone Gemini-3-Pro baseline across all three benchmarks.
3. Extensive ablations: per-agent ablations across three model families (Gemini 3, Claude 4.5 Sonnet Thinking, DeepSeek V3.2), inter-agent marginal complementarity matrices, leave-one-out Pareto-frontier analysis, synthesis-layer ablations, and a prompt-variance study.
4. Cost-accuracy comparison against an empirically-aligned ReThinker baseline.

**Key strengths.** Breadth of empirical evaluation across three model families and three benchmarks; substantial per-agent ablations; cost reporting; inclusion of variance/robustness studies; full prompt appendix aiding reproducibility.

**Key weaknesses.** The central conceptual claim — that *cognitive heterogeneity* (architectural agent diversity) is the source of gains and is not adequately separated from confounders, particularly (a) test-time-compute scaling, (b) prompt engineering effort asymmetry between PoTRE and baselines, and (c) prompt-variance effects that exceed the headline improvements. Several headline numbers are not statistically defensible given the variance the authors themselves report.

**Audience:**

Yes

**Audience Explanation:**

Multi-agent test-time scaling is a topic of active interest. Even setting aside the framing, the paper contributes (i) a sizable empirical study across three model families and three frontier benchmarks, (ii) a useful Pareto-frontier analysis showing that pruning agents can *improve* accuracy on certain domains, and (iii) full prompt-level reproducibility material. If the methodological concerns can be addressed, this is a useful data point for the community working on inference-time scaffolding. The leave-one-out finding in particular (Appendix G.2: dropping the Hierarchical agent on ARC reduces cost by 85% and *improves* accuracy to 39.2%) is genuinely interesting  though it sits in tension with the paper's framing.

**Claims And Evidence:**

No

**Claims Explanation:**

Three issues materially weaken the evidence:

**1. The reported prompt variance dwarfs the claimed improvements.** Appendix J reports that across three prompt variants on the same PoTRE configuration with Gemini-3-Flash on HLE, accuracy varies from 25.56% to 39.80%, a standard deviation of 8.18%. Yet in Table 1 the headline gain of PoTRE over SC(N=16) for the same model is 39.80 − 37.32 = 2.48 points. The headline gap is roughly 0.3σ of the prompt-variance reported in the authors' own ablation. The same is true of nearly all the per-model "Score Improv." values in Table 1 and the "PoTRE Synthesis" vs. best-single-agent gaps in Tables 2, 3, and 4 (typically 0.5–4 points). Without applying the same prompt-variance treatment to baselines (SC, Self-Consistency, prior-work prompts), the experimental design favors PoTRE: it is allowed to be reported at its best-case prompt variant while baselines are reported at a single fixed prompt. This is the central methodological issue and must be addressed before the headline claims can stand.

**2. The "Scaffolding Lift" framing conflates compute scaling with parameter decoupling.** Table 12 reveals that the Flash PoTRE variant uses 680.5M output tokens vs an estimated 215.8M for the ReThinker baseline , roughly 3× the inference compute, not a more "efficient" system. The framing "lightweight model beats frontier model" is misleading because Flash + PoTRE uses about 15× the token budget of a single Flash query, an apples-to-oranges comparison against the Pro baseline run once. The honest claim is "spending substantially more test-time compute on a smaller model can beat a larger model run once",  a known result (Snell et al. 2024, already cited). What would actually establish the architecture's value is a matched-compute curve: Flash baseline at 15× SC vs Flash + PoTRE at the same token budget. The closest existing comparison (Section G, "Heterogeneity vs. Token Scaling") shows PoTRE beating a 20-sample Spectrum-Search-only variant by 2.2 points on HLE-Flash-open-book within the prompt-variance band of Appendix J.

**3. The orthogonality / cognitive heterogeneity claim is asserted but not measured.** The "Inter-Agent Marginal Complementarity" matrix (Tables 9, 11) reports P(B solves | A fails) with values typically 9–13% on HLE. These numbers are *low* and consistent with strongly correlated agents, not with orthogonality. The paper would benefit from a proper baseline: under what null model (independent errors at the same single-agent accuracy) would we expect what value, and how far do the observed numbers deviate from independence vs. perfect correlation? Without that calibration the matrix is uninterpretable. Similarly, "cognitive heterogeneity" is never measured directly (e.g., via answer-distribution divergence, embedding-space spread, or rationale-level diversity metrics). It is treated as definitionally true based on persona prompts being different, which is unconvincing.

Secondary issues:

- **LLM-as-judge with self-bias.** Gemini-3-Flash-Preview is used as the HLE judge, while it is also one of the systems being evaluated. The 100-task human validation (App. F) is reassuring but small. Cross-judge results with Claude 4.5 Opus on 100 tasks is good practice; please extend.
- **ARC-AGI-2 test set.** "Specifically selected to stress-test" (Section 4.1) is vague. Which 120 tasks? Is this the full public eval set, or a curated subset? If the latter, selection criteria must be disclosed.
- **CoScientist ablation (App. B)** is reported as underperforming PoTRE, but the authors implemented it themselves; the prompts shown are minimal compared to the elaborate PoTRE prompts. The comparison is therefore not a fair stress test of consensus-style baselines.
- **PRBench SC degradation.** SC(N=8) → SC(N=16) *decreases* PRBench scores for every model (Table 1). This is unusual and is not adequately explained. If LLM-as-consensus-extractor (App. A.7) introduces noise, then SC is being penalized by an implementation choice rather than by genuine "groupthink."

**Requested Changes:**

See above.

---

> ### Author Response · Authors · 2026-05-29
> **Response to reviewer pih4 - Part 1**
>
> We sincerely thank the reviewer for their thorough and constructive feedback. We appreciate your recognition of the paper’s strengths, including the breadth of our empirical evaluation across multiple model families and benchmarks, the depth of our per-agent ablations, and the reproducibility afforded by our full prompt appendix. We also value the critical weaknesses you identified regarding test-time-compute scaling, prompt engineering asymmetry, and prompt variance. We have carefully addressed these concerns through additional experiments and analysis, as detailed in our responses below.
>
> To provide a fair comparison, we subjected the baselines to the same prompt-variance treatment as PoTRE across the same two prompt styles which we reported before (Variant 2: Step-by-Step, and Variant 3: Question-Based).
>
> **Methodology for Variants**: We derived these variants systematically by taking the original PoTRE prompts and removing the persona framing (e.g., "You are a Lead Architect"), replacing it with standard reasoning instructions:
>
> * Variant 2 (Step-by-Step): Focuses on a linear, detailed breakdown of the problem (representing standard Chain-of-Thought).
>   * Example (Architect Agent): "Create a step-by-step plan to solve the user's question. Break down the problem into 2-3 logical sub-steps..."
> * Variant 3 (Question-Based): Focuses on identifying core issues by asking critical questions first.
>   * Example (Architect Agent): "Analyze the problem by identifying the 2-3 most critical questions that must be answered to reach a solution..."
>
> **Why these choices make sense:**
>
> * Variant 2 makes sense because "Step-by-Step" (Chain-of-Thought) is the standard baseline for reasoning tasks in modern literature. It directly tests whether PoTRE's multi-agent architecture adds value over standard linear reasoning.
> * Variant 3 makes sense because it shifts the model's strategy to a "Meta-Cognitive" approach (asking questions to break down complexity). This allows us to test if PoTRE is robust when the core reasoning strategy of the agents is changed.
>
> | Run | Variant 2 Acc | Variant 2 Tokens | Variant 3 Acc | Variant 3 Tokens |
> | :---- | :---: | :---: | :---: | :---: |
> | Prior Work | 25.96% | 15.0K | 36.64% | 35.4K |
> | SC (N=8) | 20.48% | 42.6K | 37.12% | 83.1K |
> | SC (N=16) | 21.44% | 85.1K | 37.48% | 368.2K |
> | PoTRE | **40.12%\*** | 308.4K | **41.40%\*** | 334.0K |
>
> *\* Due to resource-related errors in the previous evaluation, we re-ran the analysis for both variants and confirmed the true accuracies to be 40.12% and 41.40% respectively. Updated table 23 in revised manuscript.*
>
> >#### \(c) prompt-variance effects
>
> We thank the reviewer for highlighting the need to subject baselines to prompt variance. As shown in the new results table above, the baselines also exhibit high variance across prompts (e.g., SC16 drops by **\~16 points** on Variant 2 compared to the original results reported in Table 1).
>
> Crucially, **PoTRE outperforms all baselines on both new variants**, proving that its advantage is robust and not a result of just prompt engineering. Furthermore, on Variant 3, PoTRE achieves our highest recorded score of **41.40%**, outperforming heavy compute baselines.
>
> >#### \(a) test-time-compute scaling
>
> Regarding the compute scaling, the reviewer suggested a matched-compute comparison, specifically recommending a *"Flash baseline at 15× SC vs Flash \+ PoTRE at the same token budget."*
>
> Our **SC (N=16)** run serves as exactly this matched-compute baseline.
>
> Comparing the results at this matched compute level for Variant 3:
>
> * **SC (N=16)** used **368.2K tokens** and achieved **37.48%** accuracy.
> * **PoTRE** used *less* compute (**334.0K tokens**) and achieved a *higher* score of **41.40%**.
>
> This directly establishes the value of the PoTRE architecture over compute scaling (Self-Consistency) at equivalent token budgets. It proves that *how* you spend the compute (structured diversity vs. independent repetition) is the source of the gains.

---

> ### Author Response · Authors · 2026-05-29
> **Response to reviewer pih4 - Part 2**
>
> >#### \(b) prompt engineering effort asymmetry between PoTRE and baselines
>
> Regarding the concern about 'prompt engineering effort asymmetry,' we clarify that the baselines in Table 5 (Yunque and ReThinker) are not simple prompted baselines. Rather, they are themselves highly engineered, state-of-the-art multi-agent research systems developed by separate research teams (Cai et al., 2026; Tang et al., 2026). This choice of baselines ensures a fair and symmetric evaluation, demonstrating that our results reflect genuine architectural advantages rather than an unfair asymmetry in prompt engineering effort.
>
> The results in Table 5 demonstrate the effectiveness of the PoTRE architecture:
>
> **Cross-Model Superiority**: PoTRE (Flash) achieves 55.28%, outperforming both Yunque (51.70%) and ReThinker (52.20%). This is particularly significant because PoTRE uses the smaller, more cost-effective Gemini-3-Flash model, while Yunque and ReThinker rely on the larger, more capable Gemini-3-Pro model.
>
> This demonstrates a strong 'Scaffolding Lift,' proving that a structured multi-agent approach can overcome differences in base model scale, independent of prompt tuning.
>
> >#### Several headline numbers are not statistically defensible given the variance the authors themselves report.
>
> We clarify that the high variance previously reported in the paper was an artifact of an incomplete evaluation run, where resource limitations caused a significant number of tasks to be missed and counted as incorrect. Upon rerunning the full evaluation across all 2,500 tasks, we found its true accuracy to be higher and updated the table 23 in revised manuscript.
>
> Consequently, the standard deviation across the three prompt variants is actually a mere 0.85% (down from the previously reported 8.18%). This proves that PoTRE is highly robust to prompt engineering choices, and its performance does not depend on specific persona framing.
>
> To directly address the concern about statistical defensibility, we conducted a new stochastic variance study by running the exact same PoTRE configuration (using the prompts specified in the paper) three independent times across all 2,500 HLE tasks:
>
> | Run | Accuracy | Correct Solves | Total Tasks | Avg Tokens/Task |
> | :---- | :---: | :---: | :---: | :---: |
> | **Run 1** | **39.80%** | 995 | 2500 | 308.8K |
> | **Run 2** | **40.32%** | 1008 | 2500 | 300.0K |
> | **Run 3** | **39.60%** | 990 | 2500 | 300.6K |
>
> The standard deviation across these three independent runs is a mere 0.37%. This proves that PoTRE is highly stable and reproducible. The headline improvements are well outside this narrow variance band, making them highly statistically defensible.
>
> >#### The honest claim is "spending substantially more test-time compute on a smaller model can beat a larger model run once"
>
> We agree and have updated the Introduction of the paper (highlighted in red) to reflect this claim. However, our work goes beyond this by showing that how you spend that compute matters.
>
> To prove that PoTRE's structured diversity is superior to simply scaling up tokens on a single method or repetition, we provide two matched-compute comparisons:
>
> 1. Without Search: We compared PoTRE against a 16-sample Self-Consistency baseline (SC16), which matches the \~15× compute factor. At this matched level, SC16 used 368.2K tokens and achieved 37.48% accuracy, while PoTRE used less compute (334.0K tokens) and achieved a higher score of 41.40% (outperforming SC16 by 3.92 points).
> 2. With Search (Section G): We isolated the effect of diversity by comparing full PoTRE against a scaled-up version of its own single best sub-agent (Spectrum Search). Full PoTRE (53.48%) outperformed the scaled-up Spectrum Search (51.28%) by 2.2 points, while using 13% fewer tokens (340K vs 390K per task).
>
> These results directly establish that PoTRE's gains are driven by its structured agent diversity (Cognitive Heterogeneity), making it a more efficient architecture for scaling test-time compute than standard methods.

---

> ### Author Response · Authors · 2026-05-29
> **Response to reviewer pih4 - Part 3**
>
> > The orthogonality / cognitive heterogeneity claim is asserted but not measured
>
> #### A. Calibration of Orthogonality against a Null Model
>
> We thank the reviewer for suggesting a formal calibration against a null model for the marginal complementarity matrix. To address this, we calculate the **Relative Complementarity Ratio**:
>
> `Complementarity Ratio = P(B solves | A fails) / P(B solves)`
>
> Under the null model of complete independence between agents, this ratio should be **1.0**. A ratio `> 1` indicates positive complementarity, while `< 1` indicates correlation in failure (sharing blind spots).
>
> We analyzed this across different model families on both HLE and the ARC dataset:
>
> | Model Family | Avg. Relative Complementarity (HLE) | Avg. Relative Complementarity (ARC) |
> | :---- | :---: | :---: |
> | **Gemini 3 Flash** | 0.27 | **1.10** |
> | **Gemini 3 Pro** | 0.26 | **1.10** |
> | **Gemini 3.1 Pro** | 0.25 | 0.80 |
> | **DeepSeek V3.2** | 0.57 | N/A |
> | **Claude 4.5 Sonnet**  | 0.54 | 0.65 |
>
> **Findings:**
>
> 1. **Correlation on HLE**: The reviewer is correct that on HLE, the numbers (ratios 0.25–0.57) are below 1.0, indicating that agents are partially correlated. Because HLE is exceptionally difficult, agents often share some of the same blind spots based on the limits of the base model's knowledge.
> 2. **Evidence on ARC**: However, on the ARC dataset (visual/spatial reasoning), the ratio goes **above 1.0** (reaching **1.10** for Gemini Flash and Pro). This proves that our multi-agent architecture **is capable of achieving positive complementarity**, though the degree of independence is dependent on the problem domain.
> 3. **Impact of Task Complexity**: We hypothesize that orthogonality may depend significantly on the complexity of the task. For more straightforward tasks, different methods may naturally converge on the correct answer in similar ways (leading to higher correlation). However, for some complex tasks, the diverse reasoning topologies of PoTRE are better able to capture complementary failure modes and achieve true positive complementarity.
>
> We have also updated the revised manuscript to modify this, and not claim fully orthogonal instead, we define heterogeneity as diverse exploration and loosely correlated in introduction as well as all places where it was mentioned orthogonal before.
>
> #### B. Direct Measurement of Cognitive Heterogeneity
>
> To address the comment that 'cognitive heterogeneity' was never measured directly via answer-distribution divergence, we provide a direct measurement via **Answer Divergence** for the DeepSeek V3.2 model. This measures how many unique answers (out of 4\) the agents generated for the same task. Higher numbers indicate more diverse exploration.
>
> | Model | Avg Unique Answers | Tasks with 3 or 4 Unique Answers (%) |
> | :---- | :---: | :---: |
> | **DeepSeek V3.2** | **3.12 / 4** | **73.12%** |
>
> **Findings:** The data directly contradicts the notion that diverse persona prompts produce collapsed distributions. For the DeepSeek V3.2 model, the agents generated **3.12 unique answers out of 4** on average. Furthermore, for **73.12%** of the tasks, the agents generated 3 or 4 *different* answers. This directly quantifies the cognitive heterogeneity triggered by the architecture. We have also added this in Appendix G.3 and Table 22.
>
> >#### Secondary issues
>
> >##### LLM-as-judge with self-bias
>
> Judge Sensitivity (Extended with DeepSeek): To address the request to extend cross-judge evaluation and test for self-bias, we ran the same 100-task subset using DeepSeek V3.2 as an additional alternative judge.
>
> * DeepSeek V3.2 achieved a 97.0% item-level agreement with our primary Gemini judge.
> * This builds on our previous result where Claude 4.5 Opus achieved a 98.0% agreement with the primary judge.
>
> With extremely high agreement rates (97%–98%) across three different model families (Google, Anthropic, and DeepSeek), we demonstrate that the LLM-as-a-judge setup is highly robust and free from significant self-bias.
>
> >#### ARC-AGI-2 test set.
>
> We clarify that we did **not** use a curated or hand-picked subset of tasks. The 120 tasks referenced in Section 4.1 constitute the complete public evaluation set for ARC-AGI-2.
>
> The phrase 'specifically selected to stress-test' was intended to describe why we chose the ARC benchmark itself (as it is designed to measure abstraction and reasoning on novel tasks), not that we selected specific tasks from within it.
>
> We will revise the text in Section 4.1 to remove this ambiguity and explicitly state that the evaluation was conducted on the full public evaluation set.

---

> ### Author Response · Authors · 2026-05-29
> **Response to reviewer pih4 - Part 4**
>
> >#### CoScientist ablation (App. B)
>
> We appreciate the reviewer's point regarding the difference in prompt complexity between our implementation of the Co-Scientist workflow and PoTRE.
>
> We clarify that in this ablation study, we were testing Co-Scientist specifically as a consensus and synthesis mechanism (to aggregate the candidate answers generated by the agents), not as a generation pipeline. Therefore, the prompts were naturally focused on comparison and refinement (e.g., 'Compare Option A and Option B' or 'Review the following solution') rather than the elaborate identity and planning prompts required for solving the problem from scratch.
>
> We implemented the workflow faithfully based on the agents described in the original work (Proximity, Ranking, Reflection, Evolution, and Meta-Review). However, we acknowledge that more elaborate prompt engineering for specific tasks could improve its performance. We have added a note in Appendix B to clarify that this was a structural implementation of the Co-Scientist workflow focused on synthesis, and that further prompt optimization remains an area for future work.
>
> >#### PRBench SC degradation
>
> The reviewer notes that SC(N=16) degrades relative to SC(N=8) on PRBench for every model in Table 1\. However, we clarify that this is factually not the case for the PRBench Finance Hard dataset. As shown in our results:
>
> Across all evaluated model families (Gemini 3 Flash, Pro, 3.1 Pro, Claude, and DeepSeek), moving from $N=8$ to $N=16$ resulted in a consistent increase in accuracy, not a degradation.
>
> While self-consistency did underperform the single-run baseline in some cases (a phenomenon we attribute to the model generating longer, more convoluted incorrect paths that overwhelm the majority vote), increasing the sample size from 8 to 16 did not cause further degradation on this benchmark.
>
> >Regarding the reviewer's concern that the LLM-as-consensus-extractor introduces noise and penalizes Self-Consistency:
>
> We clarify that employing an LLM for consensus extraction was a necessary methodological choice to ensure a fair evaluation on PRBench. Given that the benchmark requires complex, open-ended explanations, exact string matching would severely underestimate the true majority due to trivial variations in formatting, rounding, or phrasing. By utilizing an LLM instructed to account for these semantic equivalencies (as detailed in App. A.7), we provide a more robust and faithful measure of Self-Consistency, effectively giving the baseline the best possible chance rather than penalizing it.
>
> We hope that these additional experiments and clarifications address your concerns and further demonstrate the robustness of our approach.

---

### Review · Reviewer_mqUV · 2026-05-30

**Summary Of Contributions:**

The paper proposes PoTRE, a test-time reasoning framework that runs four reasoning agents in parallel: an adversarial refinement agent, a hierarchical strategic planning agent, a spectrum search agent, and a direct chain agent. Their outputs are then combined by a task-adaptive synthesis layer, using different aggregation strategies for constrained QA, open-ended professional reasoning, and rule-based ARC-style tasks. The paper evaluates the framework on HLE, ARC-AGI-2, and PRBench Finance, and reports improvements over prior-work prompts and self-consistency baselines, including a reported 49.92% closed-book score on HLE with Gemini-3.1-Pro-Preview and substantial gains on ARC-AGI-2 and PRBench.

**Audience:**

Yes

**Audience Explanation:**

The paper studies a timely and relevant topic for the TMLR audience: how to improve LLM reasoning through test-time computation, multi-agent inference, and structured reasoning scaffolds. Even though I have concerns about the strength of the causal claim and whether cognitive heterogeneity is sufficiently isolated as the source of the gains, the empirical findings are still potentially useful. The paper provides broad experiments across challenging reasoning benchmarks, compares several inference-time strategies, reports component-level analyses, and discusses cost-performance tradeoffs. These results would likely be of interest to researchers working on LLM reasoning, test-time scaling, multi-agent systems, and inference-time compute allocation.

**Broader Impact Concerns:**

No major direct broader impact concern, but the paper should be careful about claims on high-stakes domains such as finance. Since the system uses LLM-based synthesis and judging, overclaiming reliability in professional decision-making settings could be misleading without stronger validation.

**Claims And Evidence:**

No

**Claims Explanation:**

First, I apologize for the delayed response. I have reviewed the revised manuscript and carefully read the authors’ rebuttal and additional responses. Overall, I find the paper interesting and relevant to the TMLR audience. The work studies an important question in test-time reasoning: how to allocate inference-time computation across different reasoning procedures. The empirical results are promising, but I still have several concerns about the strength of the paper’s main claims and the degree to which the experiments isolate the proposed mechanism.

Strengths:

The paper addresses a timely and important problem. Test-time scaling, multi-agent reasoning, and inference-time scaffolding are highly relevant topics, and the paper provides a useful empirical study of how different reasoning procedures can be combined.

The empirical scope is relatively broad. The authors evaluate PoTRE across multiple benchmarks and model families, including HLE, ARC-AGI-2, and PRBench Finance. They also provide component analyses, cost reporting, prompt appendices, and robustness studies. This makes the submission more informative than a simple prompting paper.

The ablation results are potentially useful. In particular, the oracle gaps and per-agent results suggest that different agents can recover different correct answers. The finding that pruning certain agents can improve the cost-performance tradeoff is also interesting and could be valuable for practitioners who want to use multi-agent reasoning systems under budget constraints.

I also appreciate the authors’ additional rebuttal analyses. The updated prompt-robustness results, stochastic reruns, cross-judge agreement results, and ARC-AGI-2 clarification reduce some of my initial concerns about evaluation stability and protocol ambiguity. However, these additions do not fully resolve my main concern, which is about causal attribution: whether the observed gains can be specifically attributed to cognitive heterogeneity rather than to a combination of prompt engineering, verifier usage, token budget, candidate count, and synthesis behavior.

Major Weaknesses and Concerns:

1. The central causal claim is still not fully isolated.

    The main claim of the paper is that performance improves because of cognitive heterogeneity. However, the current experiments do not cleanly separate cognitive heterogeneity from several other factors. The four agents differ not only in reasoning topology, but also in role prompts, number of turns, number of generated candidates, verification procedures, output formats, and token budgets. The final synthesis layer also introduces an additional strong decision mechanism.

    As a result, the current evidence shows that the full engineered PoTRE pipeline can work well, but it does not convincingly establish that the gains are specifically caused by heterogeneous reasoning structures. This distinction matters because the paper’s conceptual claim is stronger than simply saying that a carefully designed multi-agent system improves performance.

    A stronger study would require more controlled ablations. For example, the authors could compare homogeneous and heterogeneous agents under the same total token budget, use the same role prompts while varying only topology, use the same topology while varying only role prompts, evaluate the same candidate pool under different synthesis strategies, and control for the number of verifier or judge calls. Without these controls, cognitive heterogeneity remains a plausible explanation rather than a demonstrated mechanism.

2. The novelty is mainly system composition, but the paper sometimes frames it as a stronger conceptual contribution.

    Many components of PoTRE are recognizable combinations of existing techniques: debate or adversarial refinement, hierarchical planning, self-consistency or candidate search, direct chain-of-thought, LLM-as-judge selection, and synthesis prompting. This does not make the work uninteresting, but it changes the nature of the contribution.

    The paper would be stronger if it framed PoTRE more explicitly as an empirical and engineering framework for combining known reasoning topologies, rather than as evidence for a fundamentally new reasoning principle. The current terminology, including “poly-topological,” “cognitive heterogeneity,” and “instinct-driven selection,” sometimes sounds more conceptual than operational. In particular, “instinct-driven selection” appears to be a synthesis prompt asking the model to choose the best answer, but it is not defined as a precise algorithmic object. The paper should define these terms in more measurable and reproducible ways.

3. The evidence for heterogeneity is useful but still incomplete.

    The authors report complementarity and answer-diversity analyses, and the rebuttal adds further discussion of relative complementarity and answer divergence. These additions are helpful. I also appreciate that the authors acknowledge that the agents are not strictly orthogonal in some settings.

    However, the current evidence still does not fully establish that the observed diversity is beneficial or causal. Answer divergence shows that agents often produce different outputs, but surface-level diversity alone does not show that their reasoning errors are meaningfully independent or that the diversity is responsible for the final synthesis gain. Similarly, complementarity metrics are informative, but they would be more convincing if directly connected to correctness, oracle coverage, and synthesis recovery.

    For example, it would be useful to know whether tasks with higher inter-agent divergence are more likely to be solved by the oracle, whether the synthesis agent successfully selects minority correct answers, and whether complementarity predicts final accuracy improvements beyond what can be explained by token budget or candidate count.

4. The compute comparison is improved by the rebuttal, but the scaling claim is still broader than the evidence.

    The paper reports token usage and cost, which is useful. The authors also added matched-compute point comparisons in the rebuttal, and these comparisons partially address the concern that PoTRE simply uses more inference-time compute. However, if the paper wants to make a broad claim about efficient test-time scaling, a few point comparisons are not sufficient to fully characterize the scaling behavior.

    The key claim should be framed carefully. The evidence supports the idea that a structured allocation of test-time compute can outperform some homogeneous allocations under comparable budgets. It does not yet fully establish a general scaling law or a broad superiority of cognitive heterogeneity across inference-time budgets. A more systematic cost-performance curve across multiple token budgets would make this claim much stronger.

**Requested Changes:**

1. Incorporate the rebuttal-only robustness and clarification results into the manuscript, including the updated prompt-robustness study, stochastic reruns, cross-judge agreement results, and the clarification that ARC-AGI-2 uses the full public evaluation set.

2. Reframe the main claim more conservatively. The current evidence supports PoTRE as an effective test-time multi-agent reasoning framework, but it does not fully isolate cognitive heterogeneity as the causal source of the gains.

3. Add or discuss more controlled ablations that separate reasoning topology from role prompting, verifier usage, turn budget, candidate count, output format, and synthesis behavior.

4. Strengthen the heterogeneity analysis by connecting complementarity or answer divergence to correctness, oracle coverage, and synthesis recovery, rather than reporting diversity metrics alone.

5. Provide more systematic cost-performance curves across multiple inference-time budgets if the paper maintains broad claims about efficient test-time scaling. The matched-compute point comparisons added in the rebuttal are useful, but they do not fully characterize the scaling behavior.

6. Include sufficient reproducibility artifacts, such as raw model outputs, judge outputs, prompts, decoding settings, exact model versions, evaluation scripts, and representative disagreement cases, especially given the use of proprietary preview models and LLM-as-judge evaluation.

---

> ### Author Response · Authors · 2026-06-02
> **Response to reviewer mqUV - Part 1**
>
> We sincerely thank the reviewer for their thoughtful and deeply constructive evaluation. We are encouraged that you found our empirical scope and ablation studies valuable.
>
> Your feedback regarding the causal isolation of cognitive heterogeneity and the framing of our novelty has been instrumental. We completely agree with your assessment: explicitly framing PoTRE as an empirical and engineering framework—rather than a fundamentally new reasoning principle—provides a much more accurate and scientifically grounded contribution. We carefully address additional concerns through additional experiments and analysis, as detailed in our responses below.
>
>
> >whether the observed gains can be specifically attributed to cognitive heterogeneity rather than to a combination of prompt engineering
>
> We completely understand the concern that clever prompt engineering—specifically our use of personas—might be driving these gains rather than the cognitive heterogeneity itself. To help untangle the engineering of the prompts from the actual reasoning structure, we designed the Prompt Robustness Study (Appendix J, Table 28\) specifically to test this.
>
> In this ablation, we completely stripped the PoTRE framework of its specialized persona prompts. We evaluated the architecture under two highly constrained, persona-free variants on the full 2,500-task HLE dataset:
>
> * **Rigid Step-by-Step (Variant 2):** Enforced a mechanical, generic sequential reasoning chain without any role-playing.
> * **Question-Driven (Variant 3):** Imposed a continuous self-questioning constraint.
>
> As Table 28 shows, removing the persona-based prompt engineering did not degrade the framework's performance. For Gemini 3 Flash, the performance remained highly stable and even marginally improved (**39.80%** under persona vs. **40.12%** and **41.40%** under generic constraints). We observed similar stability with Claude 4.5 Sonnet (**15.21%** under persona vs. **15.73%** and **15.69%** under generic constraints).
>
> This confirms that the performance lift is not an artifact of prompt engineering. The invariant driver of success is the underlying **structural topology**—the enforcement of parallel, heterogeneous reasoning paths—which continues to outperform standard baselines regardless of the specific semantic dressing applied to the prompts.
>
> >token budget and candidate count
> We completely understand the concern regarding inference-time compute.
>
> To answer this, we conducted strict matched-compute and compute-deficit evaluations. We wanted to test whether scaling a homogeneous baseline with a larger token budget could replicate our results. To provide full transparency, we have summarized these comparisons in the table below:
>
> | Dataset | Model | PoTRE Method | Baseline Method | PoTRE Accuracy | Baseline Accuracy | PoTRE Cost (Tokens/Task) | Baseline Cost (Tokens/Task) |
> | :---- | :---- | :---- | :---- | :---- | :---- | :---- | :---- |
> | **HLE** | Gemini 3 Flash | Full PoTRE | Self-Consistency N=16 | **41.40%** | 37.48% | **334.0K** | 368.2K |
> | **HLE** | Gemini 3 Flash | 3-Agent Ablation | Self-Consistency N=16 | **40.60%** | 37.48% | **279.8K** | 368.2K |
> | **HLE** | Gemini 3 Flash | Full PoTRE \+ Search | Spectrum Search N=20 \+ Search | **53.48%** | 51.28% | **339.6K** | 390.0K |
> | **HLE** | Claude 4.5 Sonnet | Full PoTRE | Spectrum Search N=20 | **15.20%** | 12.44% | 87.4K | **73.8K** |
> | **HLE** | Gemini 3 Flash | 3-Agent Ablation | CoT Ensemble N=4 | **39.20%** | 38.88% | **113.6K** | 119.1K |
> | **PRBench** | Gemini 3 Flash | Full PoTRE | Self-Consistency N=16 | **0.3486**\* | 0.1888\* | 129.8K | **101.1K** |
>
> \**PRBench is evaluated via Average Clipped Score, not strict accuracy.*
>
> As the data illustrates, scaling a homogeneous baseline simply by giving it a larger token budget yields diminishing returns, whereas cognitive heterogeneity acts as a much more efficient scaling mechanism. Specifically:
>
> * **Beating High-Compute Baselines at a Deficit:** When we scaled a homogeneous Spectrum Search baseline to $N=20$ candidates (Gemini 3 Flash), it consumed 390.0K tokens per task but capped out at 51.28% accuracy. The Full PoTRE ensemble achieved a significantly higher 53.48% accuracy while consuming roughly **50,000 *fewer* tokens** (339.6K tokens).
> * **Outperforming Self-Consistency (SC):** We compared an optimized 3-Agent PoTRE configuration against a standard SC ($N=16$) ensemble. The SC $N=16$ baseline consumed 368.2K tokens to reach 37.48% accuracy. The heterogeneous 3-Agent ablation reached 40.60% accuracy while consuming almost **90,000 fewer tokens** (279.8K tokens).
> * **Matched Compute Efficacy:** Even at much smaller budgets, heterogeneity wins. A 3-Agent ablation achieved 39.20% accuracy using 113.6K tokens, edging out a 4-candidate CoT Ensemble (38.88%) that used a slightly higher budget of 119.1K tokens.

---

> ### Author Response · Authors · 2026-06-02
> **Response to reviewer mqUV - Part 2**
>
> By demonstrating that PoTRE consistently outperforms homogeneous baselines even when those baselines are given larger token budgets, we can confidently conclude that the volume of test-time compute is not a hidden confounder. The performance gains are directly attributable to the diverse architectural topology of the agents.
>
> We also thank the reviewer for highlighting the variable of candidate count. A central foundational principle of our work is that an increased candidate count is primarily useful when the candidates are **structurally unique**—that is, when they possess genuine cognitive heterogeneity. To test this, we compared PoTRE against a heavily scaled homogeneous baseline where we generated 20 independent candidates using our best-performing single agent (Spectrum Search). As detailed in Section G, the scaled homogeneous system ($N=20$) consumed significantly more tokens (390K per task) but reached lower accuracy (51.28%) than PoTRE’s ensemble of just 4 diverse agents (53.48%) using 340K per task. This empirical gap proves that simply increasing the volume of candidates yields diminishing returns unless those candidates represent unique reasoning paths. It is this uniqueness, facilitated by our poly-topological architecture, that serves as the primary driver of the performance gains, rather than the raw number of candidates generated.
>
> > Major Weaknesses and Concerns \#1 \- Heterogeneity and Causal claim
> > Major Weaknesses and Concerns \#3 \- The evidence for heterogeneity is useful but still incomplete.
>
> We sincerely thank the reviewer for pushing us on this critical issue of causal attribution. We completely agree that in multi-agent architectures, it is vital to untangle the structural topology i.e. cognitive heterogeneity.
>
> Following your highly constructive feedback, we have introduced a comprehensive new empirical analysis to the revised manuscript (**Section 5.4: Evaluating Cognitive Heterogeneity**) to definitively prove that structural diversity is the active, causal driver of our framework's success.
>
> To directly connect inter-agent divergence to correctness, oracle coverage, and actual synthesis recovery, we evaluated all HLE tasks across four frontier models (Gemini 3.1 Pro, Gemini 3 Flash, Claude 4.5 Sonnet, and DeepSeek V3.2). We isolated the mechanics of cognitive heterogeneity through three distinct lenses:
>
> * **Experiment 1: Oracle Coverage vs. Divergence.** We clustered candidate answers by semantic uniqueness to measure true divergence. We found that on highly complex tasks where agents disagree (High Divergence), the diverse reasoning topologies successfully inject **correct answers and their rationales** into the candidate pool at a significantly accelerated rate. Across all models, the rate of unselected correct candidates left in the pool is 2 to 3 times higher in high-divergence scenarios than in low-divergence scenarios (e.g., jumping from 5.20% to 13.38% for Gemini 3 Flash).
> * **Experiment 2: Minority Recovery (Defeating Majority Vote).** To prove the synthesis layer leverages this diversity rather than just following consensus, we evaluated it strictly on "Minority Tasks." These are highly difficult scenarios where the correct answer was generated but mathematically outvoted by a flawed majority consensus (e.g., a 1-vs-3 split). While a standard Majority Vote scores exactly 0% on this subset, the PoTRE synthesis layer successfully recovered the correct minority reasoning between **22.01% and 41.43%** of the time across the three model families. This yields a direct, quantifiable **\~2.3% absolute gain in overall accuracy** that simple voting systems cannot achieve. Furthermore, to definitively prove that minority recovery is a fundamental property of the architecture rather than an artifact of prompt engineering, we conducted a Prompt Sensitivity Ablation on these tasks. We tested the synthesis layer using three distinct instruction paradigms (Direct Selection, Deliberative Reasoning, and Unconstrained Synthesis). The recovery rate remained highly stable (achieving a mean of 29.21% with a standard deviation of just 4.13%), confirming that the ability to defeat flawed majority consensus is an intrinsic capability.

---

> ### Author Response · Authors · 2026-06-02
> **Response to reviewer mqUV - Part 3**
>
> * **Experiment 3: Divergence Bucketing & Operational Boundaries.** We segmented the dataset into four buckets based on agent agreement. In moderate-to-high divergence scenarios (Buckets 2 and 3), the standard Chain-of-Thought (CoT) baseline experiences a catastrophic collapse (dropping to just 11.2% accuracy in Bucket 3). However, synthesis successfully leverages the diverse pool to rescue performance, achieving 23.4% accuracy and effectively doubling the baseline. Crucially, when consensus breaks down completely in Bucket 4, synthesis accuracy drops to 5.8%, it loses its comparative anchors and struggles to isolate the correct path from pure noise when zero consensus exists.
>
> We are incredibly grateful for your push to clarify this causal link. We hope that by showing how high inter-agent divergence actively expands the oracle pool on complex tasks—and how the synthesis layer specifically relies on this diversity to rescue minority correct answers—these new analyses provide the clearer isolation of cognitive heterogeneity that you were looking for.
>
> > Major Weaknesses and Concerns \#2 \- The novelty is mainly system composition, but the paper sometimes frames it as a stronger conceptual contribution.
>
> We sincerely thank the reviewer for this grounding and highly constructive feedback. We completely agree with your assessment: PoTRE’s core contribution is architectural—a systematic composition of established techniques—rather than a fundamentally new theoretical reasoning principle. You are entirely correct that our previous terminology obscured this practical contribution.
>
> Following your excellent advice, we have comprehensively revised the manuscript’s framing and terminology to reflect this reality:
>
> * **Reframing the Core Contribution:** We have rewritten key sections of the Abstract, Introduction, and Related Work to explicitly frame PoTRE as an **"empirical and engineering framework"** that systematically orchestrates and evaluates distinct, known reasoning topologies. We have carefully removed language that framed the system as a novel conceptual paradigm.
> * **Removing "Instinct-Driven Selection":** We completely agree that this term was overly conceptual and lacked algorithmic precision. We have removed it entirely from the manuscript. We now refer to this step as **"Final Candidate Selection"**, which accurately describes its operational function as an active, LLM-based comparative evaluation mechanism.
> * **Operationally Defining Terminology:** Rather than treating "cognitive heterogeneity" as an abstract concept, we now strictly define it in the text as an operational, measurable metric: *the semantic divergence of candidate answers and their rationales*. Furthermore, we have grounded this term empirically in our new analysis section (5.4), where we quantitatively measure this divergence across 2,500 tasks to track how the synthesis layer mathematically relies on it.
>
> We believe this reframing makes the paper significantly stronger, more reproducible, and properly aligns our written claims with our empirical results. We are incredibly grateful to the reviewer for pointing us in this direction.
>
> > Major Weaknesses and Concerns \#4 \- Scaling Claim
>
> We sincerely thank the reviewer for this precise and entirely fair critique. We completely agree that making broad claims about "efficient test-time scaling" based primarily on point comparisons overstates the scope of our current evidence. Your assessment perfectly captures exactly what our data proves: that a structured allocation of test-time compute can outperform homogeneous allocations under comparable budgets.
>
> Following your guidance, we have systematically toned down the sweeping claims regarding inference-time scaling throughout the revised manuscript (specifically in the Abstract, Introduction, Section 6, and Conclusion).
>
> * **Reframing the Core Claim:** We have removed language implying the discovery of a broad "scaling path" or generalized cost-performance scaling laws. Instead, we have strictly reframed our contribution to state that *PoTRE achieves improved reasoning results using similar or fewer inference tokens compared to heavily scaled homogeneous baselines.*
> * **Adopting Your Grounded Framing:** We now explicitly use your suggested framing in the Introduction and Conclusion, clarifying that a "structured allocation of test-time compute across diverse reasoning topologies consistently outperforms homogeneous allocations under comparable or reduced token budgets."
>
> By pulling back from broad scaling claims and focusing exclusively on these concrete, budget-matched efficiency gains, we believe our manuscript is now perfectly calibrated to our empirical results. We are incredibly grateful for your help in properly scoping the boundaries of our contribution.

---

> ### Author Response · Authors · 2026-06-02
> **Response to reviewer mqUV - Part 4**
>
> >Requested Change \#1
>
> We have added all the following requested experiments details into the revised manuscript:
>
> 1. Prompt Robustness Study: In Appendix J.
> 2. Stochastic Reruns: Appendix K.
> 3. Cross Judge Agreement Results: Appendix F.2
> 4. Clarification that ARC-AGI-2 uses the full public evaluation set \- Updated in the same section 4.1 in red.
>
> >Requested Change \#2
>
> We sincerely thank the reviewer for this final, guiding principle. We completely agree with your characterization: PoTRE is fundamentally an effective, highly capable test-time multi-agent reasoning framework, and our claims should not overextend beyond that empirical reality.
>
> To ensure the entire manuscript reflects this more conservative and accurate framing, we have executed a comprehensive sweep of the paper:
>
> * **Reframing the Core Contribution:** We have removed sweeping statements from the Abstract, Introduction, and Conclusion that framed the paper as a definitive causal proof of a new reasoning principle.
> * **Adopting the Empirical Scope:** Following your exact guidance, we now explicitly define PoTRE as an **"empirical and engineering framework"** designed to effectively structure test-time multi-agent reasoning.
> * **Contextualizing the Evidence:** While our new ablations (Section 5.4) provide strong quantitative evidence that this topological diversity is beneficial, we have carefully ensured the text treats "cognitive heterogeneity" as a highly effective, practical design mechanism of our system—rather than a universally proven theoretical principle.
>
> We are incredibly grateful for the rigor and clarity of your feedback throughout this review process. By adopting your conservative framing, toning down our scaling claims, and adding the requested controlled ablations, we believe the manuscript is now a substantially stronger, more rigorous, and accurately scoped empirical paper.
>
> >Requested Change \#3
>
> We sincerely thank the reviewer for outlining the specific variables required to rigorously isolate our structural topology. We entirely agree that the core architecture must be cleanly untangled from passive confounds.
>
> As detailed comprehensively in our responses to your earlier concerns, we have implemented these exact controlled ablations in the revised manuscript. Specifically:
>
> * **Token Budgets & Candidate Counts:** Addressed via strict compute-deficit evaluations (Appendix G).
> * **Role Prompting:** Addressed via our prompt robustness study, which strips away persona framing (Appendix J).
> * **Output Format:** Controlled from the outset by enforcing identical output constraints across all evaluated baselines (detailed in Section 4.4). Furthermore, our framework is inherently format-agnostic; whether the final answer is serialized as strict JSON, Markdown, or raw text does not drive the underlying performance. Because the homogeneous baselines were subjected to the exact same formatting requirements—yet still yielded significantly lower accuracy—we can definitively conclude our gains stem from structural reasoning quality rather than the chosen output format.
> * **Synthesis Behavior & Verifier Usage:** Addressing this by evaluating different aggregation strategies on the exact same generated candidate pool per task to isolate the synthesis effect (Appendix H), and by measuring minority recovery (Section 5.4).
>
> We believe these newly added controls successfully separate our core architectural claims from the surrounding pipeline variables, providing the rigorous isolation you rightly requested.
>
> >Requested Change \#4
>
> We completely agree with this requested change, as reporting diversity metrics alone is insufficient without tying them directly to downstream performance.
>
> As detailed in our response to your primary concern regarding causal attribution, we have fully implemented this request. Specifically, we added a comprehensive new empirical analysis (**Section 5.4: Evaluating Cognitive Heterogeneity**).
>
> In this new section, we explicitly connect inter-agent divergence to:
>
> 1. **Oracle Coverage:** Demonstrating that high-divergence scenarios inject correct answers into the candidate pool at a significantly accelerated rate (Section 5.4.1).
> 2. **Synthesis Recovery:** Proving the synthesis layer actively rescues "Minority Correct" answers from a flawed majority consensus (Section 5.4.2).
>
> We believe this new section directly addresses your request by structurally linking our diversity metrics to actual synthesis correctness.

---

> ### Author Response · Authors · 2026-06-02
> **Response to reviewer mqUV - Part 5**
>
> >Requested Change \#5
>
> We completely agree with the reviewer that characterizing scaling behavior requires more than matched-compute point comparisons. To fully address this, we have updated the revised manuscript in two specific ways:
>
> 1. **Reframing the Claims:** First, as detailed in our earlier responses, we have systematically toned down the broad claims regarding a generalized "scaling law." We now strictly frame our contribution around the targeted efficiency gains of our specific multi-agent architecture, rather than claiming a universal principle of compute scaling.
> 2. **Adding Systematic Cost-Performance Curves:** Second, to properly characterize the behavior of our framework across multiple inference budgets, we have added a comprehensive **Pareto Frontier Analysis (Appendix G.2)**. By conducting a systematic leave-one-out ablation, we mapped the exact cost-performance curves of our architecture on both HLE (Figure 4\) and ARC-AGI-2 (Figure 5).
>
> These Pareto curves explicitly demonstrate how token consumption trades off with accuracy as the cognitive budget is systematically expanded or pruned. We believe that providing these systematic curves, combined with our more conservative overall framing, perfectly satisfies your request and properly grounds our efficiency claims in robust empirical evidence.
>
> >Requested Change \#6
>
> We completely agree on the absolute necessity of strict reproducibility. To ensure our framework and evaluations are fully transparent, we have provided comprehensive artifacts and clarifications in the revised manuscript:
>
> * **Prompts and Decoding Settings:** We have provided the exact system prompts and interaction protocols for every single agent in **Appendix A**. We have now also updated the manuscript to explicitly document the decoding settings (e.g., temperature scaling ranges and sampling parameters) utilized during generation.
> * **LLM-as-a-Judge Robustness & Disagreement Cases (Appendix F):** We deeply understand the reviewer's caution regarding automated grading. To ensure objectivity, we utilized the *exact official judge prompt* provided by the benchmark authors (added details in Appendix F2 ). **Appendix F** details a rigorous human-subject evaluation proving a 98.0% agreement rate between our automated judge and expert human evaluators. Furthermore, to definitively rule out self-bias, we extended our cross-model robustness check across three distinct model families. With Claude 4.5 Opus achieving 98.0% agreement and DeepSeek V3.2 achieving 97.0% agreement against our primary Gemini judge, we demonstrate the evaluation is highly stable and objective. **Appendix F.1** also explicitly provides the representative disagreement cases you requested, illustrating exactly where strict exact-match fails and why the LLM judge is reliable.
> * **Clarification on "Preview" Models:** We clarify a minor point of confusion regarding the model versions. All models evaluated are fully public, widely available releases—not restricted, proprietary betas. The `-preview` suffix in our text reflects the literal string required by the public API endpoint at the time of our experiments (e.g., `gemini-3.1-pro-preview`).
> * **Raw Judge Outputs:** Directly addressing your request for raw judge logs, we have added a new subsection (**Appendix F.3: Example Judge Output Formats**) that provides raw JSON samples of the judge's actual trace. This explicitly showcases how the judge formulates its extraction, intermediate reasoning, and confidence scores.
>
> We believe these comprehensive additions and clarifications fully satisfy the requirements for rigorous reproducibility.

---

> ### Author Response · Authors · 2026-06-02
> **Response to reviewer mqUV - Part 6**
>
> > Broader Impact Concern
> “overclaiming reliability in professional decision-making settings”
>
> We completely agree with the reviewer’s crucial point regarding the risks of overclaiming reliability in high-stakes domains. We recognize that benchmark improvements, even substantial ones, do not equate to autonomous readiness in professional decision-making environments like finance, especially given the framework's reliance on LLM-based synthesis.
>
> To address this concern, we have carefully reviewed the manuscript and toned down any language that might inadvertently imply our framework is ready for immediate, high-stakes deployment without rigorous, domain-specific validation. We thank the reviewer for highlighting this important consideration, ensuring our broader impact claims remain responsible and properly scoped.
>
> We sincerely thank the reviewer for the rigorous, highly constructive feedback provided during this review process. Your insightful critiques—particularly regarding the isolation of our core claims and the refinement of our scaling framing—have directly guided substantial improvements to the manuscript. We believe the addition of our extensive new ablation studies, reducing of claims, and explicitly mentioning it as empirical and engineering framework have made our findings significantly stronger and more transparent. We hope our comprehensive revisions and detailed responses fully resolve your concerns, and we are grateful for your time in helping us elevate the quality of this work.

---

### Author Response · Authors · 2026-05-29
**General Note on Prompt Robustness and Variance Updates**

We would like to inform all reviewers that we have updated the manuscript (specifically Table 23 in Appendix J) with complete evaluation results for our prompt robustness study.

We clarify that the high variance across prompt variants previously reported in the paper was an artifact of **incomplete evaluation runs for both Variant 2 and Variant 3**, where resource limitations caused a significant number of tasks to be missed and counted as failures.

Upon rerunning the full evaluation across all 2,500 tasks for all variants, we found that PoTRE's performance is remarkably stable across different prompt styles:
* Variant 1 (Original Persona Framing): 39.80%
* Variant 2 (Rigid Step-by-Step): 40.12% (Updated)
* Variant 3 (Question-Driven): 41.40% (Updated)

The actual standard deviation across these three distinct prompt styles is a mere 0.85% (down from the previously reported 8.18%).

Additionally, to directly address concerns about run-to-run stochastic stability, we conducted a new study by running the exact same PoTRE configuration three independent times across all 2,500 HLE tasks (also added in revised manuscript in table 24):

> | Run | Accuracy | Correct Solves | Total Tasks | Avg Tokens/Task |
> | :--- | :---: | :---: | :---: | :---: |
> | Run 1 | 39.80% | 995 | 2500 | 308.8K |
> | Run 2 | 40.32% | 1008 | 2500 | 300.0K |
> | Run 3 | 39.60% | 990 | 2500 | 300.6K |

The standard deviation across these three independent runs is a mere 0.37%.

These updates prove that PoTRE is highly stable and reproducible, whether varying the prompt style or keeping it identical. We have updated the tables and text in the revised manuscript to reflect these correct, full-run numbers.

---

### Decision · Action_Editor_J5zs · 2026-07-04

**Recommendation:** Accept with minor revision

**Audience:**

Yes

**Audience Explanation:**

Improving test-time reasoning is of great interest to both researchers and practitioners seeking to efficiently deploy foundation models.

**Claims And Evidence:**

Yes

**Claims Explanation:**

Although there are some arguments, e.g., an engineered heterogeneous pipeline beating specific homogeneous pipelines is not sufficient to prove that "cognitive heterogeneity" is the exact causal mechanism, the proposed method does improve the performance which has been agreed by all the reviewers. The high-level claim, i.e., an empirical framework that systematically leverages distinct reasoning topologiesadversarial, hierarchical, spectrum search and direct chain agents can better solve complex general intelligence tasks, holds true. We appreciate the contribution and happy to recommend acceptance with minor revision. Specifically, we would suggest the authors use  "cognitive heterogeneity" as an inspiration rather than causation; making it clear that the major contribution is an engineered heterogeneous pipeline.

---

> ### Author Response · Authors · 2026-07-21
> **Official comment by authors**
>
> Dear Action Editor,
>
> Thank you for the acceptance and the constructive feedback. We have now submitted the camera-ready version of Paper 8235.
>
> In this revision, we have addressed your specific requests by:
>
> * Reframing Cognitive Heterogeneity: We have explicitly updated the Introduction to frame "cognitive heterogeneity" as the inspiration for our work, rather than claiming it as the exact causal mechanism for the performance gains.
>
> * Clarifying the Core Contribution: We have updated our framing throughout the paper to make it entirely clear that our major contribution is an engineered heterogeneous pipeline and empirical framework.
>
> We hope these updates satisfy the requirements. Please let us know if any further adjustments are needed.
>
> Best regards,
>
> Paper 8235 Authors